# A linear nonribosomal octapeptide from *Fusarium graminearum* facilitates cell-to-cell invasion of wheat

Lei-Jie Jia[1], Hao-Yu Tang [2], Wan-Qiu Wang[1,3], Ting-Lu Yuan[1], Wan-Qian Wei[1,3], Bo Pang[2,3], Xue-Min Gong[3], Shou-Feng Wang[2], Yu-Jie Li[1,3], Dong Zhang[1], Wen Liu[2] & Wei-Hua Tang [1]

*Fusarium graminearum* is a destructive wheat pathogen. No fully resistant cultivars are available. Knowledge concerning the molecular weapons of *F. graminearum* to achieve infection remains limited. Here, we report that deletion of the putative secondary metabolite biosynthesis gene cluster *fg3_54* compromises the pathogen's ability to infect wheat through cell-to-cell penetration. Ectopic expression of *fgm4*, a pathway-specific bANK-like regulatory gene, activates the transcription of the *fg3_54* cluster in vitro. We identify a linear, C-terminally reduced and D-amino acid residue-rich octapeptide, fusaoctaxin A, as the product of the two nonribosomal peptide synthetases encoded by *fg3_54*. Chemically-synthesized fusaoctaxin A restores cell-to-cell invasiveness in *fg3_54*-deleted *F. graminearum*, and enables colonization of wheat coleoptiles by two *Fusarium* strains that lack the *fg3_54* homolog and are nonpathogenic to wheat. In conclusion, our results identify fusaoctaxin A as a virulence factor required for cell-to-cell invasion of wheat by *F. graminearum*.

[1] National Key Laboratory of Plant Molecular Genetics, CAS Center for Excellence in Molecular Plant Sciences, Institute of Plant Physiology and Ecology, Chinese Academy of Sciences, 300 Fenglin Road, 200032 Shanghai, China. [2] State Key Laboratory of Bioorganic and Natural Products Chemistry, Center for Excellence in Molecular Synthesis, Shanghai Institute of Organic Chemistry, Chinese Academy of Sciences, 345 Lingling Road, 200032 Shanghai, China. [3] University of the Chinese Academy of Sciences, 200032 Shanghai, China. These authors contributed equally: Lei-Jie Jia, Hao-Yu Tang, Wan-Qiu Wang. Correspondence and requests for materials should be addressed to W.L. (email: wliu@sioc.ac.cn) or to W.-H.T. (email: whtang@sibs.ac.cn)

*F*usarium graminearum is a major fungal pathogen that is responsible for a number of devastating and hazardous diseases, including Fusarium head blight, crown rot and seedling blight on wheat (*Triticum aestivum*) and barley (*Hordeum vulgare*), and Gibberella stalk and ear rot on maize (*Zea mays*)[1–3]. Epidemics of Fusarium head blight occur frequently in the USA, the UK, and China, causing billions of dollars' losses not only from yield and quality reduction but also from the contamination of trichothecene mycotoxins, which threaten the health of humans and animals[4,5]. Sources for Fusarium head blight resistance are limited, and thus far only one resistance gene, *PFT*, has been identified in wheat[6], and no fully resistant cultivars are yet available[1,7]. Controlling *F. graminearum*-caused diseases will benefit from an in-depth understanding of how *F. graminearum* spreads inside the host.

Benefiting from the release of the well-annotated genome sequence of *F. graminearum* strains[8], research groups worldwide have identified more than 200 *F. graminearum* genes that are required for its full virulence on wheat, barley, and/or maize[9]. Most of these virulence genes encode fungal intracellular proteins, including transcription factors[10], protein kinases[11], phosphatases[12], Rab GTPases[13], and primary metabolism enzymes[9]. However, our knowledge concerning the factors secreted by *F. graminearum* that directly interact with and impact host plant cells remains limited, with only a secreted lipase FGL1[14] and trichothecene secondary metabolites (deoxynivalenol and nivalenol)[15,16] having been identified to date. FGL1 has been shown to inhibit callose deposition in wheat spikes through the release of polyunsaturated free fatty acids[17]. Trichothecenes bind to eukaryotic ribosomes and inhibit peptidyl transferase activity leading to protein synthesis inhibition[18]. In wheat spikes specially, trichothecenes inhibit cell wall thickening in the rachis node at the base of inoculated florets, which allows the fungus to spread from one floret to another[19,20].

As in many other fungi, *F. graminearum* possesses genes that are involved in secondary metabolite biosynthesis (SMB). These genes are organized into clusters[21], and many of them encode classic SMB-related enzymes such as non-ribosomal peptide synthetases (NRPS), polyketide synthases (PKS), or terpene cyclases (TC). The *F. graminearum* genome contains genes that code for 19 NRPSs, 15 PKSs and 7 TCs, and 67 putative SMB gene clusters[22,23]. Few clusters have been correlated with their biosynthetic products, including trichothecenes, intracellular and extracellular siderophores[24,25], zearalenone[26], fusarin C[27], aurofusarin[28], and fusaristatin A[29]. Thus far, only trichothecenes and extracellular siderophores, have been reported to be required for *F. graminearum* virulence[24,25]. Based on the number of gene clusters identified, *F. graminearum* has the potential to produce more kinds of secondary metabolites that contribute to virulence.

Characterization of these metabolites and their associated functions has been hampered in part by the absence of detectable expression of many cluster genes under the tested conditions[30]. For example, the expression of two annotated NRPS genes (*nrps5* and *nrps*9) that are linked on chromosome 3 was not detected during in vitro culturing using various media (such as complete medium, carbon starvation medium, nitrogen starvation medium, and carrot medium)[31,32]. Using laser microdissection combined with microarray analysis, we have previously identified *nrps5*, *nrps9* and six adjacent genes that are located in a 54 kb region and coexpressed during infecting coleoptiles of wheat seedlings[33]. The eight genes form *fg3_54*, a putative SMB cluster, the product of which remains elusive. This cluster has also been reported as C64[22], and its expression was detected during wheat spike infection[34]. Individual deletion of two member genes within the *fg3_54* cluster caused reduced disease symptom when inoculated onto wheat coleoptiles and spikes[33], suggesting that the *fg3_54* cluster might be involved in *F. graminearum* virulence.

Here, we describe the roles of the gene cluster *fg3_54* in *F. graminearum* virulence, uncover its cluster-specific regulator gene that enables in vitro constitutive expression of the *fg3_54* cluster, and identify a linear octapeptide as a product of *fg3_54* that facilitates *Fusarium* cell-to-cell hyphal progression in wheat, associated with suppression of host cell defense responses.

## Results

**Deletion of the *fg3_54* cluster reduces wheat infection ability.** We have previously documented that gene expression profile changes in *F. graminearum* during infection of wheat coleoptiles occur in a stage-specific manner[33]. Notably, the transcription of eight particular genes, which are closely linked on chromosome 3 to form a cluster named *fg3_54* (Fig. 1a), was almost not detected during in vitro growth but was greatly upregulated during wheat infection at ~64 h post inoculation (hpi) (Supplementary Fig. 1). *Fg3_54* contains two multifunctional structural genes, i.e., *nrps5* (FGSG_13878) and *nrps9* (FGSG_10990), both of which encode putative NRPSs[23]. Further quantitative PCR results showed that the expression of *nrps5* and *nrps9* was reduced after 72 hpi (Supplementary Fig. 1). The other six genes are *fgm5* (FGSG_10995), *fgm4* (FGSG_10994), *fgm3* (FGSG_10993), *fgm2* (FGSG_10992), *fgm1* (FGSG_10991), and *fgm9* (FGSG_10989) (Fig. 1a). *Fgm5* encodes an ABC transporter that functions in fungal virulence and azole tolerance[35]. The deduced protein of *fgm2* shows 56% identity to a polysaccharide deacetylase from a black yeast-like fungus *Aureobasidium namibiae*. We have previously shown that individual gene knockout mutants of the two member genes, *fgm5* and *fgm2*, had reduced virulence in wheat infection[33]. However, whether the cluster, in its entirety, functions in wheat infection remains to be explored.

To validate the necessity of the *fg3_54* cluster for wheat infection, we generated multiple-independent transgenic *F. graminearum* mutants by either deleting the entire *fg3_54* cluster (FG-Δ*fg3_54*) or individually inactivating the member genes (i.e. FG-Δ*fgm4*, FG-Δ*fgm3*, FG-Δ*fgm1*, FG-Δ*fgm9*, and FG-Δ*nrps5*) (Supplementary Fig. 2). *Fgm4* encodes an unknown protein containing ankyrin repeats, which are common protein–protein interaction motifs. The deduced protein of *fgm3* shows 75% overall identity to an annotated 2-isopropylmalate synthase from the plant anthracnose pathogen *Colletotrichum gloeosporioides*. *Fgm1* encodes a putative cytochrome P450 monooxygenase CYP625A1 (E-class)[36]. The deduced protein of *fgm9* shows 41% identity to an annotated short-chain alcohol dehydrogenase from a fungal pathogen, *Colletotrichum fioriniae*. FG-Δ*fg3_54* and FG-Δ*fgm4* mutants, along with Δ*nrps9*, Δ*fgm2* mutants[33], grew without detectable defects in vitro under all tested culture conditions, in aspects of radial growth and colony and conidial morphology (Supplementary Fig. 3a). These mutants also grew in a manner similar to wild type when treated with hydrogen peroxide or the iron chelator 2,2'-dipyridyl (Supplementary Fig. 3b).

Compared with wild-type strains, FG-Δ*fg3_54*, FG-Δ*fgm4*, FG-Δ*fgm3*, FG-Δ*fgm1*, and FG-Δ*fgm9* mutants all produced fewer spikelets with blight symptoms on two susceptible wheat cultivars (Zhongyuan 98–68 and Bobwhite), measured at 14 days after floret inoculation of spore suspensions (Fig. 1b and Supplementary Fig. 4). *Fgm4*-complemented strains and knockout mutants for a gene outside but adjacent to the *fg3_54* cluster (FGSG_10996) caused a similar number of symptomatic spikelets to wild-type strains (Fig. 1b and Supplementary Fig. 4). Microscopic observation showed that, at 2.5 days post inoculation (dpi), the hyphal density and invaded area in the paleae of the

florets inoculated by *FG-Δfg3_54* mutant were lower and smaller than in those inoculated by the wild type. In addition, *FG-Δfg3_54* mutant hyphae often stopped at concave–convex edges of epidermal cells, while wild-type hyphae often penetrated the epidermal cell wall (Supplementary Fig. 5a). At 8 dpi, hyphae of *FG-Δfg3_54* mutant strains grew in the inoculated floret tissues similarly to the wild type, but no mutant hyphae grew in the rachis at the base of the inoculated floret while wild-type hyphae were abundant (Fig. 1c). These results demonstrate that the *fg3_54* cluster contributes to *F. graminearum* virulence in wheat head blight infection and that loss of the *fg3_54* cluster leads to delayed hyphal spreading.

To perform a more detailed analysis of the deletion strains, we then used a previously established wheat coleoptile infection system[33]. When wounded coleoptiles of 3-day-old wheat seedlings were inoculated with wild-type *F. graminearum*, dark brown-to-black 1.2–1.5-cm-long lesions were observed at 7 dpi (Fig. 1d, e). Inoculation with independent transgenic strains of *FG-Δfg3_54* resulted in lesions that were less than 0.2 cm long, which was only one tenth the size of those caused by the wild-type strains. Individual knockout mutants of the adjacent genes, FGSG_10996 and FGSG_13879, caused lesions that were similar in size to wild type (Fig. 1d). Excluding *FG-Δfgm3*, the other seven mutants caused significantly smaller lesions than the wild-type strains to varying degrees. The sizes of lesions caused by the *FG-Δfgm2, FG-Δfgm1,* and *FG-Δfgm9* mutants were 40%–60% the size caused by the wild-type strain (Fig. 1d). The average lesion size caused by *FG-Δnrps9* mutant strains was ~30% the size caused by the wild-type strain. The *FG-Δfgm5* and *FG-Δfgm4* mutants, and the N-terminal partial deletion mutants of *nrps5* (*FG-Δnrps5*), caused very small lesions, with sizes similar to those caused by *FG-Δfg3_54* (Fig. 1d).

In addition to wheat infection, *F. graminearum* can also cause maize stalk rot. In maize stalk, however, lesions caused by *FG-Δfg3_54, FG-Δfgm4, FG-Δfgm1,* or *FG-Δfgm9* mutants were similar to those caused by wild type (Supplementary Fig. 4c, d). This finding is consistent with the absence of expression of *fg3_54* cluster member genes during maize stalk infection up to 6 dpi (Supplementary Fig. 1)[37]. These results indicate a specific role of *fg3_54* in wheat infection.

**Lacking *fg3_54* leads to a deficiency in cell-to-cell invasion**. We then evaluated the effects of various *fg3_54*-related mutants on fungal invasion process *in planta* and the associated plant defense responses at a cellular level. A monomeric red fluorescent protein (mRFP) and an *Anemonia majano* cyan fluorescent protein (AmCyan) were used to label *F. graminearum* hyphae[38], yielding the fluorescent derivatives of the wild-type strains (*FG-WT-RFP* and *FG-WT-AmCyan*) and the fluorescent derivatives of the mutants (*FG-Δfg3_54-RFP, FG-Δfg3_54-AmCyan, FG-Δfgm4-RFP* and *FG-Δfgm5-AmCyan*) (Supplementary Table 1,2). Strains *FG-Δfg3_54-RFP, FG-Δfgm4-RFP,* and *FG-Δfgm5*-AmC-yan caused small lesions on wheat coleoptiles, similar to *FG-Δfg3_54, FG-Δfgm4* and *FG-Δfgm5* respectively (Fig. 2a–e). After inoculation at the wounded edge of wheat coleoptiles, spores of *FG-Δfg3_54-RFP, FG-Δfgm4-RFP* and *FG-WT-RFP* germinated similarly to form germ tubes by 0.5 dpi, and the germ tubes further extended and developed similarly into hyphae. Their hyphae also grew similarly in the wounded site within 1 dpi (Fig. 2c). The hyphal fronts of both wild type and mutants grew in coleoptile cells, reaching 300–400 μm away from the wounded edge (Fig. 2d). This length is approximately the maximum length of one epidermal cell of a wheat coleoptile. At 1 dpi, macroscopic symptoms (lesion) were inconspicuous on coleoptiles inoculated with *F. graminearum*. Microscopic analysis also showed that cell

wall autofluorescence (which appears green under the GFP filter) could reveal the contours of plant cells surrounding invading hyphae of both wild-type and mutant strains (Fig. 2c).

Obvious differences between wild type and mutants were observed at later stages. *FG-WT-RFP* and *FG-WT-AmCyan* hyphae spread ~3 mm by 3 dpi from the wounded edge through cell-to-cell invasion (Fig. 2c, Supplementary Fig. 5b) and caused lesion larger than 10 mm by 7 dpi (Fig. 2e). In contrast, the hyphae of *FG-Δfg3_54-RFP, FG-Δfgm4-RFP,* and *FG-Δfgm5-AmCyan* mutant strains were arrested in host cells close to the inoculation site and failed to penetrate into neighboring cells over time (Fig. 2c, Supplementary Fig. 5b). Strikingly, the walls of plant cells surrounding mutant hyphae were drastically thicker and displayed stronger autofluorescence, as compared to those surrounding the wild-type hyphae (Fig. 2c). These cell wall invaginations resemble the structure of papillae, which are known to be an early plant defense response to other fungal pathogens[39]. The size of the papillae varies up to 5 μm in-depth and 6 μm in width (see yellow arrowheads in Fig. 2c and Supplementary Fig. 5b). Moreover, a hole-like structure with reduced fluorescence was often observed in the center of the invaginations (Fig. 2c, Supplementary Fig. 5c).

In addition, in comparison to the mutant hyphae, *FG-WT-RFP* and *FG-WT-AmCyan* hyphae were more often observed crossing plant cell walls, which can be interpreted as invasion into the adjacent plant cells after colonization of the former ones (Fig. 2c and Supplementary Fig. 5b). For a better quantitative comparison of *F. graminearum* hyphal invasion, here we use the term 'penetration ratio', which was calculated by dividing the number of penetration events (e.g., one event is defined as one hypha observed crossing a plant cell wall), by the number of plant cells within a confocal viewing area. At 3 dpi, the wild-type derivative FG-WT-RFP had a penetration ratio of ~3, which was much higher than that (~0.2) determined for the mutant derivatives *FG-Δfg3_54-RFP, FG-Δfgm4-RFP* or *FG-Δfgm5-AmCyan* (Fig. 2f), indicative of a significant reduction in invasiveness due to *fg3_54* deletion. Consistent with these results, the fungal biomass of the mutants and *FG-WT-RFP* (as estimated by quantitative PCR after reverse transcription of fungal tubulin RNA relative to wheat reference RNA), although similar at 1 dpi, differed significantly at 3 dpi, i.e., the fungal biomass of *FG-WT-RFP* was more than twofold greater than the mutants (Fig. 2g).

Although not previously demonstrated in *F. graminearum* infection of wheat coleoptiles, the formation of papillae is a common plant response to the fungal penetration attempt, and papillae typically contain callose[40]. Using aniline blue, a callose stain, we found that uninoculated or mock-inoculated control coleoptile cells were poorly labeled and that stained regions were located primarily at the wounded edge as discrete foci (Supplementary Fig. 5c, d). Similar weak callose staining signals were found when staining *FG-WT-RFP*-inoculated coleoptile cells at 3 dpi (Supplementary Fig. 5c, d). In contrast, *FG-Δfg3_54-RFP*-inoculated coleoptiles showed strong callose signals, particularly at cell wall invagination regions in the plant cells neighboring those containing hyphae (Supplementary Fig. 5c). These callose signals often formed continuous lines along the hyphal fronts. These results indicate that a host defense response is more profoundly evoked when *F. graminearum* lacks *fg3_54*, and that wild-type *F. graminearum* inhibits callose deposition during its invasion.

In summary, and as shown in Fig. 2h, *F. graminearum* lacking the *fg3_54* cluster or its key member genes showed no defects in spore germination and hyphal growth in wounded plant cells by 1 dpi, but later failed to penetrate plant cell walls to invade adjacent intact plant cells by entering the cell-to-cell invasion mode. This observation is consistent with the observation that *fg3_54* gene

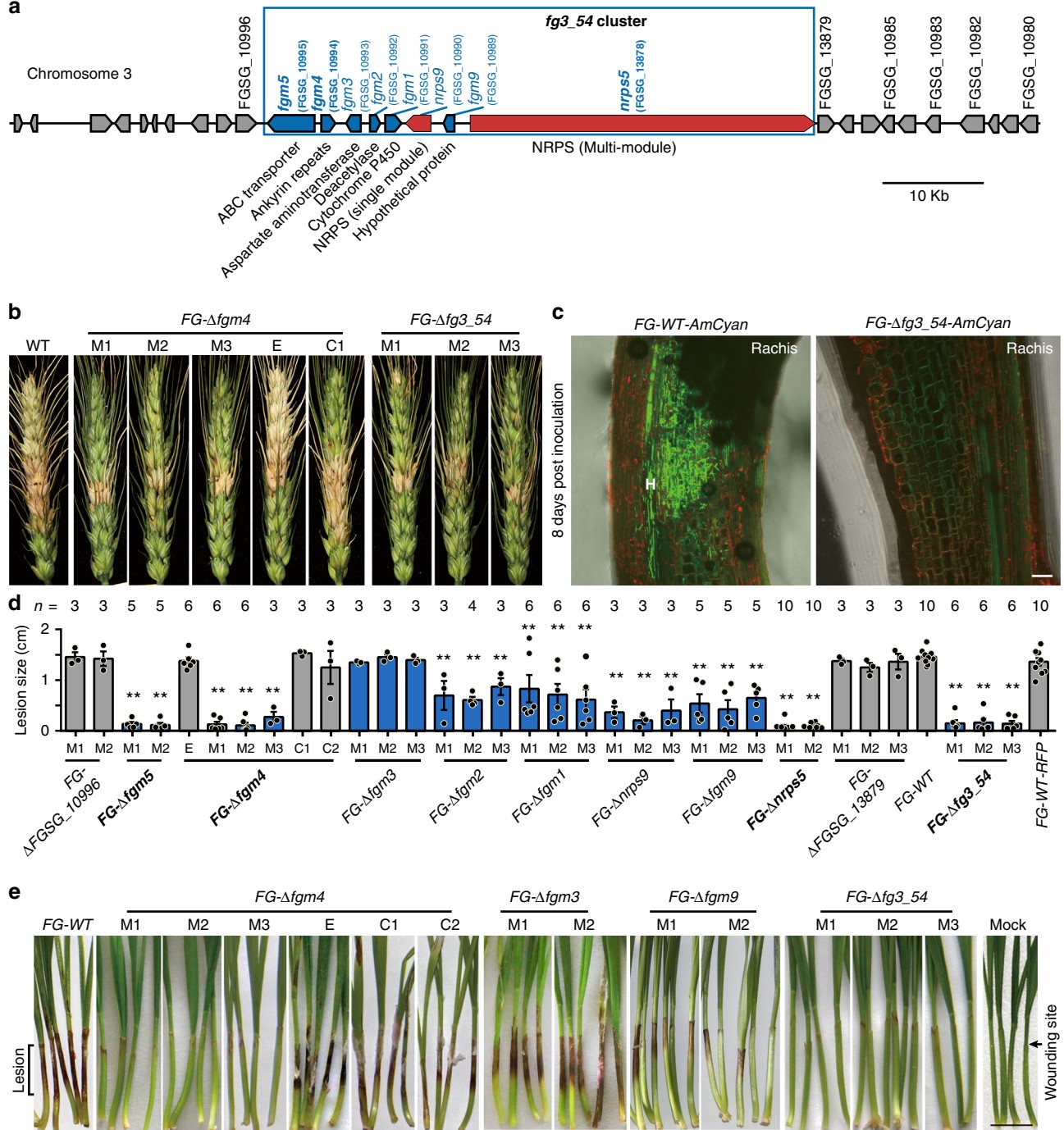

**Fig. 1** *Fg3_54* cluster deletion impairs the wheat infection ability of *F. graminearum*. **a** The *fg3_54* gene cluster and flanking gene composition.
**b** Representative images of wheat spikes at 14 days after inoculation with the indicated strains. **c** Microscopic images of the rachis at the base of the inoculated floret at 8 days after inoculation with the fluorescently tagged wild type or mutant strains. Note *FG-WT-AmCyan* hyphae are present in the rachis, but the mutant is absent. **d** Chart showing the lengths of lesions on wheat coleoptiles at 7 days after inoculation with indicated strains. Individual data points are overlaid with the corresponding bar graph. Error bars indicate the standard error of the mean (s.e.m.), *n* denotes number of independent experiments. The asterisk indicates a significant decrease in comparing to wild type (WT) with the two-tailed unpaired Student's *t*-test (*$P < 0.05$; **$P < 0.01$). Source data are provided in Supplementary Data 3. **e** Representative images of wheat seedlings at 7 days after inoculation with the indicated strain. M1, M2, M3: independent transgenic lines of the indicated mutants; E: ectopic insertion lines; C1, C2: independent complemented strains of the indicated gene deletion mutants. Scale bars correspond to 20 μm **c** and 1 cm **e**

expression was induced at ~2.7 dpi, but not earlier[33]. These results implicate a role for the *fg3_54* product(s) in establishing the cell-to-cell invasion mode in wheat, which allows fungal hyphae to invade host internal cells that actively launch a defense response after fungal colonization of wounded cells at the host edge.

Hyphae of the *FG-Δfgm5-AmCyan* mutant alone were blocked around the inoculation site (Fig. 2e). However, when spores of the

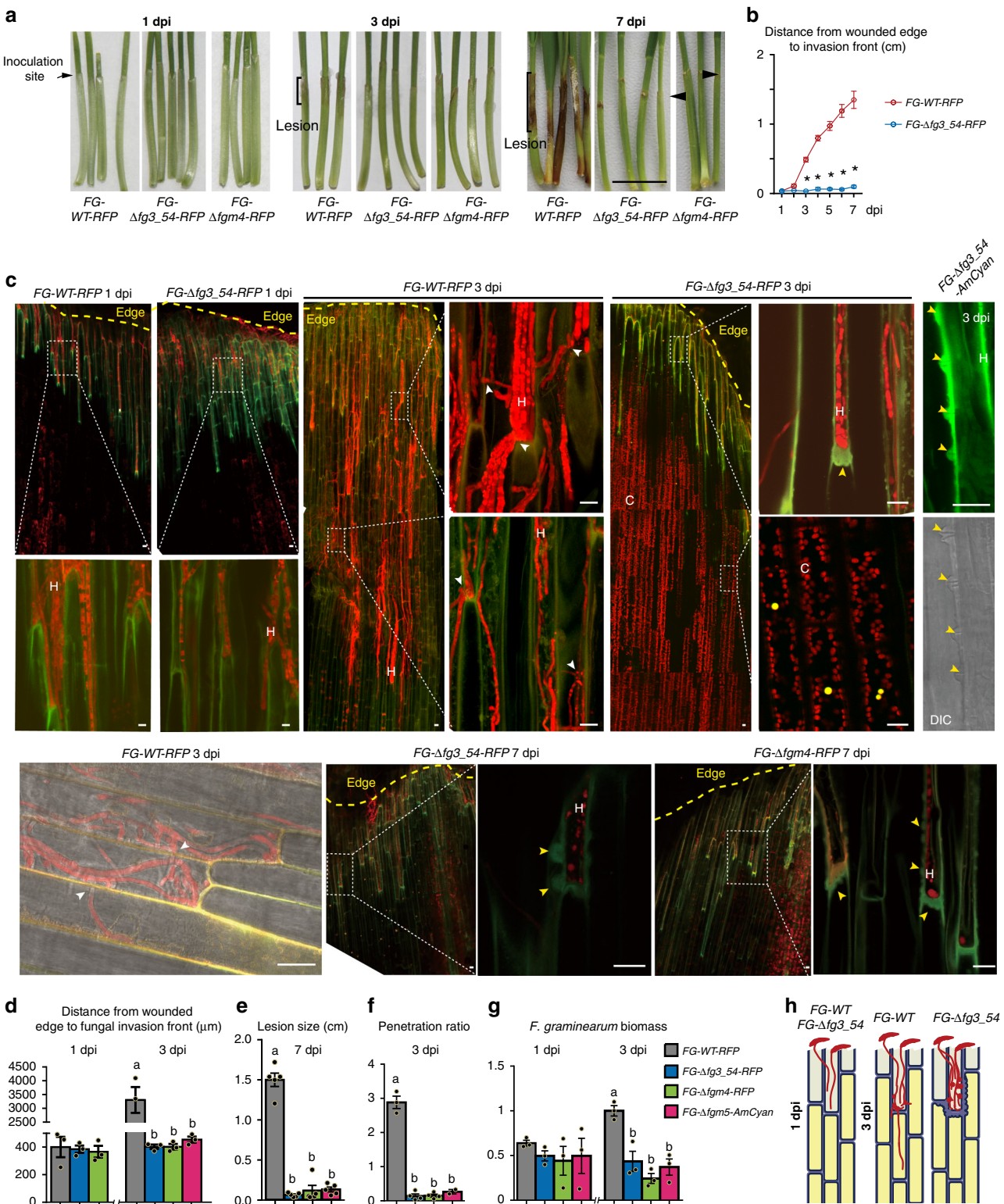

*FG-Δfgm5-AmCyan* mutant were mixed with the *FG-WT-RFP* strain and then inoculated on wheat coleoptiles, *FG-Δfgm5-AmCyan* and *FG-WT-RFP* hyphae grew similarly inside coleoptiles, and no typical papillae were observed (Supplementary Fig. 5e). This result suggests that products of *fg3_54* from wild-type strain may facilitate the spread of *FG-Δfg3_54* mutant inside host tissues, possibly via secreted and diffusible molecule(s).

**FGM4 turns on the expression of *fg3_54*-related genes**. The above findings demonstrated the involvement of a diffusible *fg3_54*-related product in the cell-to-cell penetration process in wheat infection and thus generated interest in the functions of genes within this sequence. *Fg3_54* contains *nrps5* and *nrps9*, both of which encode NRPSs (Fig. 1a). NRPSs are often giant proteins in which functional domains are organized into modules. Each module typically consists of an adenylation (A) domain that

**Fig. 2** *F. graminearum fg3_54* deletion mutants have defects in cell-to-cell invasion in wheat coleoptiles. **a** Representative images of wheat seedlings after inoculation with the indicated strains expressing fluorescent proteins. dpi: days post inoculation. Black arrowheads point to small lesions restricted to the wounded edge. Black scale bars represent 1 cm. **b** Chart showing the hyphal invasion distance in wheat coleoptiles from 1 to 7 dpi. Data are means ± s.e.m. Asterisks indicate a significant difference ($P < 0.001$) compared with *FG-WT-RFP* according to the two-tailed unpaired Student's *t* test. **c** Confocal images of the indicated strains of *F. graminearum* growing inside wheat coleoptiles at the indicated time points. White scale bars represent 20 μm. White arrowheads indicate hyphal passage of the plant cell wall; yellow arrowheads indicate papillae-like structures. C: red signals from the chloroplasts. H: *Fusarium* hyphae. **d–g** Chart showing the measurements of wheat coleoptile infection at the indicated time points. Data are means ± s.e.m. ($n = 3$). Different letters indicate a significant difference ($P < 0.05$) based on one-way ANOVA followed by Tukey's multiple comparison test **d**, $F = 37.48$, df $= 3$, $P < 0.0001$; **e**, $F = 159.6$, df $= 3$, $P < 0.0001$; **f**, $F = 186.1$ df $= 3$, $P < 0.0001$; **g**, $F = 16.55$, df $= 3$, $P = 0.0009$). Source data are provided in Supplementary Data 3. **h** Schematic diagram of *F. graminearum* invasion in wheat coleoptiles

is responsible for amino acid recognition and activation, a thiolation (T) domain for activated monomer channeling and a condensation (C) domain for transpeptidation to elongate the growing peptide chain as well as optional domains for specialized modifications (e.g., epimerization, E; and reduction, R) of aminoacyl substrates or peptidyl intermediates[41,42]. NRPS9 is a **M1**-($A_1$-$T_1$) di-domain protein that likely acts as a loading module for starter unit incorporation. NRPS5 contains seven such elongation modules, i.e., **M2**-($A_{2a}$-$C_2$-$A_{2b}$-$T_2$)-**M3**-($C_3$-$A_3$-$T_3$-$E_3$)-**M4**-($C_4$-$A_4$-$T_4$-$E_4$)-**M5**-($C_5$-$A_5$-$T_5$-$E_5$)-**M6**-($C_6$-$A_6$-$T_6$-$E_6$)-**M7**-($C_7$-$A_7$-$T_7$-$E_7$)-**M8**-($C_8$-$A_8$-$T_8$-R), and could coordinate with NRPS9 to constitute an assembly line and program the production of an octapeptide product.

However, despite numerous attempts, wild-type *F. graminearum* strain failed to produce any distinct peptide products when compared to the *FG-Δfg3_54* mutant under various in vitro cultivation conditions. Given that *nrps9* and *nrps5* were poorly expressed in vitro (Supplementary Fig. 6a), we hypothesized that both genes might be under stringent transcriptional control and would function specifically for octapeptide production *in planta*. In searching for regulator genes that can turn on its expression to facilitate product identification, we first tested *FgLaeA*, a known general regulator of secondary metabolism[43]. However, *fg3_54* cluster genes expression did not increase in strains constitutively expressing *FgLaeA* (Supplementary Fig. 6b).

We then explored the possible existence of a specific transcription regulator within the cluster. After reanalyzing the *fg3_54* sequence, we noticed *fgm4*, which encodes FGM4, a protein with overall low sequence homology to Aps2 from *Fusarium semitectum* and ToxE from *Cochliobolus carbonum* (Fig. 3a); however, it shares with these transcription factors a basic DNA-binding domain with variable ankyrin-like repeats, which is characteristic for bANK family proteins[44]. *FG-Δfgm4* mutants share a largely reduced invasiveness with *FG-Δfg3_54* and *FG-Δnrps5* during wheat infection, causing drastically smaller lesions (<0.2 cm long) on coleoptiles and fewer spikelets with blight symptoms (Fig. 1b, d, e). The deficiency of *FG-Δfgm4* in coleoptile infection can be complemented by in trans expression of *fgm4*. In the complemented strain *FG-Δfgm4::fgm4*, the lesion size was restored to that of the wild-type strain. Therefore, *fgm4* appears to be functionally associated with *fg3_54* structural genes and is necessary for *F. graminearum*-mediated wheat infection. We used a yeast two hybrid assay to show that FGM4 could activate transcription in yeast (Fig. 3b), and FGM4 fused to mRFP or enhanced green fluorescent protein (eGFP) displayed a subcellular localization that was partially overlapping with DAPI signals (for DNA), indicating its localization in nuclei (Fig. 3c). These results further supported that FGM4 may function as a cluster-specific transcriptional regulator of *fg3_54*.

To test whether FGM4 could activate the transcription of *nrps9* and *nrps5* and facilitate product characterization, we introduced *fgm4* driven by the constitutive promoter *pEF1*[37]. The resulting derivative *WT-OE::fgm4* was cultured in vitro, and then was

subjected to an RNA sequencing-coupled transcriptional analysis (Fig. 3d). Consequently, most of the biosynthetic gene clusters concerning secondary metabolites, including that known to encode the trichothecene mycotoxins deoxynivalenol and nivalenol[4,15,16], were not responsive to the overexpression of *fgm4* (Supplementary Fig. 6c). However, overexpression of *fgm4* selectively enhanced the transcription of the *fg3_54* cluster genes (including *nrps5* and *nrps9*), rather than the flanking genes (Supplementary Table 3); in particular, the large RNA product resulting from *nrps5*, the reads of which cover a ~35 kb region, include untranslated and coding sequences but lack the three intron sequences (Fig. 3d). Quantitative and semiquantitative PCR after reverse transcription verified that the *fg3_54* cluster genes showed increased expression when cultured in vitro (Supplementary Fig. 7). These findings support the notion that FGM4 acts as a pathway-specific transcriptional regulator capable of activating the expression of *nrps5* and *nrps9*.

**The product of NRPS5 and NRPS9 is an octapeptide**. We then scaled-up the *WT-OE::fgm4* culture in vitro. Remarkably, compared with the wild-type and *FG-Δfg3_54* mutant strains, *WT-OE::fgm4* accumulated a new product, fusaoctaxin A ($[M + H]^+$ *m/z*: calcd. 773.5131 for $C_{36}H_{68}N_8O_{10}$, found 773.5135) (Fig. 4a). According to spectral analysis by 1D and 2D NMR and subsequent tandem mass spectrometry (MS), this product is a linear octapeptide with the residues organized as [γ-amino butyl acid (GABA)]$_1$-[L-Ala]$_2$-[D-*allo*-Ile]$_3$-[D-Ser]$_4$-[D-Val]$_5$-[D-Ser]$_6$-[D-Leu]$_7$-[L-Leuol]$_8$, in which the stereochemistry of related residues was determined using a modified Marfey's method, combined NMR analysis and ultimately a synthesized authentic compound for comparison (Fig. 4b, Supplementary Data 1). Overall, the composition and residue organization of fusaoctaxin A are highly consistent with the catalytic logic of the assembly line composed of NRPS9 and NRPS5, which likely utilizes GABA as a starter unit and sequentially incorporates seven extender units composed of the residues L-Ala, L-*allo*-Ile, L-Ser, L-Val and L-Leu. During the process, each of the residues that are tethered on modules (**M3**-**M7**) containing an E domain can undergo an epimerization reaction to produce a D-configuration before the transpeptidation reaction occurs. The elongation of the peptidyl chain might be terminated by module **M8**-mediated L-Leu incorporation, followed by R domain-catalyzed 4 electron reduction to release the resulting octapeptide from the assembly line as an alcohol (Fig. 4c)[45].

Liquid chromatograph(LC)–MS analysis of individually deleting *nrps9* and *nrps5* in the *fgm4*-overexpressing derivative *WT-OE::fgm4* further verified that the biosynthesis of fusaoctaxin A involves the activities of NRPS9 and NRPS5 (Fig. 4a). The mutant strains *FG-Δnrps9::fgm4* and *FG-Δnrps5::fgm4* were incapable of producing this octapeptide; however, deleting *fgm1*, *fgm2*, *fgm3*, *fgm5*, or *fgm9* in the derivative *WT-OE::fgm4* did not affect the production of fusaoctaxin A, thereby excluding them from roles in the biosynthesis of fusaoctaxin A.

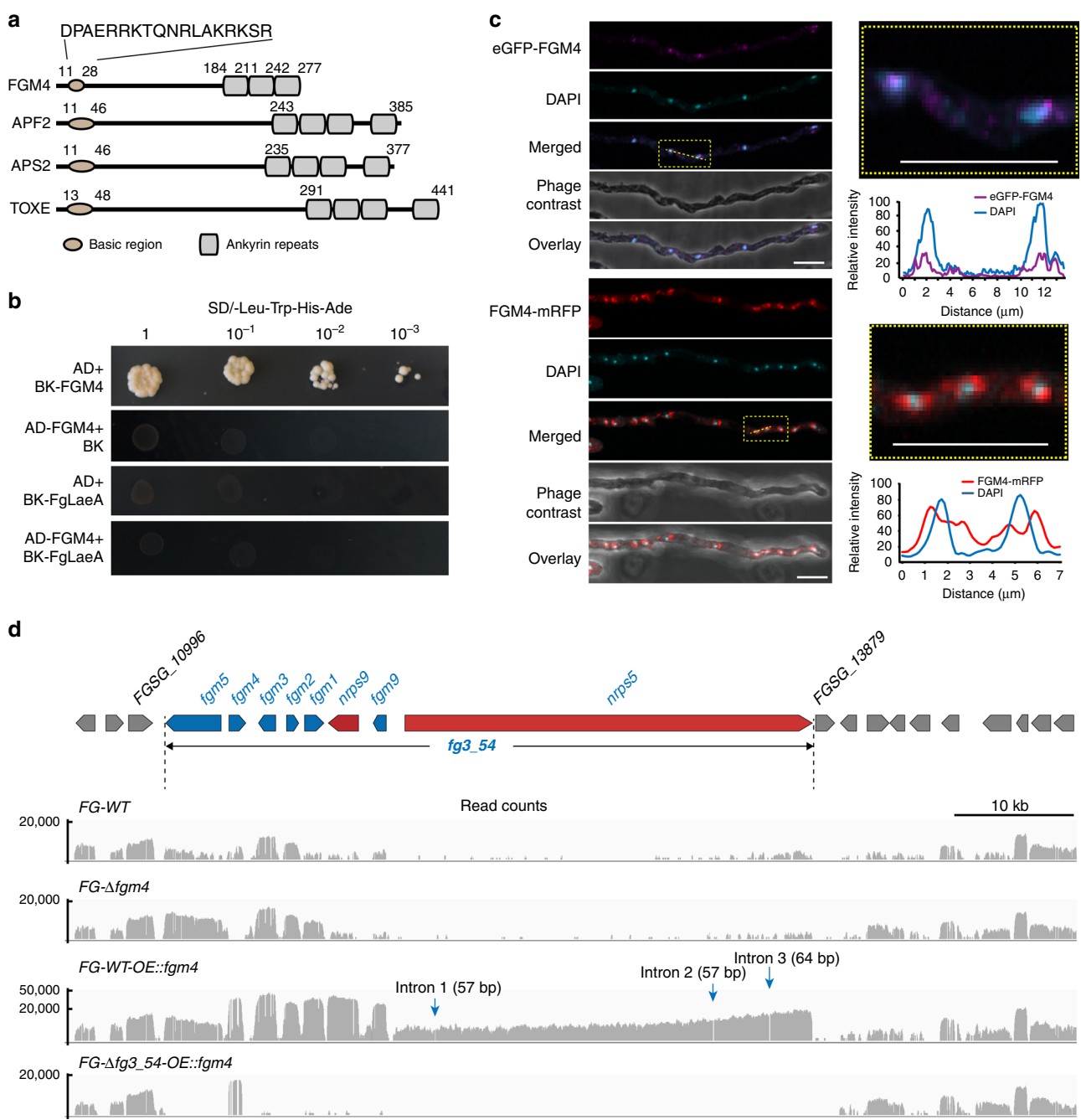

**Fig. 3** FGM4 is the key regulator inducing expression of the *fg3_54* cluster. **a** FGM4 has a similar domain structure to several basic-ankyrin repeats transcription factors. Three ankyrin repeat domains are predicted in the C terminal region. Numbers indicate amino acid residue positions. **b** Growth of yeast cells cotransformed with representative constructs. Transformants were spotted on SD/-Leu-Trp-His-Ade medium. **c** Subcellular localization of FGM4 as determined by confocal microscopy. eGFP and mRFP were fused to the N terminus and C terminus of FGM4, respectively. Relative signal intensities of the respective emission fluorescence along the lines drawn across the hyphae are charted on the right. Bars represent 10 μm. **d** Coverage plots of RNA-seq reads over the *fg3_54* cluster and flanking genes

After characterization of fusaoctaxin A in fungal culture, we set out to test whether it can be produced when *F. graminearum* infects plants. We conducted an analysis of *F. graminearum*-infected wheat tissues by LC–MS (Fig. 5a). Fusaoctaxin A was detected when the wild-type strain was used; in contrast, it was not detected when the *FG-Δfg3_54* mutant was used. Furthermore, the amount of fusaoctaxin A in infected coleoptiles was quantified by liquid chromatography tandem mass spectrometry (LC–MS/MS). Approximately 0.3 and 0.7 nmol per coleoptile endogenous fusaoctaxin A were detected in wild-type *F.*

*graminearum* infected wheat coleoptiles collected at 3 dpi and 7 dpi, respectively (Fig. 5b).

**Fusaoctaxin A restores cell-to-cell invasion ability in mutants.**
To determine whether fusaoctaxin A is responsible for invasion defects of *fg3_54* deletion mutants, this octapeptide, either in vitro purified or chemically synthesized, was added in varying concentrations onto *FG-Δfg3_54-RFP* mutant-inoculated wheat coleoptiles at 1, 2, and 3 dpi, when the genes within *fg3_54* are

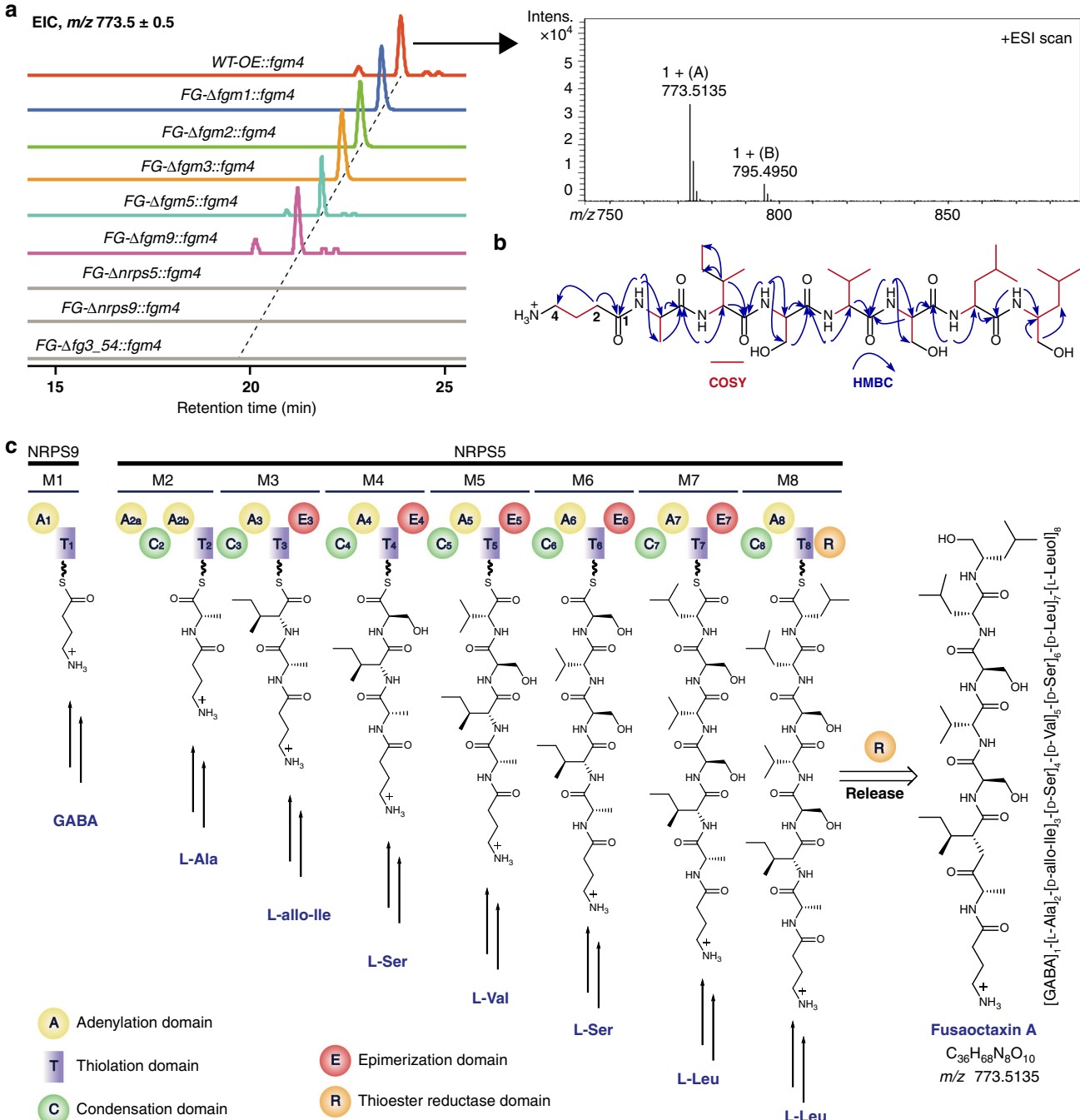

**Fig. 4** Identification of fusaoctaxin A as the main product of *fg3_54*. **a** Extracted ion chromatographic comparison of fusaoctaxin A production of the *fgm4*-overexpressing derivatives of the indicated strains of *F. graminearum*. The electrospray ionization spectra of fusaoctaxin A are indicated in the frame. For HPLC–MS analysis, the ESI *m/z* [M + H]$^+$ mode for the extracted ion was 773.5 ± 0.5. **b** The planar structure of fusaoctaxin A elucidated by NMR spectra ($^1$H, $^{13}$C, HSQC, $^1$H-$^1$H COSY, and HMBC). **c** Domain organization of non-ribosomal peptide synthetases and the biosynthetic route to fusaoctaxin A. NRPS5 and NRPS9, which constitute an assembly line programming the biosynthesis of fusaoctaxin A in a linear manner. The functional domains of NRPSs is indicated in color, including C (green), A (yellow), T (purple), E (red), and R (orange)

approximately expressed in the wild-type *F. graminearum*. Remarkably, the ability of *FG-Δfg3_54-RFP* to invade wheat was restored by fusaoctaxin A in a dose-dependent manner (above the threshold 1 nmol per seedling, which is similar to the levels of endogenous fusaoctaxin A). Larger lesions were formed, e.g., when 6 nmol of the octapeptide was used, *FG-Δfg3_54-RFP* produced lesions ~1.2 cm long at 7 dpi (Fig. 5c, d). Furthermore, the addition of fusaoctaxin A (above 20 nmol per spike) also increased the

number of symptomatic spikelets caused by *FG-Δfg3_54* (Fig. 5e, f), further supporting the notion that fusaoctaxin A also functions in wheat head blight infection. In contrast, the addition of a chemically synthesized non-relevant octapeptide, GIAVSTAG did not increase the coleoptile lesion size or symptomatic spikelet number caused by *FG-Δfg3_54* (Fig. 5c–f). Therefore, the disease enhancement in wheat by fusaoctaxin A was not due to non-specific effects of added carbon and nitrogen sources.

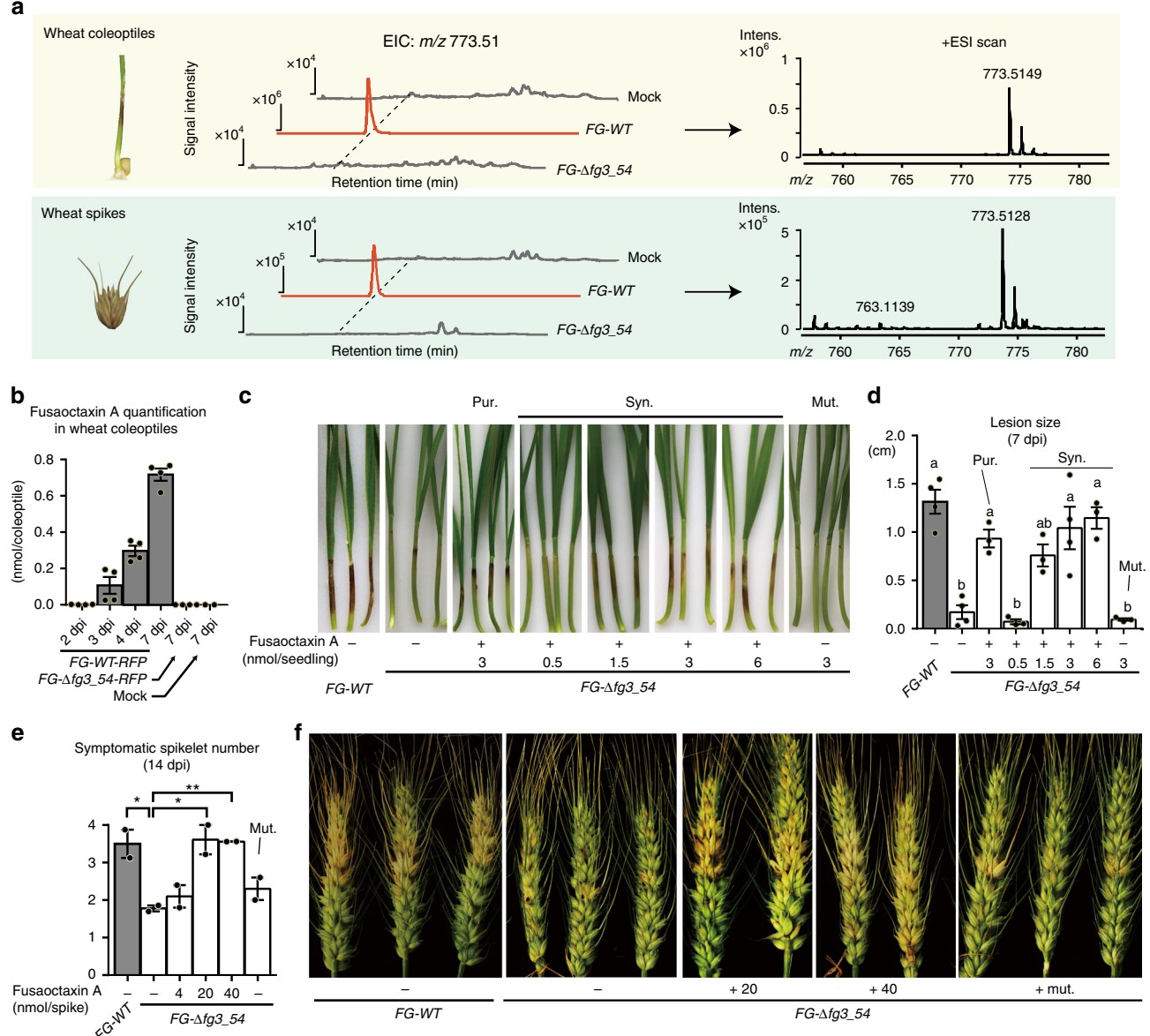

**Fig. 5** Detection of endogenous fusaoctaxin A in infected wheat tissues and infection complementation by exogenous fusaoctaxin A. **a** Extracts analysis of wheat coleoptiles and spikes infected by *F. graminearum* at 7 dpi using LC-ESI-MS. The extracted ion chromatography (EIC) of 773.51 (left) and mass spectrum (right) of the samples are shown. The experiments were repeated three times, and the results of one representative experiment are shown. **b** Quantification of endogenous fusaoctaxin A in infected wheat coleoptiles using LC–MS/MS. **c**, **d** Representative images and measurements of lesions on wheat seedling coleoptiles infected by indicated *F. graminearum* strains at 7 dpi. pur.: purified fusaoctaxin A from fungal extracts; syn.: Artificial synthesized fusaoctaxin A; mut: chemically synthesized control peptide GIAVSTAG. Different letters indicate a significant difference (*P* < 0.05) based on one-way ANOVA followed by Tukey's multiple comparison test (*F* = 16.17 df = 7, *P* < 0.0001). **e**, **f** The number of symptomatic spikelets and symptoms of wheat spikes infected by indicated *F. graminearum* strains at 14 dpi. Data are means ± s.e.m. Asterisks indicate a significant difference (*P* < 0.05) compared with *FG-Δfg3_54* according to the two-tailed unpaired Student's *t*-test (*P* < 0.05; **P* < 0.01). Source data are provided in Supplementary Data 3

Confocal microscopy showed that in the presence of 3 nmol fusaoctaxin A per seedling, *FG-Δfg3_54-RFP* hyphal spread inside of coleoptiles (Fig. 6a) and penetrated wheat cell walls with a penetration ratio of ~3 at 3 dpi (Fig. 6b, c). Under the same condition, autofluorescence and callose staining from the walls of plant cells neighboring those containing mutant hyphae were significantly reduced, as seen with the wild-type strain (Fig. 6a, b and Supplementary Fig. 8). Consequently, fusaoctaxin A is capable of complementing the deficiency of the mutant *FG-Δfg3_54* during cell-to-cell invasion and suppressing host defense (e.g., cell wall callose deposition) in wheat infection. Notably, the

addition of fusaoctaxin A even enhanced wheat coleoptile infection with the wild-type *F. graminearum* (Fig. 6d).

**Fusaoctaxin A confers wheat invasion ability in *Fusarium* species.** Comparative genomic analysis has shown that *fg3_54* homologs are present in the genomes of several *Fusarium* species[46] (Fig. 7a, b), including *F. culmorum* and *F. pseudograminearum*, which are closely related to *F. graminearum* in phylogeny and capable of causing Fusarium crown rot on wheat and barley, and *F. avenaceum*[47] and *F. oxysporum* FOSC 3-a,

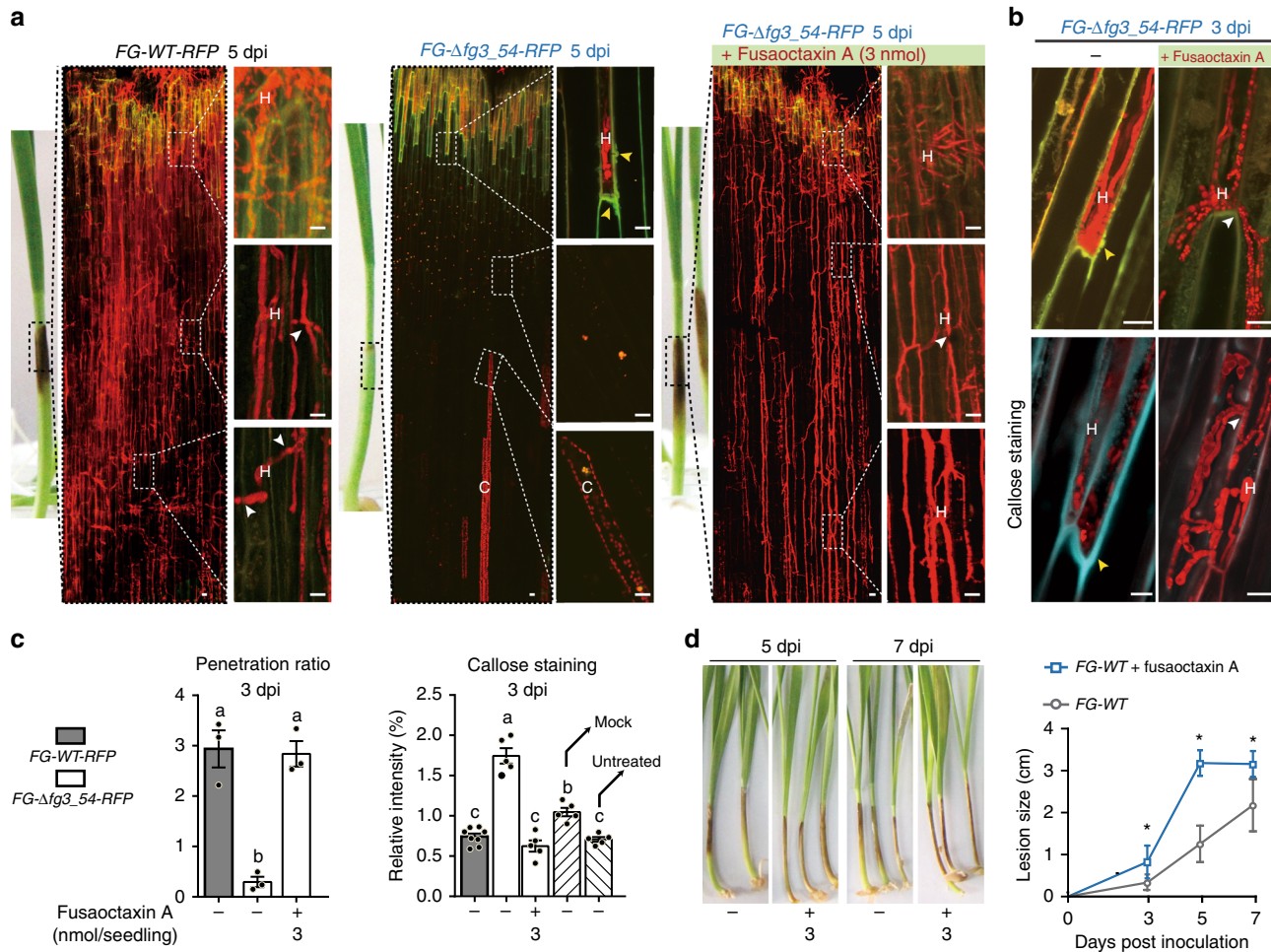

**Fig. 6** Fusaoctaxin A can restore cell-to-cell invasion in mutants and enhance wild-type virulence in wheat. **a** Fluorescent microscopic images of wheat coleoptiles infected by the indicated strains. Images are overlay of RFP (fungal RFP and wheat chlorophyll autofluorescence) and GFP (plant cell wall autofluorescence) channels. **b** Confocal microscopic images of wheat coleoptile cells infected by indicated strains. The lower panel shows aniline blue staining (indicating the plant cell wall component callose) of these cells. H: *F. graminearum* hyphae. C: chlorophyll. The white arrowheads indicate hyphal penetration of plant cell wall. Scale bar = 20 μm. **c** Measurements of the penetration ratio and callose staining intensities of wheat coleoptile infection. Data are means ± s.e.m. Different letters indicate a significant difference ($P < 0.05$) based on one-way ANOVA followed by Tukey's multiple comparison test. ($F = 32.13$ df = 2, $P = 0.0006$; $F = 58.7$ df = 4, $P < 0.0001$). **d** Measurements and representative images of lesions on wheat seedling coleoptiles infected by wild-type *F. graminearum* with or without fusaoctaxin A. Data are mean ± s.d. Asterisks indicate a significant difference compared with *FG-WT* according to the two-tailed unpaired Student's *t*-test (*$P < 0.05$; **$P < 0.01$). Source data are provided in Supplementary Data 3

which are more distantly related and were isolated from barley and human patients suffering *Fusarium* keratitis[48], respectively. Most likely, these *Fusarium* species have the potential to produce fusaoctaxin A. *F. avenaceum*, the species previously unknown as a wheat invader, was selected to infect wheat seedling and indeed caused dark brown lesions on coleoptiles (Fig. 7c). In contrast, fungal pathogens with genomes lacking a *fg3_54* homolog, including *F. oxysporum* f. sp. *cubense* tropical race 4 (*Foc* TR4, a soil-borne fungus causing the destructive disease *Fusarium* wilt on banana)[49] and *F. poae*[50] and the non-*Fusarium* species *Verticillium dahliae* Kleb[51] and *Metarhizium robertsii*[45], did not (Fig. 7c). These results suggested that the fungal ability in wheat coleoptile infection correlates to the fusaoctaxin A production ability.

We then examined whether fusaoctaxin A can confer wheat infection ability to non-wheat pathogens. Remarkably, in the presence of fusaoctaxin A, *Foc* TR4 became pathogenic and caused lesions on coleoptiles during wheat seedling infection, with a length of ~0.6 cm when 3 nmol of the octapeptide was

added per seedling (Fig. 7d). Intriguingly, this octapeptide also enhanced *F. poae* infection of wheat coleoptiles and caused a lesion size similar to wild-type *F. graminearum*, but it could not stimulate the non-*Fusarium* species *M. robertsii* to behave in a similar invasion manner (Fig. 7d). Observations at the cellular level showed that *Foc* TR4 hyphae could grow at the wounded edge of wheat coleoptiles but could not invade further into the lower parts. With the addition of exogenous fusaoctaxin A, *Foc* TR4 hyphae spread from the wounded edge into the lower parts of coleoptiles through cell-to-cell penetration (Fig. 7e). These results demonstrate that fusaoctaxin A, the biogenesis of which relies on a *fg3_54*-like sequence, functions as a genus-specific virulence factor that is necessary for *Fusarium*-mediated wheat infection.

**Fusaoctaxin A has selective effects on host cells**. We then examined how fusaoctaxin A specifically enables wheat infection and whether it manipulates host cells to achieve susceptibility

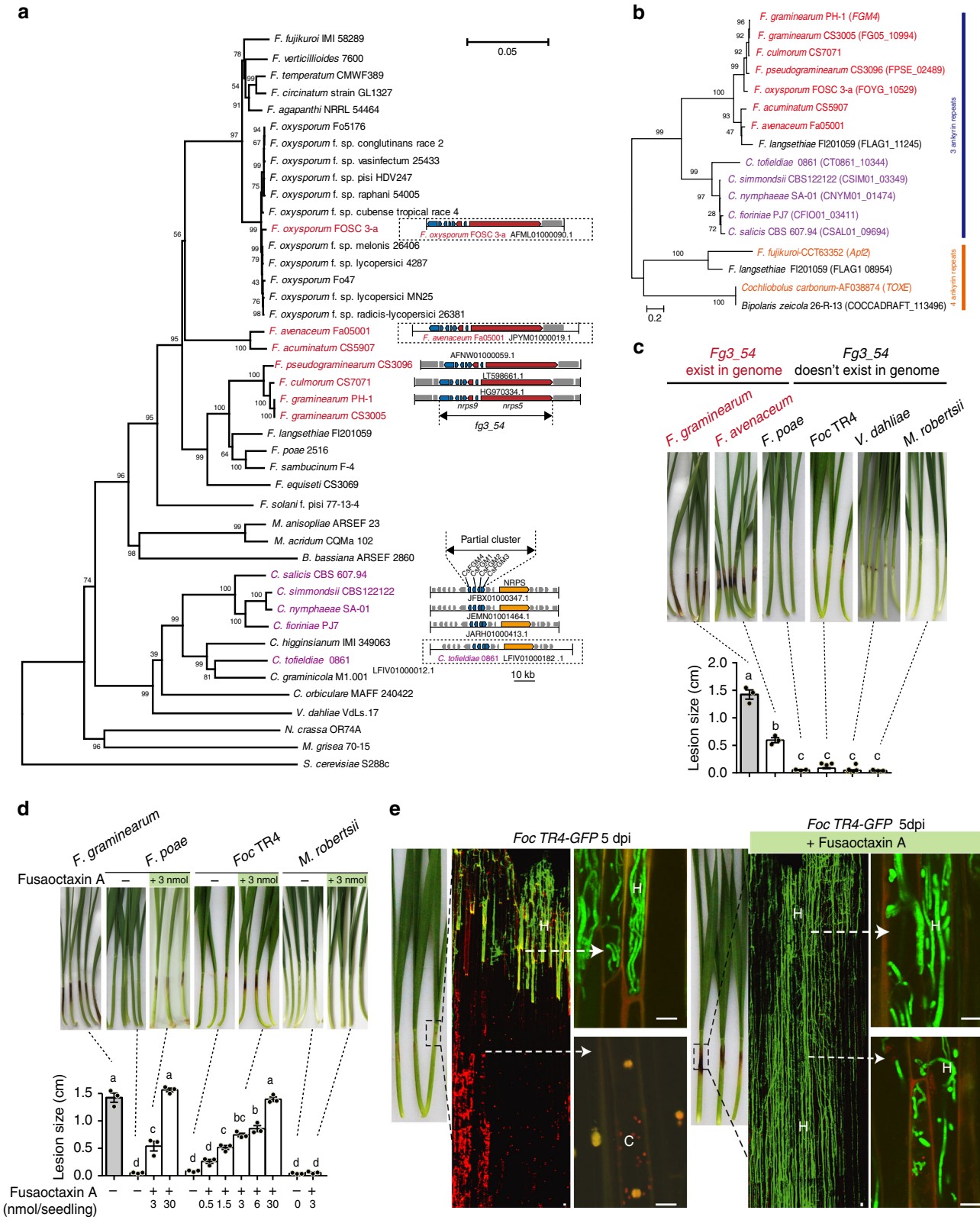

using chemically synthesized pure compound. Unlike other mycotoxins such as victorin, which directly induces disease symptom (even when the fungal pathogen *Cochliobolus victoriae* is absent)[52], fusaoctaxin A (using 3 nmol per seedling) caused no visible lesions on wheat coleoptiles up to 7 dpi, except for slight bleaching (Fig. 8a). During a 5-h treating period, this octapeptide caused no significant changes in either wheat cell viability (assessed by acridine orange and ethidium bromide staining, Fig. 8b) or cell membrane integrity (assessed by the ability to plasmolyze upon salt treatment, Fig. 8c), but caused a subtle change in the subcellular localization pattern of chloroplasts in coleoptile cells (Fig. 8d). In untreated coleoptiles, chloroplasts

**Fig. 7** Other fungal genomes include *fg3_54* homologs, and fusaoctaxin A enables invasion of wheat by non-pathogenic *Fusarium* species. **a** Phylogenetic tree of 43 fungal species showing the organization of the *fg3_54* cluster in the genomes if present. The neighbor-joining tree was built based on the DNA sequences of the DNA-directed RNA polymerase II subunit (RPB2). Species containing syntenic *fg3_54* cluster (all 8 genes) are shown in red. The scale bar indicates the mean number of nucleotide substitutions per site. **b** Phylogenetic tree obtained by neighbor-joining analysis of FGM4 homologous proteins. The numbers at the internal nodes indicate the percentage of replicate trees in which the associated taxa clustered together in a bootstrap test of 2000 replicates. **c**, **d** Representative images and measurements of lesions on wheat seedling coleoptiles inoculated with indicated fungal strains with or without supplement of fusaoctaxin A. Lesions were imaged and measured at 7 dpi. Data are means ± s.e.m. Different letters indicate significantly different levels ($P < 0.05$) based on one-way ANOVA followed by Tukey's multiple comparison test (**c**: $F = 246.3$, $df = 5$, $P < 0.0001$; **d**: $F = 157.6$, $df = 11$, $P < 0.0001$). **e** Symptom analysis of wheat coleoptiles infected by *Foc TR4-GFP* at 5 dpi, with or without supplement of fusaoctaxin A. Images are overlays of the RFP (chlorophyll autofluorescence) and GFP (fungal GFP and plant cell wall autofluorescence) channels. Source data are provided in Supplementary Data 3

localize close to plasma membrane, presumably squeezed by a central vacuole, while in fusaoctaxin A-treated cells, chloroplasts are more randomly scattered.

In searching for even earlier cellular responses, we identified an effect on plasmodesmal regulation after a 3-h treatment using the Drop-ANd-See assay. The non-fluorescent membrane-permeable agent 5(6)-carboxyfluorescein diacetate (CFDA) can be cleaved by intracellular esterases, generating the fluorescent CF, which is membrane-impermeable and can only spread among live cells through plasmodesmata[53]. After loaded on mock-inoculated coleoptiles, CF spread deeply (~2 mm from the edge, Fig. 8e), irrespective of the presence of fusaoctaxin A. However, the application of fusaoctaxin A on uninfected coleoptiles may not reflect its effect on fungal induced responses in infected coleoptiles. Moreover, if performing the assay on mutant or non-pathogenic *Fusarium* strains with invasiveness that can be restored by fusaoctaxin A, the addition of fusaoctaxin A will introduce additional interference from the resulting fungal growth. We have shown that the insect fungal pathogen *M. robertsii* cannot invade internal cells even with the addition of fusaoctaxin A (Fig. 7d), which can avoid other interferences in plasmodesmal permeability due to hyphal growth. Therefore, we used *M. robertsii* to inoculate wheat coleoptile to evaluate the effects of fusaoctaxin A *per se*. In *M. robertsii*-inoculated coleoptiles, CF fluorescence was restricted to a small region close to the edge (<0.2 mm), indicating plasmodesmata closure in wheat tissues challenged by the fungus (Fig. 8e). Strikingly, in *M. robertsii*-inoculated coleoptiles treated with fusaoctaxin A, the CF signals spread ~2.5 mm from the inoculated edge (Fig. 8e), indicating that fusaoctaxin A was able to override the *M. robertsii*-caused restriction of CF spreading in wheat coleoptiles. We infer that manipulation of plasmodesmal permeability is an action of this octapeptide.

In addition to assessing fusaoctaxin A effects on wheat, we also wondered whether it caused conserved cellular responses among cell types and species. We found that fusaoctaxin A (at a 50 μM concentration) showed no effects on in vitro germination or the growth of pollen tubes, a model of fast-growing plant cells (Supplementary Fig. 9a), but it caused chlorosis and water-soaking within 1 h when infiltrated into leaves of *Nicotiana benthamiana* (Supplementary Fig. 9b). Also, fusaoctaxin A showed no cytocidal effects on mammalian cell lines (Supplementary Fig. 9c). These results indicate that fusaoctaxin A has selective effects on host cells.

**Wheat gene expression changes in response to fusaoctaxin A.** To elucidate the molecular basis underlying these cellular responses, we further diagnosed the wheat global gene expressional responses to fusaoctaxin A by transcriptomic analysis. Eight samples of wheat coleoptile tissues (three biological replicates each sample) were subjected to RNA sequencing (Fig. 9a).

The first group containing three samples of wheat coleoptiles collected at 72 h after mock-, wild-type *F. graminearum*- or *FG-Δfg3_54*-inoculation was designed to compare responses to fungal infection with or without fusaoctaxin A production. We identified 13,215 downregulated and 9469 upregulated genes in coleoptiles infected by wild-type *F. graminearum*, in comparison to the mock-inoculated samples. By contrast, only 2746 genes were downregulated and 2124 genes were upregulated in *FG-Δfg3_54* mutant-inoculated coleoptiles. 11,458 of the 13,215 down-regulated and 7285 of the 9469 upregulated genes in wild-type infected coleoptiles were not down- and upregulated in *FG-Δfg3_54* infected coleoptiles, respectively. Therefore, these 11,458 and 7285 genes represent the fungal-produced fusaoctaxin A suppressed and induced genes during infection, respectively.

However, the expressional difference between wild-type and mutant strains infected samples was caused not only by whether or not fusaoctaxin A was produced but also by the greater fungal biomass and more advanced infection in wild-type strains infected coleoptiles than in mutant infected coleoptiles. To better identify responses to fusaoctaxin A, we designed the second RNA sequencing group containing five samples consisting of wheat coleoptiles that were untreated, 3 h after fusaoctaxin A- or mock-treated, and 24 h after fusaoctaxin A- or mock-treated (Fig. 9a). Regarding timing, fungal *fg3_54* cluster gene expression was not detected during coleoptile infection up to 40 hpi, and its high expression was detected at 64 hpi. The differential gene expression caused by fungus-producing fusaoctaxin A or not detected in coleoptiles at 72 hpi roughly corresponded to 8–24 h post fusaoctaxin A treatment. Principal component analysis showed that transcriptomes of fusaoctaxin A-treated coleoptiles were more similar to untreated ones than to the transcriptomes of infected coleoptiles (Fig. 9b).

In comparison to the untreated sample, we identified 2232 downregulated and 2170 upregulated genes in coleoptiles treated with fusaoctaxin A for 3 h. Furthermore, 1714 of the 2232 downregulated and 1509 of the 2170 upregulated genes in fusaoctaxin A-treated coleoptiles were not down- and upregulated in the mock-treated samples, respectively. After treatment with fusaoctaxin A for 24 h, we found that 2274 of the 2882 downregulated and 2140 of the 2228 upregulated genes in fusaoctaxin A-treated coleoptiles were not down- and upregulated in the mock-treated samples, respectively (Fig. 9c).

Given that exogenous fusaoctaxin A may also have effects that are not identical to their native function during infection, we considered the common gene set between the fungal-produced fusaoctaxin A suppressed genes (11,458) and the exogenous fusaoctaxin A suppressed ones (1714 genes after 3 h of treatment and 2274 genes after 24 h of treatment) as the best candidates for fusaoctaxin A suppressed (Downregulated) Genes (FADG3 and FADG24, respectively) (Fig. 9c). The FADG3 group contained 868 genes. Gene ontology (GO) analysis showed a significant overrepresentation (*q*-value <0.05, gene count >3) of 41

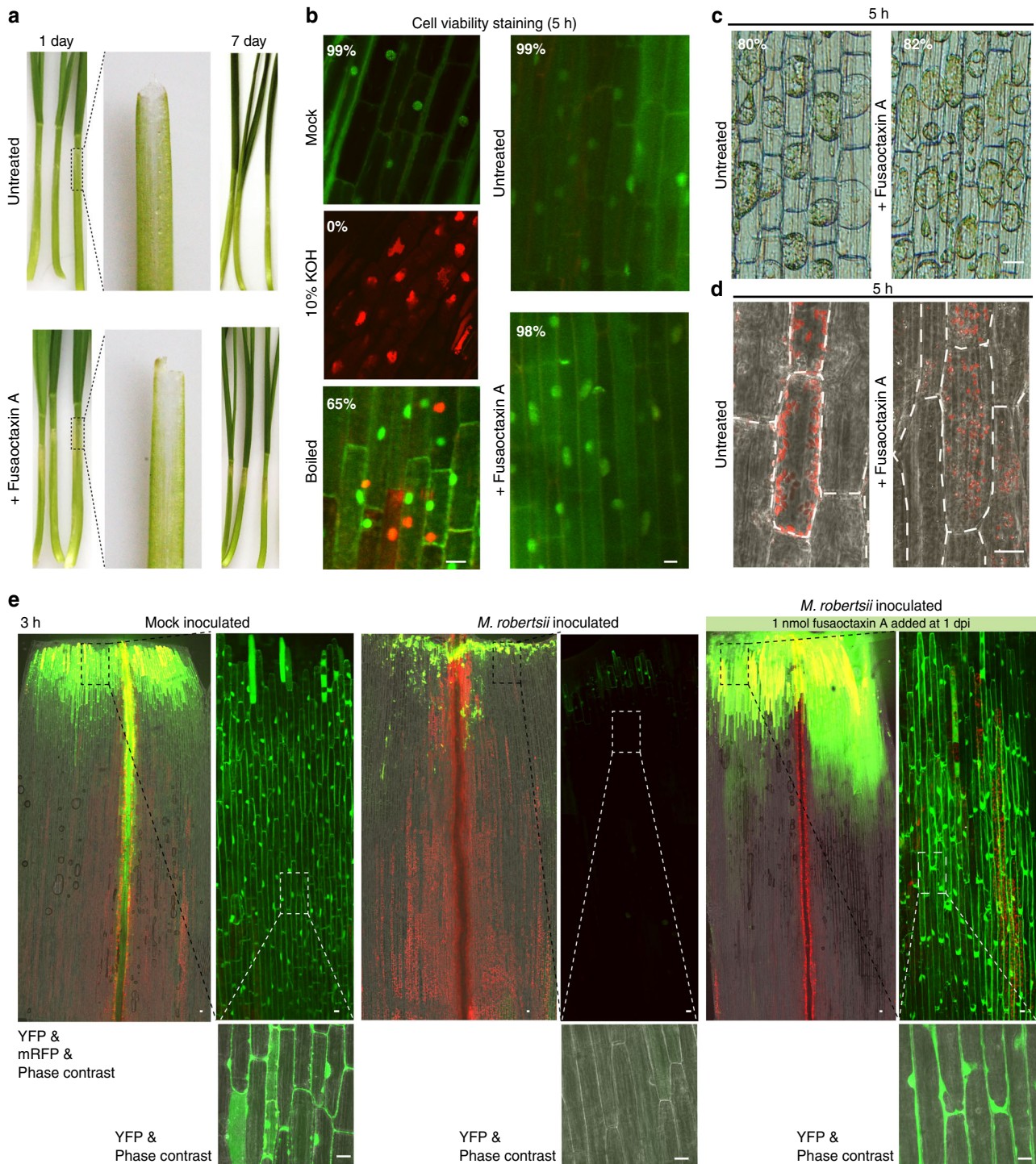

**Fig. 8** Fusaoctaxin A changes chloroplast subcellular localization and terminates the plasmodesmal closure response. **a** Wheat seedlings with or without the octapeptide applied at the top edge of the coleoptile. The middle panel shows the coleoptiles that were gently detached from the seedling immediately before imaging. **b** Viability of wheat coleoptile cells with various treatment assayed by Acridine orange/ethidium bromide (AO/EB) staining. Boiled: the coleoptile treated by 95 °C water for 5 min. The average percentage of live cell (green signals from AO staining in nuclei) numbers divided by the sum of live and dead cells (orange signals from EB staining in nuclei) in each sample is labeled in the image. **c** Representative micrographs comparing the cell plasmolysis ability in wheat coleoptile with or without octapeptide treatment for 5 h. The average percentage of cells that exhibited plasmolysis in each sample is labeled in the image. **d** Representative confocal micrographs comparing wheat coleoptile cells with or without the octapeptide treatment for 5 h. A single plane of RFP confocal images is overlaid with the phase contrast light field. Red signals are autofluorescence from chlorophyll. Broken white lines mark the margins of individual cells. **e** Drop-ANd-See assay of wheat coleoptiles showing the plasmodesmal permeability differences after fusaoctaxin A treatment for 3 h. Three-day-old wheat seedlings were mock-inoculated (left) or inoculated with *Metarhizium robertsii*. White scale bar = 20 μm

chloroplastic activity related ('photosynthesis', 'chlorophyll bio-synthetic process', etc.) and 7 defense/immunity related categories ('innate immune response', 'defense response by callose deposition', etc.) in the FADG3 group. The FADG24 group contained 862 genes, among which 29 chloroplastic activity related GO terms ('photosynthesis', 'protein-chromophore linkage', etc.), 8 cytoskeleton-related GO terms ('structural constituent of cytoskeleton', 'microtubule', etc.) and 8 cell wall biosynthesis or modification-related GO terms ('cell wall biogenesis', 'cell wall pectin metabolic process', etc.) were significantly overrepresented (Fig. 9d).

It is noteworthy that FADG3 contains very few pathogenesis-related (PR) genes, which are known to be important in Fusarium head blight resistance[54]. A total of 21 PR genes detected by RNA sequencing encode 6 pathogenesis-related protein 1.1 (PR1), 3 β-1-3-glucanases (PR2), 4 chitinases (PR3), 4 vacuolar defense proteins (PR4), 2 thaumatin-like proteins (PR5), and 2 non-specific lipid transfer proteins (PR14) (Supplementary Fig. 10a). Except for a *PR14* belonging to FADG24, most PR genes were not suppressed or increased by fusaoctaxin A. Instead, many of FADG3 and FADG24 defense-related genes encode chloroplastic localized proteins (such as chloroplastic F-type $H^+$-transporting ATPases, chloroplastic NAD(P)H-quinone oxidoreductase subunits and chloroplastic peptidyl-prolyl cis-trans isomerases). Furthermore, FADG3 contains many genes encoding photosynthesis essential components, including the electron transport components PetC and PetF, suggesting that fusaoctaxin A suppresses the overall photosynthesis function[55] (Supplementary Data 2). This phenomenon may help to explain the early subcellular-redistribution (Fig. 8d) and later disappearance of chloroplasts (Fig. 8a) in fusaoctaxin A-treated and *F. graminearum* infected coleoptiles. These findings are also in line with reports that chloroplasts play a pivotal role in mediating early immune responses[56]. In addition, three chloroplastic calcium sensing receptor[57] (CAS)-like genes showed expression that was suppressed by fungal-produced or exogenous fusaoctaxin A (Supplementary Fig. 10). It has been reported that CAS is a thylakoid membrane-associated $Ca^{2+}$-binding protein that plays an important role in pathogen-associate molecular pattern-triggered immunity[57]. Suppression of *CAS*-like gene expression and overall chloroplast genes might contribute to fusaoctaxin A-induced susceptibility in wheat.

Interestingly, three plasmodesmata callose-binding protein (PDCB)-like genes showed expression that was suppressed by fungal-produced or exogenous fusaoctaxin A (Supplementary Fig. 10). Real-time PCR verified the expression of one PCDB-like gene (Supplementary Fig. 10d). PDCBs are X8 domain–containing proteins that are localized to the neck region of plasmodesmata and positively regulate callose accumulation to lead to plasmodesmal closure[58]. Among the 33 putative callose synthase (i.e. 1,3-beta-glucan synthase) genes, 7 genes were suppressed during wild-type *F. graminearum* infection, but not during cluster deletion mutant infection (Supplementary Fig. 10e). These findings are in line with the idea that fusaoctaxin A may interfere with plasmodesmata callose accumulation to prevent plasmodesmata closure upon fungal infection.

The common gene set between the fungal-produced fusaoctaxin A-induced genes (7285) and the exogenous fusaoctaxin A-induced genes (1509 genes after 3 h of treatment and 2140 genes after 24 h of treatment) are considered to be the best candidates for Fusaoctaxin A-induced (Upregulated) Genes (FAUG3 and FAUG24, respectively) (Fig. 9c). Strikingly, none of the GO categories related to chloroplastic activities or defense were overrepresented in FAUG3 or FAUG24, but 11 and 6 GO terms related to transmembrane transportation were significantly overrepresented in FAUG3 and FAUG24, respectively (Fig. 9d).

These observations are in line with the idea that *F. graminearum* produces fusaoctaxin A to manipulate host nutrients transportation or to interfere with host ion-mediated cell defense signaling.

**Discussion**

We characterized fusaoctaxin A, an unusual non-ribosomal octapeptide that is selectively produced by *F. graminearum in planta*, which serves as a new virulence factor during wheat infection. This virulence factor is genus specific, and it facilitates the invasion of *Fusarium* species through a cell-to-cell penetration process, during which host defense responses were largely inhibited, e.g., papillae-like cell wall depositions and plasmodesmata closure were largely inhibited (Fig. 10). Fusaoctaxin A functions in a manner differently from previously characterized virulence factors of *F. graminearum*, the secreted lipase FGL1[17], deoxynivalenol and nivalenol[59]. Our findings represent a key step to understand small molecule-mediated plant-pathogen interactions during *Fusarium* invasion.

In the field, plants are frequently wounded by pests, hail, or mechanical injury. Many fungi, even non-wheat pathogens, such as *Foc* TR4 and *M. robertsii* can grow within the wounded region (Fig. 7), but not further invade into interior tissues. The capability for cell-to-cell invasion into interior tissues, which can be conferred by fusaoctaxin A, is an important facet of the pathogenicity and host adaptation. Recently, the importance of cell-to-cell invasion through plasmodesmata, in addition to penetration of the first plant cell by the appressorium, has also been highlighted in rice blast fungus infection[60]. *F. graminearum* hyphae may also cross plant cell walls through pit fields (i.e., plasmodesma-enriched areas) during wheat spike infection[19].

The induction of PR gene expression, salicylic acid production, papillae formation and plasmodesmal closure are common defense responses of plant[39,61]. Our cellular observation and RNA sequencing results indicated that fusaoctaxin A does not induce most PR gene expression or salicylic acid production genes expression.

The *F. graminearum* genome contains 19 NRPSs, among which NRPS5 is the only one that terminates by R to produce a linear peptide and contains the largest number of A domains. NRPS A domains select monomers for incorporation, and the amino acid residues occupying the ten key positions that are relevant for substrate specificity within the A domains are referred to as the non-ribosomal code or Stachelhaus code[62]. Previous knowledge on codes for fungal NRPSs is limited. The product identification of NRPS5 and NRPS9 increases our knowledge of fungal non-ribosomal codes for amino acid selection (Supplementary Fig. 11).

Since the discovery of penicillin, NRPSs have been widely studied as important enzymes that are capable of synthesizing peptidic natural products. Non-ribosomal peptides (NRPs) can be grouped according to their biological activities[63]. The antibiotic vancomycin is a linear heptapeptide that prevents biosynthesis of the bacterial cell wall[64]. In plant-pathogen interaction, the amphipathic syringomycin from *Pseudomonas syringae* can insert into the plant plasma membrane and form transmembrane pores, permitting ions to flow freely across the membrane[65]. The *Magnaporthe oryzae* avirulence gene *ACE1* belongs to a gene cluster that is specifically expressed during infection. It encodes a hybrid between a polyketide synthetase and a non-ribosomal peptide synthetase[66], the product of which might be a tyrosine-derived cytochalasan compound[67]. HC-toxin is a cyclic tetra-peptide that functions as a virulence factor for the fungus *Cochliobolus carbonum* on its host, maize[68]. To the best of our knowledge, fusaoctaxin A is the only linear non-ribosomal peptide that has been identified as virulence factor in phytopathogens.

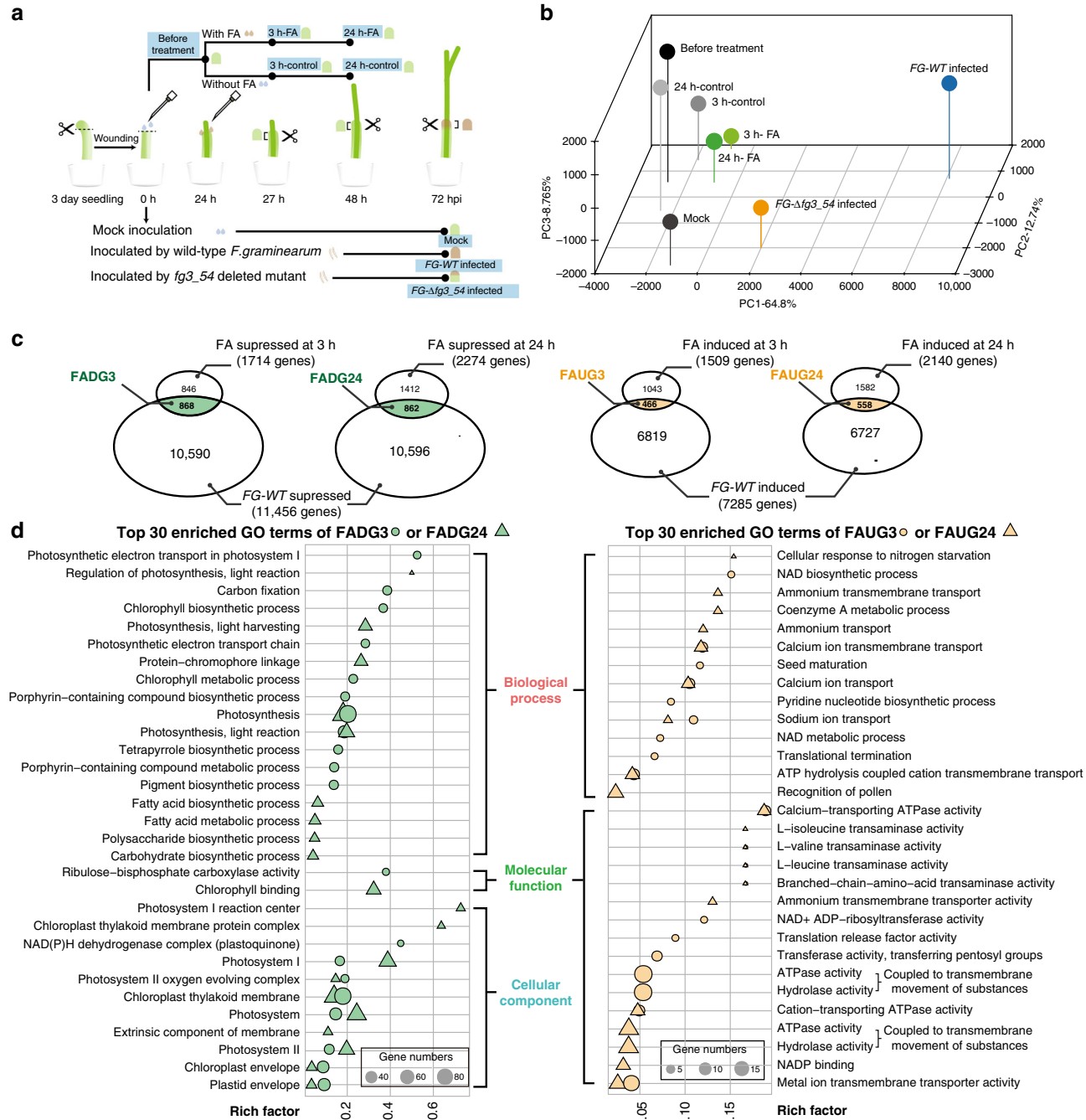

**Fig. 9** RNA sequencing analysis of wheat gene responses to fusaoctaxin A. **a** Schematic diagram of RNA-seq sample arrangements. The sample names are highlighted in the blue background. 3 h-FA and 24 h-FA: wheat coleoptiles collected at 3 h and 24 h, respectively, after treatment with 1 nmol fusaoctaxin A in 0.5% Tween-20 water solution (2 μL per seedling). 3 h-control and 24 h-control: wheat coleoptiles collected at 3 h and 24 h, respectively, after treatment with 0.5% Tween-20 water solution (2 μL per seedling). **b** Principal component analysis of the wheat transcriptomes of RNA-sequenced samples. **c** Venn diagram illustration of the best candidates for fusaoctaxin A suppressed or induced genes. FADG3/24: best candidates for Fusaoctaxin A suppressed (Downregulated) Genes at 3/24 h after treatment. FAUG3/24: the best candidates for Fusaoctaxin A-induced (Upregulated) Genes at 3/24 h after treatment. **d** The top 30 gene ontology (GO) categories enriched in fusaoctaxin A suppressed (left) or induced (right) genes. GO categories within biological process, cellular component, and molecular function are separately grouped and arranged based on their rich factor ranks. The size of the circle or triangle represents the comprising gene number

There also is no sequence similarity between fusaoctaxin A and known ribosomal peptides with biological activities, such as Cell-Penetrating Peptides, which can pass through cell membranes with no interactions with specific receptors[69], pepstatin, which is an Actinomyces-produced hexa-peptide containing the unusual amino acid statine and inhibits aspartyl proteases[70], and plant peptide hormones (CLE, TDIF, RALF etc.)[71]. It has been reported that *F. oxysporum* uses a functional homologue of the plant regulatory peptide RALF (rapid alkalinization factor) to cause disease in plants[72]. The estimated endogenous working

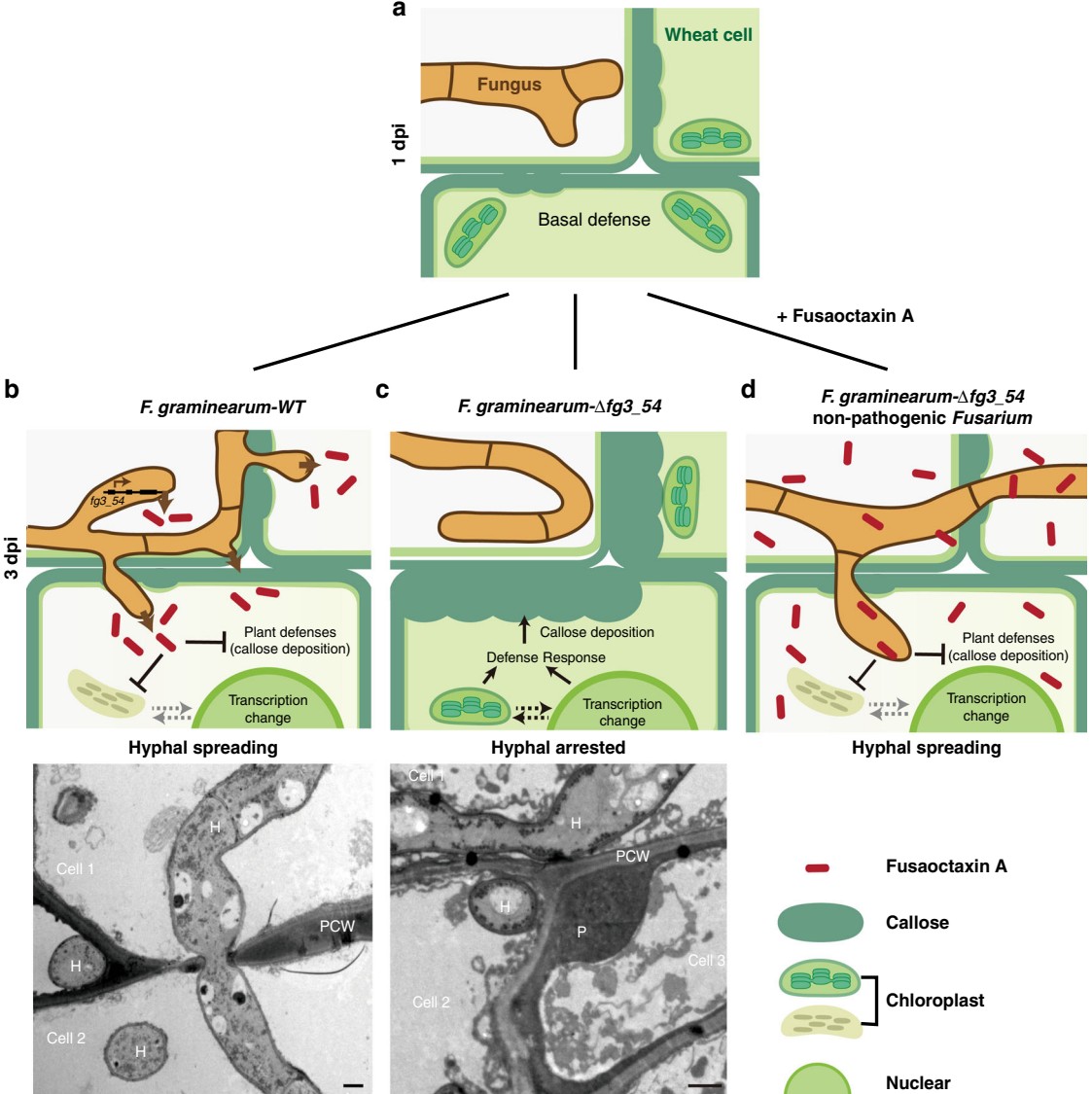

**Fig. 10** Schematic diagram of the proposed action of fusaoctaxin A during wheat infection. When landing on wounded wheat tissues, most fungal spores can germinate and grow hyphae within 1 dpi, but neighboring intact wheat cells will launch basal defense responses, including callose deposition, chloroplast responses, and plasmodesmata closure, therefore hyphal growth can be temporarily hindered **a**. Later, wild-type *F. graminearum* hyphae can activate *fg3_54* cluster expression and produce fusaoctaxin A (FA). FA can be diffused into neighboring cells, cause transcriptional changes, host defense suppression and plasmodesmata reopening, which allows fungal hyphae to penetrate cell wall and enter cell-to-cell invasion mode **b**; the strains lacking *fg3_54* (mutant *F. graminearum* or non-wheat pathogen Fusarium species) cannot produce FA, and the host defenses accumulate to a stronger degree, and subsequently block fungal growth **c**. With exogenous application, FA can inhibit wheat defense responses and then enable hyphal cell-to-cell invasion in wheat **d**. Transmission electron microscopic images are shown in the bottom. PCW plant cell wall, P: papillae-like cell wall deposition, H: *F. graminearum* hyphae. Scale bar represent 1 μm

concentration of fusaoctaxin A is in the same order of magnitude as the working concentration of some plant CLE peptide hormones, including CLV3 in shoot apical meristem development regulation and KIN in vascular tissue development regulation[73].

Considering that fusaoctaxin A selectively acts on plant cells and has no toxic effects on animal cells, the target of fusaoctaxin A may be plant specific. Our results for the cellular responses and gene expression analysis suggest that its target might be present in the chloroplast and/or plasmodesmata. Uncovering the mechanism of actions of fusaoctaxin A should facilitate the development of cultivars with durable resistance against various *Fusarium*-associated diseases.

## Methods

**Fungal strains and constructions**. All fungal strains used in this study are listed in Supplementary Table 1. The wild-type strain *F. graminearum* PH-1 (NRRL 31084) was used as the parent strain for transgenic construction. Conidia and mycelia were produced in liquid mung bean broth and yeast extract peptone dextrose (YEPD) liquid medium, respectively. *Fusarium avenaceum* (isolate M130-N458, CGMCC 3.3630) and *Fusarium poae* (isolate NF732, CGMCC 3.4601) were obtained from China General Microbiological Culture Collection Center. *Verticillium dahlia* V592, *Metarhizium robertsii* ARSEF 2575 and *Fusarium oxysporum* f. sp. *cubense* tropical race 4 (NRRL 54006) were obtained from Drs. Huishan Guo, Chengshu Wang and Sijun Zheng respectively.

The gene replacement constructs were generated using the split-marker recombination approach[74] as illustrated in Supplementary Fig. 2c. Supplementary Table 2 lists the primer sequences. PCR products were transformed into protoplasts of PH-1, and the resulting hygromycin-resistant transformants were screened using genomic DNA PCR with primers internal (Primer IF and IR) and external (Primer

U and D) to the target gene. Gene deletion mutants were further verified by Southern blot analysis (Supplementary Fig. 2).

For fgm4 complementation assay, a 2.4-kb region containing the gene and the promoter region was amplified using the primer pair PC9 and PC10 (Supplementary Table 2) and then cloned into JM45 (a vector with a neomycin-resistant cassette). The resulting construct (JM70) was transformed into protoplasts of FG-Δfgm4.

The plasmids used to study FGM4 subcellular localization were derived from the vma3 promoter-mRFP vector[38]. Fgm4 cDNA was amplified and inserted in frame at the 5′-end of the mRFP or 3′-end of the eGFP coding sequence. For gene constitutive expression, cDNA sequences were amplified and inserted into vectors driven by the pEF1 promoter[37].

A transcription activation experiment was performed using the MATCHMAKER GAL4 Two-Hybrid System (Clontech). The plasmids and the primers used in this study are provided in Supplementary Table 2.

**Quantitative RT-PCR and RNA sequencing analysis**. Total RNA was extracted from mycelia and/or infected wheat coleoptiles using TRIzol reagent (Invitrogen) according to the manufacturer's protocol. cDNA was prepared using reverse transcriptase M-MLV (TaKaRa Biotechnology). Quantitative real-time PCR experiments were performed with SYBR Green detection on an iCycler (Bio-Rad). The β-tubulin gene (FGSG_09530) was used as a reference to F. graminearum. Ta54227, which encodes a cell division control protein AAA-superfamily of ATPases and is ranked one of the most stable genes in wheat, was used as a reference for wheat[75].

For RNA-seq, total RNA was extracted and checked using an Agilent Bioanalyzer 2100 (Agilent technologies) and then used for library preparation and sequencing (Illumina HiSeq 2500) by the Shanghai Biotechnology Corporation (Shanghai). For RNA-seq of in vitro-grown F. graminearum, two biological replicates of wild-type PH-1, FG-Δfgm4 mutant, WT-OE::fgm4 and FG-Δfg3_54:: fgm4 strains were sequenced. For each sample 31–44M clean RNA-seq reads were obtained and mapped to the F. graminearum PH-1 genome sequence [http://www. broadinstitute.org/annotation/genome/fusarium_group/MultiDownloads.html] using TopHat v2.0.9. Transcript abundance estimations of the annotated gene catalogue were performed with Cufflinks v2.1.1. Coverage plots of the RNA-seq reads were generated using the Integrative Genomics Viewer[76].

Three biological replicates of the following samples were subjected to RNA sequencing: mock-inoculated coleoptiles (3 dpi), wild-type F. graminearum-inoculated coleoptiles (3 dpi), FG-Δfg3_54-inoculated coleoptiles (3 dpi), coleoptiles after treated with fusaoctaxin A (1 nmol per seedling in 0.5% Tween-20 water solution) for 3 h, coleoptiles treated with solvent (0.5% Tween-20) for 3 h, coleoptiles treated with fusaoctaxin A for 24 h and coleoptiles treated with solvent (0.5% Tween-20) for 24 h. Total RNA was isolated using EasyPure Plant RNA kit (Transgen biotech) according to the manufacturer's protocol. RNA integrity was assessed using the Bioanalyzer 2100 (Agilent Technologies) with a minimum RNA integrated number (RIN) value of 7. For each sample, 33–95M RNA-seq clean reads were obtained that mapped to the Triticum aestivum Chinesespring genome sequence [ftp://ftp.ensemblgenomes.org/pub/plants/release-39/fasta/ triticum_aestivum/dna/Triticum_aestivum.TGACv1.dna.toplevel.fa.gz] using HISAT2 (hierarchical indexing for spliced alignment of transcripts) v2.0.4[77]. Sequencing read counts were calculated using Stringtie[78,79] (v.1.3.0). Then, expression levels from different samples were normalized by the Trimmed Mean of M values (TMM) method[79]. The normalized expression levels of different samples were converted to FPKM (Fragments Per Kilobase of transcript per Million mapped fragments). The edgeR package of R was used to analyze the difference between intergroup gene expression, the P-values were calculated, and the multiple hypothesis test was performed. The P-value threshold was determined by controlling the FDR (False Discovery Rate) with the Benjamini algorithm. The corrected P-value is called the q-value. Differentially expressed genes (DEGs) were defined as transcripts with a fold change in expression level (according to the FPKM value) greater than 2.0 and a q-value less than 0.05. GO enrichment analysis was performed with the clusterProfiler package of R and the enrichment criteria including a q-value < 0.05. Heatmaps of specific genes were generated using the pheatmap package of R. PCA analysis was visualized using the scatterplot3d package of R.

**Bioinformatics and statistical analysis**. The phylogenetic tree (Neighbor-Joining method, MEGA6) was constructed based on the DNA sequences of the DNA-directed RNA polymerase II subunit (RPB2)[23]. Genome alignment of related pathogenic fungal contigs containing the fg3_54 cluster or neighboring genes was performed using Mauve[80].

The sample sizes and repeated experimental time for wheat infection and maize stalk infection assays are provided in Supplementary Data 3. Data are expressed as the means ± s.e.m unless otherwise indicated. The Student's t-test (unpaired, two-tailed) and one-way ANOVA followed by Tukey's multiple comparison test were calculated in Excel (Microsoft) or GraphPad Prism 7 (GraphPad software).

**Fungal growth and infection assays**. For the fungal stress sensitivity assays, each strain was grown in a 9-cm Petri dish containing solid CM with $H_2O_2$ or 2,2'-

dipyridyl (Sigma-Aldrich). Diameters of 4-day-old colonies were measured to evaluate sensitivity.

Wheat (Triticum aestivum) cultivars Zhongyuan 98–68 and Bobwhite were used for floret infection and for coleoptile infection[33]. Maize (Zea mays) cultivar B73 was used for the stalk infection assay[37]. Seeds were randomly planted and cultivated under the same condition. 5, 20, and 10 μL of spore suspension ($10^6$ per mL) were inoculated on coleoptiles, florets of wheat and maize stalk, respectively. Lesion sizes on the coleoptiles were measured at 7 dpi, and symptomatic spikelets on wheat spikes and lesion sizes on maize stalks were measured at 14 dpi.

Fusaoctaxin A was artificially synthesized by ChinaPeptides (Shanghai). The octapeptides were prepared as a 5 mM stock solution in 0.5% Tween-20, diluted to the indicated concentration with 0.5% Tween-20, and then 2 μL of solution per seedling was added to the wounded site of coleoptiles at 1, 2, and 3 dpi. When using purified fusaoctaxin A for infection assay, the experiments were performed in a double-blinded manner.

For detection of fusaoctaxin A in the F. graminearum-wheat pathosystem, ~0.2 g of F. graminearum-infected coleoptiles and spikes at 7 dpi were immersed in methanol and subjected to vacuum extraction for 30 min. The extracts were then evaporated and redissolved in 1 ml of methanol. The dissolved samples were analyzed using the Agilent G6520A accurate-mass Q-TOF LC–MS system with an Agilent Zorbax column (SB-C18, 4.6 × 250 mm, 5 μm). The injected volume was 10 μL. The flow rate was 1 mL per min, and the gradient elution used mobile phase A (water supplemented with 0.1% HCOOH) and mobile phase B (acetonitrile supplemented with 0.1% HCOOH). The gradient profile was as follows: 0–2 min (5% phase B), 2–20 min (5% to 90% phase B), 20–25 min (90% phase B), 25–25.5 min (90% to 5% phase B), and 25.5–30 min (5% phase B). The mass range was 150–1500 m/z; nebulizer pressure 40 psig; drying gas N2 350 °C, 9 L/min; ESI Vcap 3500 V, fragmentor 160 V; skimmer 65 V; and Oct RF Vpp 750 V.

To measure the concentration of fusaoctaxin A in the wheat coleoptiles, the crude extracts were analyzed using LC–MS/MS on a QTRAP 6500+ mass spectrometer coupled to an ExionLC system (AB SCIEX). Chromatographic separation was performed on a Waters ACQUITY UPLC HSS C18 SB column (2.1 mm × 50 mm, 1.8 μm). The injected volume was 2 μL. The mobile phase consisted of a solvent A (0.1% formic acid in water) and a solvent B (acetonitrile). The flow rate was 0.6 mL per min. The flow gradient was as follows: 0–1.5 min (5% solvent B), 1.5–3 min (5% to 95% solvent B), 3–4 min (95% solvent B), 4–4.1 min (95% to 5% solvent B) and 4.1–6 min (5% solvent B). Fusaoctaxin A was detected by electrospray ionization using multiple reaction monitoring in positive ion mode at m/z 773.0 to 157.0. The MS parameters were as follows: 5.5 kV of capillary voltage, 120 V of declustering potential, 54 V of collision energy, 550 °C of source temperature, 55 psi of ion source gas 1 (GS1) and 55 psi of ion source gas 2 (GS2). The Fusaoctaxin A standard was dissolved in methanol at concentrations of 10 μM, 1 μM, 100 nM, 50 nM, and 10 nM to generate a standard curve for calculating concentration. Analyst 1.6.1 (AB SCIEX) software was used for data acquisition and analysis.

**Production, purification and characterization of fusaoctaxin A**. The WT-OE:: fgm4 strain was incubated in solid Complete Medium for 3 days. A spot of mycelium was then transferred to a 500-mL Erlenmeyer flask containing 150 mL of YEPD liquid medium at a shaking speed of 150 rpm for 3 days. The fermentation culture was extracted with HP-20 resin, and then eluted with methanol to yield crude extracts with an evaporator. The crude sample was dissolved in a methanol solution with 40% formic acid and was then subjected to purification on a MCI chromatographic column, yielding a fraction containing fusaoctaxin A. This fraction was separated on a gel column (Sephadex LH-20, 1.5 × 200 cm), and was further purified by HPLC for semipreparation on a phenomenex Luna C18 column (5 μm, 250 × 10 mm), where MeCN/H₂O containing 0.01% TFA was used for elution. The fraction containing fusaoctaxin A was evaporated under reduced pressure to yield an amorphous mud. The NMR data for fusaoctaxin A was recorded on a Bruker Advance III (500 MHz).

Fusaoctaxin A characterization data: $^1$H NMR (500 MHz, DMSO-$d_6$): δ 8.11 (d, $J$ = 7.4 Hz, 1 H), 8.07 (d, $J$ = 7.8 Hz, 1 H), 7.94 (d, $J$ = 7.8 Hz, 1 H), 7.87 (d, $J$ = 8.9 Hz, 1 H), 7.87 (d, $J$ = 8.2 Hz, 1 H), 7.66 (d, $J$ = 7.8 Hz, 1 H), 7.43 (d, $J$ = 8.7 Hz, 1 H), 4.42 (dd, $J$ = 8.9, 5.3 Hz, 1 H), 4.41 (m, 1 H), 4.36 (dt, $J$ = 7.8, 6.4 Hz, 1 H), 4.30 (dt, $J$ = 7.8, 6.4 Hz, 1 H), 4.23 (m, 1 H), 4.23 (m, 1 H), 3.74 (m, 1 H), 3.59 (d, $J$ = 6.4 Hz, 2 H), 3.55 (m, 1 H), 3.50 (m, 1 H), 3.28 (dd, $J$ = 10.5, 5.0, 1 H), 3.17 (dd, $J$ = 10.5, 6.9, 1 H), 2.93 (m, 2 H), 2.21 (m, 2 H), 1.98 (m, 1 H), 1.83 (m, 1 H), 1.80 (m, 2 H), 1.60 (m, 1 H), 1.52, (m, 1 H), 1.45 (m, 2 H), 1.31 (m, 1 H), 1.26, (m, 2 H), 1.20 (d, $J$ = 7.3 Hz, 3 H), 1.07 (m, 1 H), 0.85 (m, 3 H), 0.83 (m, 3 H), 0.83 (m, 3 H), 0.82 (m, 3 H), 0.80 (m, 3 H), 0.79 (m, 3 H),0.78 (m, 3 H), 0.76 (d, $J$ = 6.9 Hz, 3 H); $^{13}$C NMR (125 MHz, DMSO-$d_6$): δ 172.4, 171.3, 171.2, 171.1, 170.7, 170.0, 169.8, 63.9, 61.6, 61.4, 57.4, 55.1, 54.7, 51.3,48.7,48.2,47.6, 40.6, 39.9, 37.3, 32.0,30.6, 25.8, 24.2, 24.1, 23.4, 23.1, 21.7, 21.4, 21.0, 19.0, 17.6, 18.8, 14.3, 11.6; +ESI-HRMS was analyzed on the Bruker (UHR-TOF) maXis 4G, [M + H]$^+$ cald. for $C_{36}H_{69}N_8O_{10}$ 773.5131, found 773.5135. Detailed characterization data are provided in Supplementary Data 1. Chemical structures were drawn using the Nature Chemistry template, ChemDraw files are provided in Supplementary Data 4.

**Modified Marfey's analysis**. For sample hydrolysis, fusaoctaxin A (50 µg) was dissolved in 6 N HCl (100 µL), and then was subjected to heating at 110 °C for 12 h. The hydrolysate was concentrated under a vacuum. The resulting dryness was treated with 1 M NaHCO$_3$ (20 µL) and an acetone solution with 1% L-FDAA (40 µL) at 40 °C for 1 h. The reaction mixture was neutralized with 1 M HCl (20 µL), diluted with MeCN (100 µL) and filtered with PTFE (0.22 µm) prior to LC/MS analysis.

For the preparation of L-FDAA-derived amino acid standards, each solution of 50 mM L - or D-amino acids dissolved in H$_2$O (50 µL) was treated with 1 M NaHCO$_3$ (20 µL) and an acetone solution with 1% L-FDAA (100 µL) at 40 °C for 1 h. The reaction mixture was neutralized with 1 M HCl (20 µL), diluted with MeCN (810 µL) and filtered with PTFE (0.22 µm) prior to LC/MS analysis. Analyses of an aliquot (5 µL) by LC–MS (LTQ-XL, Thermo scientific) were conducted on a phenomenex Luna C18 column (5 µm, 250 × 10 mm) at 50 °C and 1 mL per min over a 55 min linear gradient elution period using MeOH/H$_2$O (with a 5% isocratic modifier of 1% formic acid in MeCN, from 25% to 70%) and monitored using the ESI-positive mode.

**Microscopic analysis**. Fungal and plant samples were imaged on an Olympus BX51 or Olympus Fv10i confocal microscope. Callose staining and dye loading assays were performed on coleoptiles that had been detached from seedlings. Samples were stained with 0.05% (w/v) aniline blue in 0.067 M phosphate buffer (pH 6.8) for 15 min and then rinsed with the same phosphate buffer.

For the plasmolysis assay, coleoptiles were detached from seedlings after fusaoctaxin A treatment at the indicated times, and mounted on a glass slide. Microscopic images were captured before and after treatment with 1 M NaCl for 5 min. The cells within a region ~3 mm away from the wounded edge were imaged.

Cell viability assays were performed using the Acridine Orange/Ethidium Bromide Assay Kit (40746ES01, Yeasen). Octapeptide-treated and mock-treated coleoptiles at 5 h were stained with Acridine orange and ethidium bromide (1 µg per mL, 1:1) in phosphate-buffered saline (PBS) at room temperature for 5 min, then washed twice with PBS. Slides were observed using an Olympus Fv10i with excitation/emission spectra of 502 nm/526 nm for AO and 359 nm/461 nm for EB. Green signals (from AO staining) in nuclei indicate live cells (as illustrated in mock-treated cells), and orange signals (EB staining) in nuclei indicate dead cells (as illustrated in cells treated with 10% KOH for 5 min). The coleoptile cells within a region 3 mm away from the wounded edge were measured.

The plasmodesmal permeability assay was performed according to Drop-ANd-See method[53]. Carboxyfluorescein diacetate (Sigma-Aldrich) was prepared as a 50 mM stock in dimethyl sulfoxide, and diluted to 1 mM with distilled water. Next, 1 µL of dye solution was placed at the wound edge of the coleoptiles of wheat seedlings. The seedlings were incubated for 5 min, and then the wound edge was rinsed briefly and gently. Subsequently, the coleoptiles were detached from the seedling and immediately mounted on a glass slide for microscopy.

For transmission electron microscopy observation, 3 dpi *F. graminearum*-inoculated wheat coleoptiles were fixed with 2.5% glutaraldehyde in PBS (50 mM, pH 7.2) under vacuum infiltration, and stored at 4 °C overnight. The tissues were then washed with 50 mM PBS and fixed for 2 h in 1% osmic acid in PBS (50 mM, pH 7.2) at 4 °C. After three washes with 50 mM PBS, the tissues were dehydrated in an ethanol series (from 60% to a final concentration of 100% ethanol) and embedded in Epon812 resin. After staining with uranyl acetate for 5 min and lead citrate for 5 min, the grids were observed under an electron microscope (Hitachi, H-7650) at 80 kV.

**Chemical treatment on pollen tube and tobacco leaves**. Tobacco (*Nicotiana tabacum* cv Gexin No. 1) pollen tubes were cultured in medium [20 mM Mes, pH 6.0, 3 mM Ca(NO$_3$)$_2$, 1 mM KCl, 0.8 mM MgSO$_4$, 1.6 mM boric acid, 2.5% (w/v) Suc, and 24% (w/v) polyethylene glycol 4000]. The indicated concentration of fusaoctaxin A was added to the medium at the onset, and then pollen was allowed to germinate for 4.5 h before images were acquired.

For compounds treatment in *Nicotiana benthamiana* leaves, 50 µM fusaoctaxin A, 50 µM GIAVSTAG, 0.5% Tween-20 and sterile H$_2$O were infiltrated into *N. benthamiana* leaves separately. The infiltrated plants were grown at 25 °C. The phenotype was checked at 1 h after infiltration.

**Toxicity investigation on mammal cells**. To investigate the toxicity of fusaoctaxin A, three types of cell lines were used: the kidney cancer cell line ACHN, embryonic kidney 293T Cell, and liver cancer cell line Hep G2. Cells (250 cells per well) were placed in a 384-well plate (Corning), which was incubated for 16–20 h. Then the preformed compound working solution and positive reference working solution were added to the corresponding wells and further incubated for 72 h in the cell culture incubator. An equal volume of CellTiter-Glo® reagent was added to the corresponding well and incubated at room temperature for 10 min to stabilize the luminescent signal. The luminescence was recorded using PHERAStar Plus.

**Reporting summary**. Further information on experimental design is available in the Nature Research Reporting Summary linked to this article.

## Data availability

RNA-seq data have been deposited in the GEO database under accession codes GSE89867 and GSE117934. Accession codes for DNA sequences are listed in Supplementary Table 4. The source data underlying Figs. 1d, 2b, 2d, 2e, 2f, 2g, 5b, 5d, 5e, 6c, 6d, 7c, 7d and Supplementary Figs. 1b, 2b, 2c, 3a, 3b, 4b, 4d, 5d, 6b, 7a, 7b, 7c, 9a, 9c, 10d are provided in Supplementary Data 3. Other data and genetic materials used for this paper are available from the authors upon reasonable request.

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

## Acknowledgements

We thank Dr. Sheila McCormick for editing the manuscript, Drs. Chen Yang and Lixia Liu for advice and technical instructions on amino acid analysis. Thanks to SIPPE core facility Drs. Wenli Hu and Yining Liu for support on MS. This work was supported by the Natural Science Foundation of China (31730077), the National Key Research and Development Program of China (2016YFD0100600), the Natural Science Foundation of China (21520102004), the Chinese Academy of Sciences (QYZDJ-SSW-SLH037), the Strategic Priority Research Program of the Chinese Academy of Sciences (XDPB0400, XDB11020500, XDB20020200) and the National GMO project (2016ZX08009-003).

## Author contributions

W.T. and W.L. conceived the project. L.J. performed most genetic experiments. H.T. performed structural elucidation of fusaoctaxin and configuration determination. W. Wang performed fusaoctaxin activity and quantity analysis. T.Y. and W. Wei performed some microscopic observations. B.P. and S.W. conducted the experiments involving fusaoctaxin purification. Y.L. and X.G. conducted some infection assays. D.Z. analyzed gene expression data. W.T. and W.L. wrote the manuscript.

## Additional information

**Competing interests:** The authors declare no competing interests.

