## [Peer Review File · Nature Communications]

Reviewers' comments:

Reviewer #1 (Remarks to the Author):

The manuscript by Dr Tang and co-workers describes the identification of a tri-peptide from *Fusarium graminearum*, which is involved in cell-to-cell movement in wheat. The study was initiated by a previously published observation that a gene cluster comprising NRPS5 and NRPS9 (FG3_54) was active in a coleoptile infection system. Furthermore, the group has previously shown that deletion of several genes in the cluster results in decreased virulence. In the present study this was further investigated by individual deletion of all members of the gene cluster, which also led to decreased virulence. Infection was then monitored using a previously developed wheat coleoptile infection system, where the mutants showed shorter lesions than the wild type (app. 30-60 % reduction). A fluorescent marker was also used to monitor the spread of wt and deletion mutants in the coleoptile infection system, which revealed that the mutant strains failed to penetrate neighboring cells. Staining of wheat cells showed that callose is formed as a defense reaction against the deletion mutants, but not against wt. A local transcription factor, FGM4, was identified through a yeast two-hybrid system. The TF has similarity to TFs from other NRPS gene clusters (this can be further highlighted). GFP fusion used to show that the protein is located to the nucleus. OE:FGM4 increase transcription of gene cluster (but apparently NRPS5 is less expressed than the others). To identify the metabolites produced by the gene cluster HPLC-UV was first used by this was unsuccessful. Instead TLC with ninhydrin staining was used and a compound was isolated and elucidated by NMR. The compound sILeuol consist of three amino acids (D-seryl-D-leucyl-L-Leucinol) and addition of the compound restored virulence in some deletion mutants (partial) and also enabled *F. oxysporum* to infect wheat.

The manuscript describes a very extensive study of the gene cluster and as the authors also state, this is one of only few examples of compounds from *F. graminearum* that has a proven role for pathogenicity. In many ways this is a very impressive work, but there are however some issues that must be addressed.

One of the most important things is the biosynthetic pathway. As NRPS5 contains eight modules and NRPS9 one module, it seems unlikely that the product should be a three amino acid compound. This discrepancy is only vaguely addressed in the manuscript and needs more attention. The authors show that premature deletions of NRPS5 at the C-terminal result in reduced virulence. To complete the examination deletion of the N terminal domains of NRPS5 could provide clues to the role of NRPS5. It could be that the first five modules are not required for sILeuol (which would explain the uneven transcription of the gene seen in figure 3d).

A more thorough description of the NRPSs is also needed (prediction of substrate, Stachelhaus codes..)

Identification of the compound(s) produced by the gene cluster is only done partially. First the extracts of Wt, KO and OE were analyzed by HPLC-UV (unsuccessfully) and then by TLC. Here ninhydrin staining helped to identify one major band, which was subsequently isolated and elucidated (sILeuol). When inspecting the gel there seems to be another strong band just above sILeuol and one or two more weaker bands. These bands were apparently not further examined, although these could be the potential end product or intermediates of the metabolic pathway. These bands should also be isolated and examined by LC-MS. Furthermore, the raw extracts from the different mutant strains should be analyzed by high resolution LC-MS, which could reveal other compounds which only are expressed by the OE mutant in vitro (e.g. using XCMS: generate heat map + PCA).

The observation, that sILeuol affects defense response in wheat, is very interesting. But the subsequent qRT-PCR of selected defense genes (WT vs KO) did not reveal expression differences. Here RNAseq or similar would have been more useful.

Minor issues

Describe which promoter was used for overexpression of the local transcription factor.

How was secondary metabolites extracted from in vitro cultures prior to HPLC-UV and TLC analyses?

Reviewer #2 (Remarks to the Author):

The manuscript NCOMMS-16-28154 entitled "A nonribosomal Peptide facilitates Cell-to-Cell Invasion of *Fusarium graminearum* in Wheat" by Jia et al., dissect the influence of the deletion of a putative two non-ribosomal peptide synthetase (NRPS)-encoding secondary metabolite gene cluster on *F. graminearum* virulence on wheat. Previous studies have identified this cluster to be only present in a subset of *Fusarium* species and expression analysis showed that cluster gene members are expressed during specific stages of infection. In this study the authors leverage the existing knowledge to elucidate in more detail the influence of whole cluster deletion mutants on pathogenicity and virulence on wheat and can show that deletion of the whole cluster as well as several single gene deletion mutants (some of them previously published) show reduced ability to infect coleoptiles and spikes as well as reduced cell-to-cell penetration abilities. The authors ectopically over-express a putative transcription factor (*fgm4*) and use this mutant to elucidate the putative secondary metabolite produced by the cluster

The concept of effectors or secondary metabolites to influence *Fusarium* virulence has previously been described for several *Fusarium* species and does in itself not represent a major advancement to the field of plant pathology. The data that build the basis for analyzing this specific cluster have been previously published. Some of the single cluster gene deletion mutants have already been shown to have reduced virulence (e.g. the ABC transporter encoding *fgm5*); some of them with contradicting results (deletion of *nrps9* was previously reported to have WT-like virulence). The novelty of this study lies in the proposed identification of the chemical compound produced by cluster 54. The authors ectopically constitutively express a putative transcription factor, *fgm4*, and isolate a compound that they identify as a tripeptide D-seryl-D-leucyl-L-leucinol. Addition of extracts from OE::*fgm4* and synthetically obtained peptide restore virulence of the cluster knock-out mutant and even enable a non-wheat pathogenic *Fusarium oxysporum* strain to infect wheat (Is this true for only the synthetically obtained peptide or also extracts from OE::*fgm4*?). However, the study fails to explain how a tripeptide could be made by a gene cluster that harbors two NRPS, one with one A domain, the other containing eight A domains. It is unprecedented that large NRPS like the ones studied here would only make a small peptide, without explaining any cleaving of a larger metabolite into the identified tripeptide. Furthermore, the presence of epimerase domains in NRPS5 would suggest that the incorporated amino acids would change their L/D confirmation, unless some would be non-functional. In order to provide more inside into the biosynthesis of the compound produced from cluster 54 more analytical efforts are needed. Single deletion strains of every cluster gene putatively involved in modification of the initial chemical scaffold are desirable to explain how the non-proteinogenic leucinol is being made, which gene could be involved in the reduction of the carboxy group.

The authors ectopically express *fgm4* and perform RNA-seq experiments to show expression of cluster 54 is specific. The Southern analysis of the OE::*fgm4* strains seem to be missing and should be added. Why was the transcription factor not over-expressed by a promoter replacement at the original gene locus? It would be more convincing if the scale in Fig. 4d would be uniform. Also more information on Fig S6e is needed to indicate where the data was obtained from. Since the authors performed RNA-seq analysis, what is the change in global gene expression of OE::*fgm4* to WT? The data in Fig.S7 does not contain gDNA controls. Also, do the primer combinations span introns?

Overall, the elucidation of the chemical compound that the authors attribute to the gene cluster is not strong enough to provide evidence that the identified tripeptide really is the final product made by cluster 54 and therefore I do not recommend publication in its current form.

Minor comment:

The Southern strategies in Fig S2 are unconventional. Why did the authors not use a flank as probe that bind in both, the WT and the mutants. Without a schematic overview where the restriction enzymes are located the strategies are hard to follow. It also appears that all three whole cluster gene deletion mutants have different patterns. More details are needed to assure what parts of the cluster are deleted. And which of these mutants was used to create OE::fgm4?

Response to review:

We appreciate both review, your comments are very helpful. Accordingly, we performed new experiments including RNA sequencing of tripeptide sILeuol treated wheat samples, construction of N-ter deletion mutant of FgNRPS5 and product analysis, examination of sILeuol effects on more Fusarium species, etc. We also made more speculations on the complete product heptapeptide of FgNRPS5, and incubated synthesized heptapeptide with *F. graminearum* and identified sILeuol tripeptide in the culture, which helps to understand sILeuol might be a processed product of Fg3_54 cluster. With these new data and analysis, we added a new Figure (Fig. 6), modified three Figures (Fig. 2,3,8), added nine more supplementary figures and two more supplementary tables to address your previous concerns. Here we submit a substantially revised manuscript (a clear version at first and a version with changes highlighted in the end) with point-to-point response to review below for your consideration.

Reviewer #1-1 *“The manuscript describes a very extensive study of the gene cluster and as the authors also state, this is one of only few examples of compounds from F. graminearum that has a proven role for pathogenicity. In many ways this is a very impressive work, but there are however some issues that must be addressed. One of the most important things is the biosynthetic pathway. As NRPS5 contains eight modules and NRPS9 one module, it seems unlikely that the product should be a three amino acid compound. This discrepancy is only vaguely addressed in the manuscript and needs more attention.”*

Answer: We appreciate reviewer’s recognition of our work, and we agree with reviewer 1 (also reviewer 2) that the biosynthetic pathway is the key concern. FgNRPS5 comprises “A”-6×“C-A-PCP-E”-“C-A-PCP-R”, presumably produces a heptapeptide or octapeptide, while the active metabolite we identified is a tri-peptide, how comes? A reasonable guess, the tripeptide sILeuol might be a processed product from the original heptapeptide. In previous submission, we only vaguely mentioned this possibility. In this revision, following suggestions from both reviewers, we made efforts in three parts (as explained below in response to the points of reviewer#1-2, -3, -4, labeled as biosynthetic pathway related -1, -2, -3).

Reviewer #1-2 *“The authors show that premature deletions of NRPS5 at the C-terminal result in reduced virulence. To complete the examination deletion of the N terminal domains of NRPS5 could provide clues to the role of NRPS5. It could be that the first five modules are not required for sILeuol (which would explain the uneven transcription of the gene seen in figure 3d).”*

Answer (biosynthetic pathway related-1): According to review suggestion, we constructed N-ter deletion mutant of FgNRPS5 lacking the first “A” domain (Supplementary Fig 13a) in FGM4 overexpressing strain. TLC analysis of its medium extracts showed reduced accumulation on N1 product (Supplementary Fig 13f). The N-ter deletion mutant strain also showed reduced virulence in wheat coleoptile infection (Supplementary Fig. 13), consistent with putative role for N-ter first module of NRPS in increase efficiency of initiation of peptide biosynthesis. We also made the FgNRPS5 lacking N-ter 5 modules mutant, found it has reduced virulence in wheat coleoptile infection. But we are still in the process of

transforming FGM4 overexpression fragment into this mutant, so haven't got metabolite analysis results, due to time limits. Along with C-ter deletion mutants can not produce the tripeptide at all (Supplemental Fig. 10 and 12), this additional piece of evidence (shown in supplementary figure 13 and ms text line 352-362 in this revision) slightly favor the hypothesis that sLeuol corresponds to the C-ter modules of FgNRPS5.

Reviewer #1-3 “*Identification of the compound(s) produced by the gene cluster is only done partially. First the extracts of Wt, KO and OE were analyzed by HPLC-UV (unsuccessfully) and then by TLC. Here ninhydrin staining helped to identify one major band, which was subsequently isolated and elucidated (sLeuol). When inspecting the gel there seems to be another strong band just above sLeuol and one or two weaker bands. These bands were apparently not further examined, although these could be the potential end product or intermediates of the metabolic pathway. These bands should also be isolated and examined by LC-MS. Furthermore, the raw extracts from the different mutant strains should be analyzed by high resolution LC-MS, which could reveal other compounds which only are expressed by the OE mutant in vitro (e.g. using XCMS: generate heat map + PCA).*”

Answer (biosynthetic pathway related-2): We agree that the identification of the gene cluster products. We made more efforts to indentify weaker band (N2) and product of $\Delta fgm9$ mutant, resulted an additional Supplementary Fig. 11, and stated in results part lines 344-351, 363-368, as below:

“To further test whether sLeuol is a direct product of FG3_54 cluster NRPSs, we performed chromatography analysis of various mutant strains with *FGM4* overexpression. The ninhydrin positive bands (N1 and N2) were detected in *FGM4* overexpression strains of $\Delta fgm1$, $\Delta fgm2$, $\Delta fgm3$ and $\Delta fgm5$, but not the *NRPS5* or *NRPS9* deletion mutants (Supplementary Fig. 10). Furthermore, the EICs show that N1 ($[M+H]^+$: 318.24) and N2 ($[M+H]^+$: 302.24) are absent in the extracts of $\Delta nrps5$, $\Delta nrps9$ and $\Delta fg3_54$ mutants with *FGM4* constitutively expressed (Supplementary Fig. 11d, e). These results support that NRPS5 and NRPS9 are required for the production of the ninhydrin positive metabolites.”

“Furthermore, in *FGM4* overexpression strains of $\Delta fgm9$, an additional ninhydrin positive band N3 is present (Supplementary Fig. 11a). Q-TOF MS analysis of recovered N3 metabolites showed that its m/z is 316.22 (Supplementary Fig. 14a, b), slightly smaller than sLeuol (m/z 318.24). The N3 metabolite is not detected in other mutant strains with *FGM4* expression (Supplementary Fig. 14c,d). FGM9 showed similarity to a short-chain alcohol dehydrogenase, in line with the idea that FGM9 may modify FgNRPS5 original products, probably to achieve leucinol at C-ter.”

Reviewer #1-4 “*A more thorough description of the NRPSs is also needed (prediction of substrate, Stachelhaus codes..)*”

Answer (biosynthetic pathway related-3): Yes, in this revision, we provide more thorough description of FgNRPS5 and extracted the nonribosomal code (also called Stachelhaus code) in A domains of FgNRPS5, made alignment with known codes to help substrate prediction, resulted an additional supplementary figure 19. In this revision, we stated these description in the discussion as below (lines 540-567): “The RNA-seq reads of FgNRPS5 cover 35 Kb region (Fig. 3d), indicating that the full-length protein should consist of 11,299 a.a., and

comprise “A”-6 × “C-A-PCP-E”-“C-A-PCP-R” multimodular NRPS (Fig. 1a). The presence of E (epimerization) domains in FgNRPS5 suggests that the incorporated amino acids would change their conformation from L to D. The presence of R (reductase) domain suggests two types of product releasing by reduction. Some R domains are capable of a four-electron reduction, in which the peptidyl thioester attached to the PCP is reduced first to an aldehyde and then to an alcohol⁶⁵. The FgNRPS5 C-terminal R domain resembles Lys2-type reductase which produces an aldehyde from a carboxylic acid as in *Aspergillus flavus* LnaA and LnbA with an A-T-R domain arrangement⁶⁶. Therefore FgNRPS5 might produce a heptapeptide composed of six D-amino acid residues followed by a L-amino acid-derived aldehyde. Given that FGM9 dehydrogenase might reduce an aldehyde to an alcohol, it is possible that the complete final product of NRPS5 and FGM9 is a heptapeptide composed of six D-amino acid residues followed by a L-amino-alcohol. Therefore the tripeptide sLeuol which has two D-amino acid residues followed by an L-amino-alcohol might be a processed product correspond to the last three modules of FgNRPS5.

“NRPS A domains select monomers to incorporate, and the amino acid residues occupying the ten key positions that are relevant for substrate specificity within A domains are referred to as the nonribosomal code or Stachelhaus code⁶⁷. However nonribosomal codes from FgNRPS5 showed low similarity to known codes from bacteria and fungi NRPS (Supplemental Fig. 19), therefore it is difficult to predict substrates of FgNRPS5.”

We further moved beyond, because we found that: “Interestingly, nonribosomal codes of FgNRPS5 A2, A4 are similar to A6, and A3, A5 are similar to A7 (Supplemental Fig. 19). If our speculation is true that sLeuol does correspond to the last three modules of NRPS, the substrates of A6 and A7 could be serine and leucine respectively, then substrates for A2, A4 might be serine, and substrates for A3 and A5 might be leucine. Given that E domains also present in modules 2-7, the complete product of FgNRPS5 and FGM9 can be predicted as D-Seryl-D-Leucyl-D-Seryl-D-Leucyl-D-Seryl-D-Leucyl-L-Leucinol (slslsLeuol). In a preliminary test of this prediction, we applied chemically synthesized slslsLeuol to *in vitro* cultured *F. graminearum*, tripeptide sLeuol (m/z 318.24) can be identified in the medium along with heptapeptide slslsLeuol (m/z 718.47) (Supplementary Fig. 20). sLeuol was also identified in the medium of $\Delta fg3_54$ *F. graminearum* incubated with slslsLeuol (Supplementary Fig. 20 c,d). Therefore, it is possible that tripeptide sLeuol is the processed product of slslsLeuol, while the processing was achieved by genes not located in FG3_54 cluster. Furthermore, the addition of heptapeptide slslsLeuol also restored virulence of $\Delta fg3_54$ on wheat coleoptiles (Supplementary Fig. 20e), and caused rapid cell death of tobacco leave cells (Supplementary Fig. 16b). These results suggest that the heptapeptide might be highly toxic, and probably processed immediately after synthesized, which may explain why we couldn’t detect the heptapeptide either in FGM4-expressing *F. graminearum* or in *F. graminearum* infected wheat tissues.”

Supplementary Figure 20: Identification of sIsIsLeuol and sLeuol in medium of fungal culture.

(a–d) Q-TOF LC-MS analysis of sIsIsLeuol (a and b) and sLeuol (c and d) in the medium extracts of fungal culture. Fungal strains were cultured in 200 ml YEPD with 1 mg sIsIsLeuol added for 3 days. (e) Addition of heptapeptide sIsIsLeuol aids $\Delta fg3_54$ mutant invasion in wheat coleoptiles. Wheat seedlings were inoculated with $\Delta fg3_54$ and then treated with sIsIsLeuol at 1 dpi. Lesions were measured at 7 dpi. Data are representative of three individual experiments. Data are means \pm s.d. (sample size $n=12$). * $P < 0.001$ (Student's t test). Scale bar represents 1 cm.

The above experiments based on speculation supports that the tripeptide sLeuol can be final

product of FgNRPS5 and FG3_54 cluster. So we would like to include this part of results (in blue) in this manuscript to relieve concerns on sILeuol biosynthesis pathway, but we also have some hesitation because the speculations of substrate prediction are purely based on signature a.a. similarity. Also it suggests genes not included in this cluster may also be involved in processing sILeuol. Many more product analysis can be done based on our individual gene deletion in FGM4 overexpression strain cultures. But it would take another whole paper to elucidate the complete biosynthesis pathway, really beyond the scope of our manuscript, which is identification of a novel simple compound as an effector directly functions in such an important crop disease, wheat Fusarium head blight, and initial elucidation of its mechanism in pathogenesis.

Reviewer #1-5 “*The observation, that sILeuol affects defense response in wheat, is very interesting. But the subsequent qRT-PCR of selected defense genes (WT vs KO) did not reveal expression differences. Here RNAseq or similar would have been more useful.*”

Answer: We agree, and performed RNA seq to compare wheat gene responses in coleoptiles infected with wild-type, mutant *F. graminearum* lacking FG3_54 cluster, or treated with sILeuol. With these RNA seq data, we now identified 791 candidate genes as wheat defense related genes that suppressed by sILeuol (Supplementary Table 6), and enhanced our results part in lines 411-433. For example, “GO enrichment analysis further showed that genes with 1,3-β-D-glucan synthase activity (GO:0003843) are significantly enriched in the 791 genes. These results are in line with that sILeuol might suppress callose (i.e. 1,3-β-D-glucan) synthesis to facilitate *F. graminearum* invasion in wheat.”

Reviewer #1-6 Minor issue “*Describe which promoter was used for overexpression of the local transcription factor.*”

Answer: The FGM4 transcription regulator was driven by the *F. graminearum* constitutive expression promoter EF1-A (FGSG_08811, translation elongation factor 1 alpha), as we previously used in Zhang et al., 2016 PLoS Pathogens 12(3): e1005485. We made this clear in methods part, under the subtitle “Fungal Transformation”.

Reviewer #1-7 Minor issue “*How was secondary metabolites extracted from in vitro cultures prior to HPLC-UV and TLC analyses?*”

Answer: As we showed in previous version Supplemental Figure 8a (=Supplementary Fig. 9a in this revision), the medium fraction of in vitro culture were extracted by ethyl acetate, followed by evaporation. Then resolved in 90% methanol:n-hexane(1:1 v:v), the methanol extracts were then evaporate again. The residues were dissolved in water with 1% DMSO for activity analysis or in methanol for LC and TLC analyses.

Reviewer #2-1 “*The concept of effectors or secondary metabolites to influence Fusarium virulence has previously been described for several Fusarium species and does in itself not represent a major advancement to the field of plant pathology. The data that build the basis for analyzing this specific cluster have been previously published. Some of the single cluster gene deletion mutants have already been shown to have reduced virulence (e.g. the ABC transporter encoding fgm5); some of them with contradicting results (deletion of nrps9 was*

previously reported to have WT-like virulence). The novelty of this study lies in the proposed identification of the chemical compound produced by cluster 54. The authors ectopically constitutively express a putative transcription factor, *fgm4*, and isolate a compound that they identify as a tripeptide D-seryl-D-leucyl-L-leucinol.”

Answer: We appreciate reviewer’s recognition the novelty of identification of a specific compound product of Fg3_54 cluster as a virulence factor of *F. graminearum*. We like to add the following points to argue the significance of our discovery.

1) In the field of plant pathology, particularly for important pathogens in *Fusarium* species, identified effectors are still limited, such as 'secreted in xylem'(SIX) proteins of *Fusarium oxysporum*, secreted lipase FGL1 and trichothecene mycotoxins DON of *Fusarium graminearum*. sLeuol, as a simple linear tripeptide, is distinct from all known virulence effectors, represents a new type peptide/secondary metabolite effector.

2) In Zhang et al., 2012 Plant Cell paper, we previously detected the expression of the eight member genes in this cluster during infection of wheat coleoptiles in a whole-genome microarray hybridization with a very special laser microdissection sampling, and reported knockout mutants of two member genes FGM5 (ABC transporter) or FGM2 (a putative polysaccharide deacetylase) had reduced virulence in wheat spike infection, but the knockout of *FgNRPS9* didn’t reduce *F. graminearum* virulence in wheat spike infection. The role of putative core nonribosomal peptide synthase enzyme *FgNRPS5* wasn’t identified. *FgNRPS5* was even thought to be a pseudogene based on the presence of stop codons and frameshifts in the predicted sequence. So, until this work, it was still not clear whether the whole cluster functions in wheat spike infection (i.e. wheat head blight infection), or just some individual genes contribute to virulence independent of the cluster product. It is the solid evidence in this work that proved the cluster product is a virulence factor. It is probably not that significant to identify a couple genes that are required for full virulence in *F. graminearum*, given that more than 200 genes have been reported as so (see PHI base: pathogen host interactions database, <http://www.phi-base.org/>). It is significant to identify an acting molecule directly functions in native pathogen-host interactions.

3) Coleoptile and spike are different organs of wheat, it is thought fungal pathogens may have core effectors that function when infect various organs, also have organ-specific effectors that only function when infect one organ but not another. So it is understandable that knockout of *FgNRPS9* didn’t reduce *F. graminearum* virulence in wheat spike infection, but reduced virulence in wheat coleoptile infection.

Reviewer #2-2 “Addition of extracts from OE::*fgm4* and synthetically obtained peptide restore virulence of the cluster knock-out mutant and even enable a non-wheat pathogenic *Fusarium oxysporum* strain to infect wheat (Is this true for only the synthetically obtained peptide or also extracts from OE::*fgm4*?).”

Answer: Thanks for comment. We now made it clear in the revision, both medium extracts of FGM4-expressing *F. graminearum* and the synthetic sLeuol restored virulence of cluster knockout mutant (Fig. 4a) and enabled *Foc TR4* in invade wheat (Fig. 6).

Reviewer #2-3 “However, the study fails to explain how a tripeptide could be made by a gene cluster that harbors two NRPS, one with one A domain, the other containing eight A domains.

It is unprecedented that large NRPS like the ones studied here would only make a small peptide, without explaining any cleaving of a larger metabolite into the identified tripeptide. Furthermore, the presence of epimerase domains in NRPS5 would suggest that the incorporated amino acids would change their L/D confirmation, unless some would be non-functional. In order to provide more inside into the biosynthesis of the compound produced from cluster 54 more analytical efforts are needed. Single deletion strains of every cluster gene putatively involved in modification of the initial chemical scaffold are desirable to explain how the non-proteinogenic leucinol is being made, which gene could be involved in the reduction of the carboxy group.”

Answer: This point is similar to reviewer#1-1. With new experiments we performed in the revision, the tripeptide sLeuol which has two D-amino acid residues followed by an L-amino-alcohol might be a processed product correspond to the last three modules of FgNRPS5. Please see answers biosynthetic pathway related 1-3 in page 1-5.

Reviewer #2-4 “*The authors ectopically express fgm4 and perform RNA-seq experiments to show expression of cluster 54 is specific. The Southern analysis of the OE::fgm4 strains seem to be missing and should be added. Why was the transcription factor not over-expressed by a promoter replacement at the original gene locus?”*

Answer: Thanks for pointing this out, the Southern blot verification of OE::fgm4 strains, using an intron-less FGM4 genomic DNA fragment as probe, is provided in Supplementary Fig. 8b in this revision. We agree that FGM4 overexpression can also be achieved by promoter replacement at the original gene locus. But in the split marker (Hyg resistance) recombination procedure (Catlett et al., 2003) we used, it will also introduce the hygromycin resistance gene (2 kb) into the cluster, which brings concerns in causing chromosomal architecture changes in the cluster region. Because *FGM4* and *FGM5* are located together with opposite transcription direction (Fig. 1a), the introduced fragment will be just upstream of the adjacent *FGM5* gene promoter. It is possible that the transcription of *FGM5* might be affected. Therefore we constructed transgenic strains carrying the *FGM4* gene driven by the fungal constitutive promoter other than promoter replacement at the original gene locus.

Reviewer #2-5 “*It would be more convincing if the scale in Fig. 4d would be uniform.”*

Answer: In this revision, we set the scale of RNA seq plot to the same in Fig. 4d.

Reviewer #2-6 “*Also more information on Fig S6e is needed to indicate where the data was obtained from.”*

Answer: For Supplementary Fig. 6 (renumbered to Supplementary Fig. 7e), the list of secondary metabolite biosynthesis core enzymes were obtained from the following reference: Connolly, L. R., K. M. Smith, and M. Freitag. 2013. 'The *Fusarium graminearum* histone H3 K27 methyltransferase KMT6 regulates development and expression of secondary metabolite gene clusters', *PLoS Genet*, 9: e1003916.

The expression data for these genes were obtained from the following papers and subjected to global normalization:

In vitro: Seong, K. Y., Zhao, X., Xu, J. R., Güldener, U. & Kistler, H. C. Conidial germination in the filamentous fungus *Fusarium graminearum*. *Fungal Genet. Biol.* 45, 389–399 (2008).

In wheat coleoptile: Zhang, X. W. et al. In planta stage-specific fungal gene profiling elucidates the molecular strategies of *Fusarium graminearum* growing inside wheat coleoptiles. *Plant Cell* 24, 5159–5176 (2012).

In maize stalk: Zhang, Y. et al. Cellular tracking and gene profiling of *Fusarium graminearum* during maize stalk rot disease development elucidates its strategies in confronting phosphorus limitation in the host apoplast. *PLoS Pathog.* 12, e1005485 (2016).

Reviewer #2-7 “*Since the authors performed RNA-seq analysis, what is the change in global gene expression of OE::fgm4 to WT?*”

Answer: Based on the RNA-seq comparison of *F. graminearum* strains WT-OE:FGM4 and WT, a total of 465 genes increased expression levels and 530 genes reduced expression levels significantly in WT-OE:FGM4, comparing to WT. The total gene number of *F. graminearum* identified in RNA seq is around 14,000, so the total differential expressed genes between WT-OE:FGM4 and WT is relatively small, only 7% of total genes. We now provided the gene lists in Supplementary Table 5. Researcher may find candidate genes responsible for processing sILeuol among these DEGs.

Reviewer #2-8 “*The data in Fig.S7 does not contain gDNA controls. Also, do the primer combinations span introns?*”

Answer: The primers for FGM4 span intron, others are not. We provided gDNA controls for FGM5 knockout and FgNRPS5 C-ter deletion mutants. All the RT-PCR primers information is provided in Supplemental Table 2d.

Reviewer #2-9 “*Overall, the elucidation of the chemical compound that the authors attribute to the gene cluster is not strong enough to provide evidence that the identified tripeptide really is the final product made by cluster 54 and therefore I do not recommend publication in its current form.*”

Answer: In the revision, we provided more evidence to prove that sILeuol is the final product made by Fg3_54 cluster, including:

- 1) Construction of N-ter deletion of first A module of FgNRPS5, product analysis showed reduced accumulation of N1 band (sILeuol), consistent with that first A module often facilitate nonribosomal protein biosynthesis initiation.
- 2) Identified the intermediate product of Fg3_54 in FGM9 deletion mutant with OE:FGM4.
- 3) Adding the speculated complete product of FG3_54 sILeuol to *F. graminearum*, sILeuol can be detected, presumably as a processed product.
- 4) Identified m/z 318.24 metabolite in *F. graminearum* infected wheat spike, showed that sILeuol is a real native player in *Fusarium*-wheat infestation.

Because the biosynthesis pathway for sILeuol is complicated, fully elucidation of the biosynthesis pathway will require a lot more work, which is obviously beyond the scope of this work. We will do further analysis to dissect biosynthesis of sILeuol.

Response to review:

We appreciate your reviews and comments. Accordingly, we performed new experiments, including RNA sequencing of tripeptide-treated wheat samples and product analysis of FgNRPS5 N-ter deletion and FGM9 deletion mutants in the background of FGM4 overexpression. More importantly, we included more text in which we speculate about what the original product of FgNRPS5 might be, based on non-ribosomal codes in adenylation domains, and then chemically synthesized the suspected heptapeptide (i.e. speculated original product). When the heptapeptide was incubated with *F. graminearum*, sLeuol tripeptide was detected in the culture, which supports the idea that sLeuol might be a processed product of FgNRPS5. With these new data and analyses, we added a new Figure (Fig. 6), modified three Figures (Fig. 2,3,8), added nine more supplementary figures and two more supplementary tables, to address your concerns. Here we submit a substantially revised manuscript (a clean version and a version with changes highlighted) with point-to-point responses to reviewer comments. (Point-by-point response in blue)

Reviewers' comments:

Reviewer #1 (Remarks to the Author):

The manuscript by Dr Tang and co-workers describes the identification of a tri-peptide from *Fusarium graminearum*, which is involved in cell-to-cell movement in wheat. The study was initiated by a previously published observation that a gene cluster comprising NRPS5 and NRPS9 (FG3_54) was active in a coleoptile infection system. Furthermore, the group has previously shown that deletion of several genes in the cluster results in decreased virulence. In the present study this was further investigated by individual deletion of all members of the gene cluster, which also led to decreased virulence. Infection was then monitored using a previously developed wheat coleoptile infection system, where the mutants showed shorter lesions than the wild type (app. 30-60 % reduction). A fluorescent marker was also used to monitor the spread of wt and deletion mutants in the coleoptile infection system, which revealed that the mutant strains failed to penetrate neighboring cells. Staining of wheat cells showed that callose is formed as a defense reaction against the deletion mutants, but not against wt. A local transcription factor, FGM4, was identified through a yeast two-hybrid system. The TF has similarity to TFs from other NRPS gene clusters (this can be further highlighted). GFP fusion used to show that the protein is located to the nucleus. OE:FGM4 increase transcription of gene cluster (but apparently NRPS5 is less expressed than the others). To identify the metabolites produced by the gene cluster HPLC-UV was first used by this was unsuccessful Instead TLC with ninhydrin staining was used and a compound was isolated and elucidated by NMR. The compound sLeuol consist of three amino acids (D-seryl-D-leucyl-L-Leucinol) and addition of the compound restored virulence in some deletion mutants (partial) and also enabled *F. oxysporum* to infect wheat.

The manuscript describes a very extensive study of the gene cluster and as the authors also state, this is one of only few examples of compounds from *F. graminearum* that has a proven role for pathogenicity. In many ways this is a very impressive work, but there are however some issues that must be addressed.

Answer: We appreciate reviewer's recognition of our work.

One of the most important things is the biosynthetic pathway. As NRPS5 contains eight modules

and NRPS9 one module, it seems unlikely that the product should be a three amino acid compound. This discrepancy is only vaguely addressed in the manuscript and needs more attention. The authors show that premature deletions of NRPS5 at the C-terminal result in reduced virulence. To complete the examination deletion of the N terminal domains of NRPS5 could provide clues to the role of NRPS5. It could be that the first five modules are not required for sLeuol (which would explain the uneven transcription of the gene seen in figure 3d).

Answer: We agree with reviewer 1 (also reviewer 2) that the biosynthetic pathway is a key concern. FgNRPS5 comprises “A”-6×“C-A-PCP-E”-“C-A-PCP-R”, which presumably would produce a heptapeptide or octapeptide, but the active metabolite we identified is a tripeptide. How to explain this? We hypothesize that the tripeptide sLeuol might be processed from the original heptapeptide. In previous submission, we only briefly mentioned this possibility. In this revision, according to the reviewer’s suggestion, we constructed a N-ter deletion mutant of FgNRPS5, lacking the first “A” domain (Supplementary Fig 13a) in FGM4 overexpressing strain. Thin layer chromatography analysis of extracts showed reduced accumulation of the N1 product (Supplementary Fig. 13f). The N-ter deletion mutant strain also showed reduced virulence in wheat coleoptile infection (Supplementary Fig. 13), consistent with a role for the N-ter first module of NRPS in increasing the initiation efficiency of peptide biosynthesis. Given that C-ter deletion mutants cannot produce the tripeptide at all (Supplementary Fig. 10 and 12), this additional piece of evidence (shown in Supplementary Fig. 13 and text lines 352-362 in this revision) supports the hypothesis that sLeuol corresponds to the C-ter modules of FgNRPS5.

A more thorough description of the NRPSs is also needed (prediction of substrate, Stachelhaus codes..)

Answer: We agree and therefore provide a more thorough description of FgNRPS5 and extracted the nonribosomal code (also called Stachelhaus code) in A domains of FgNRPS5, aligned them with known codes to help substrate prediction. This analysis resulted in an additional Supplementary Figure, Suppl. Fig. 19. We discussed these results in lines 550-569: “The RNA-seq reads of FgNRPS5 cover a 35 Kb region (Fig. 3d), suggesting that the full-length protein is an “A”-6 ×“C-A-PCP-E”-“C-A-PCP-R” multimodular NRPS (Fig. 1a). The presence of E (epimerization) domains in FgNRPS5 suggests that the incorporated amino acids could change their conformation from L to D. The presence of a R (reductase) domain suggests that product(s) are released by reduction. Some R domains are capable of a four-electron reduction, in which the peptidyl thioester attached to the PCP is reduced first to an aldehyde and then to an alcohol⁶⁵. The FgNRPS5 C-terminal R domain resembles a Lys2-type reductase, which produces an aldehyde from a carboxylic acid, as in *Aspergillus flavus* LnaA and LnbA, which have an A-T-R domain arrangement⁶⁶. Therefore FgNRPS5 might produce a heptapeptide composed of six D-amino acid residues followed by a L-amino acid-derived aldehyde. Given that FGM9 dehydrogenase might reduce an aldehyde to an alcohol, it is possible that the complete final product of NRPS5 and FGM9 is a heptapeptide composed of six D-amino acid residues followed by a L-amino-alcohol. Therefore the tripeptide sLeuol, which has two D-amino acid residues followed by an L-amino-alcohol, might be a processed product corresponding to the last three modules of FgNRPS5.

“NRPS A domains select monomers to incorporate, and the amino acid residues occupying the ten key positions that are relevant for substrate specificity within A domains are referred to as

the nonribosomal code or Stachelhaus code67. However nonribosomal codes from FgNRPS5 showed low similarity to known codes from bacterial and fungal NRPS (Supplemental Fig. 19), therefore it is difficult to predict the substrates of FgNRPS5.”

We further moved forward, because we found that: “Interestingly, the nonribosomal codes of FgNRPS5 A2, A4 are similar to that of A6, and those of A3, A5 are similar to those of A7 and A8 (Supplementary Fig. 19d). If sLeuol does correspond to the last three modules of NRPS as we inferred above, the substrates of A6 and A7 will be serine and leucine respectively. Then based on the nonribosomal codes similarity, it can be speculated that the substrates for A2, A4 might be serine, and the substrates for A3 and A5 might be leucine. Given that E domains are also present in modules 2-7, the complete product of FgNRPS5 and FGM9 is predicted to be D-Seryl-D-Leucyl-D-Seryl-D-Leucyl-D-Seryl-D-Leucyl-L-Leucinol (slslslLeuol). In a preliminary test of this prediction, we applied chemically synthesized slslslLeuol to in vitro cultured *F. graminearum*. The tripeptide sLeuol (m/z 318.24) was identified in the medium, as well as the heptapeptide slslslLeuol (m/z 718.47) (Supplementary Fig. 20 a-d). In contrast, sLeuol (m/z 318.24) was not identified in the medium of budding yeast or *Foc* TR4 incubated with slslslLeuol. sLeuol was also identified in the medium of Δ fg3_54 *F. graminearum* incubated with slslslLeuol (Supplementary Fig. 20 c,d). Therefore, it is possible that tripeptide sLeuol is the processed product of slslslLeuol, and that processing was achieved by genes not located in the FG3_54 cluster. Furthermore, the addition of the heptapeptide slslslLeuol also restored virulence of Δ fg3_54 on wheat coleoptiles (Supplementary Fig. 20e), and caused rapid cell death of tobacco leaf cells, as indicated by large areas of water soaking and loss of chlorophyll fluorescence (Supplementary Fig. 16b). We hypothesize that the heptapeptide might be highly toxic, and therefore processed immediately after synthesis, which might explain why we couldn’t detect the heptapeptide in either FGM4-expressing *F. graminearum* or in *F. graminearum* infected wheat tissues.” (line 570-592 in the revised manuscript)

Identification of the compound(s) produced by the gene cluster is only done partially. First the extracts of Wt, KO and OE were analyzed by HPLC-UV (unsuccessfully) and then by TLC. Here ninhydrin staining helped to identify one major band, which was subsequently isolated and elucidated (sLeuol). When inspecting the gel there seems to be another strong band just above sLeuol and one or two more weaker bands. These bands were apparently not further examined, although these could be the potential end product or intermediates of the metabolic pathway. These bands should also be isolated and examined by LC-MS. Furthermore, the raw extracts from the different mutant strains should be analyzed by high resolution LC-MS, which could reveal other compounds which only are expressed by the OE mutant in vitro (e.g. using XCMS: generate heat map + PCA).

Answer: We agree that identification of the gene cluster products was only partial. We made more efforts to identify the weaker band (N2) and the product(s) of Δ fgm9 mutant, resulting in an additional Supplementary Fig. 11, and stated in results, lines 344-351, 363-368, as below:

“To further test whether sLeuol is a direct product of FG3_54 cluster NRPSs, we performed chromatography analysis of various mutant strains with FGM4 overexpression. The ninhydrin positive bands (N1 and N2) were detected in FGM4 overexpression strains of Δ fgm1, Δ fgm2, Δ fgm3 and Δ fgm5, but not in the NRPS5 or NRPS9 deletion mutants (Supplementary Fig. 10). Furthermore, the extracted ion chromatography shows that N1 ([M+H]⁺: 318.24) and N2

([M+H]⁺: 302.24) were absent in the extracts of Δ nrps5, Δ nrps9 and Δ fg3_54 mutants when FGM4 was constitutively expressed (Supplementary Fig. 11d, e). These results support that NRPS5 and NRPS9 are required for the production of the ninhydrin positive metabolites.”

“Furthermore, in FGM4 overexpression strains of Δ fgm9, an additional ninhydrin positive band, N3, was present (Supplementary Fig. 11a). Q-TOF LC-MS analysis of recovered N3 metabolites showed that its m/z ([M+H]⁺) is 316.22 (Supplementary Fig. 14a, b), slightly smaller than sLeuol (m/z 318.24). The N3 metabolite was not detected in other mutant strains with FGM4 expression (Supplementary Fig. 14c,d). FGM9 showed similarity to a short-chain alcohol dehydrogenase, in line with the idea that FGM9 may modify FgNRPS5 original products, probably to achieve leucinol at the C-ter.”

In summary, the added experiments support the idea that the tripeptide sLeuol can be the final product of FgNRPS5 and FG3_54 cluster, so we added this part of results (Supplementary Fig. 20, and 16b partial) in the revised manuscript. Although the speculations of substrate prediction are purely based on signature a.a. similarity, **these experimental results set up a link between sLeuol and a likely FgNRPS5 product**, which could relieve some of our concerns on sLeuol biosynthesis pathway. However it also suggests that genes not included in this cluster might be involved in processing sLeuol. It would take another whole paper to elucidate the complete biosynthesis pathway. The complete complex biosynthesis pathway of this simple molecule is really beyond the scope of this manuscript, which is identification of a novel simple compound as an effector directly functions in such an important crop disease, wheat Fusarium head blight, and initial elucidation of its mechanism in pathogenesis.

The observation, that sLeuol affects defense response in wheat, is very interesting. But the subsequent qRT-PCR of selected defense genes (WT vs KO) did not reveal expression differences. Here RNAseq or similar would have been more useful.

Answer: We agree, and therefore performed RNA seq to compare wheat gene responses in coleoptiles infected with wild-type, mutant *F. graminearum* lacking FG3_54 cluster, or treated with sLeuol. With these new data from RNA seq, we now identified 791 wheat genes as candidates of wheat defense-related genes whose expression is suppressed by sLeuol (Supplementary Table 6), and thereby enhanced our results, in lines 411-433. For example, “GO enrichment analysis further showed that genes with 1,3- β -D-glucan synthase activity (GO:0003843) are significantly enriched in the 791 genes. These results are in line with the idea that sLeuol might suppress callose (i.e. 1,3- β -D-glucan) synthesis to facilitate *F. graminearum* invasion in wheat.”

Minor issues

Describe which promoter was used for overexpression of the local transcription factor.

Answer: The FGM4 transcription regulator was driven by the *F. graminearum* constitutive expression promoter EF1-A (FGSG_08811, translation elongation factor 1 alpha), as used in Zhang et al., 2016 PLoS Pathogens 12(3): e1005485. In this revised manuscript, we made this clear in methods part, under the subtitle “Fungal Transformation”.

How was secondary metabolites extracted from in vitro cultures prior to HPLC-UV and TLC

analyses?

Answer: As shown in previous version Supplemental Figure 8a (=Supplementary Fig. 9a in this revision), the medium fraction of culture was extracted by ethyl acetate, followed by evaporation, then resolved in 90% methanol:n-hexane(1:1 v:v), the methanol extracts were then evaporated again. The residues were dissolved in water with 1% DMSO for activity analysis or in methanol for LC and TLC analyses. We added this to the legend for Supplementary Fig. 9a.

Reviewer #2 (Remarks to the Author):

The manuscript NCOMMS-16-28154 entitled “A nonribosomal Peptide facilitates Cell-to-Cell Invasion of *Fusarium graminearum* in Wheat” by Jia et al., dissect the influence of the deletion of a putative two non-ribosomal peptide synthetase (NRPS)-encoding secondary metabolite gene cluster on *F. graminearum* virulence on wheat. Previous studies have identified this cluster to be only present in a subset of *Fusarium* species and expression analysis showed that cluster gene members are expressed during specific stages of infection. In this study the authors leverage the existing knowledge to elucidate in more detail the influence of whole cluster deletion mutants on pathogenicity and virulence on wheat and can show that deletion of the whole cluster as well as several single gene deletion mutants (some of them previously published) show reduced ability to infect coleoptiles and spikes as well as reduced cell-to-cell penetration abilities. The authors ectopically over-express a putative transcription factor (fgm4) and use this mutant to elucidate the putative secondary metabolite produced by the cluster

The concept of effectors or secondary metabolites to influence *Fusarium* virulence has previously been described for several *Fusarium* species and does in itself not represent a major advancement to the field of plant pathology. The data that build the basis for analyzing this specific cluster have been previously published. Some of the single cluster gene deletion mutants have already been shown to have reduced virulence (e.g. the ABC transporter encoding fgm5); some of them with contradicting results (deletion of nrps9 was previously reported to have WT-like virulence).

Answer: We appreciate reviewer's recognition the novelty of identification of a specific compound product of FG3_54 cluster as a virulence factor of *F. graminearum*. We like to add the following points to argue the significance of our discovery.

1) In the field of plant pathology, particularly for important *Fusarium* pathogens, identified effectors are still limited. Identified effectors include 'secreted in xylem'(SIX) proteins of *Fusarium oxysporum*, secreted lipase FGL1, and trichothecene mycotoxin DON of *Fusarium graminearum*. Here we describe sLeuol, as a simple linear tripeptide, which is distinct from all known virulence effectors. Furthermore, we showed in Supplementary Fig. 15 and by RNA seq data (Supplementary Table 6) that sLeuol is not just a toxin metabolite and suppressing expression of only a limited number of wheat genes. Therefore, we believe sLeuol represents a new type peptide/secondary metabolite effector.

2) In our 2012 Plant Cell paper (Zhang et al.) we detected the expression of the eight member genes in this cluster during infection of wheat coleoptiles in a whole-genome microarray hybridization with a very special laser microdissection sampling, and reported on the phenotypes of knockout mutants of two member genes. FGM5 (ABC transporter) and FGM2 (a putative polysaccharide deacetylase) both had reduced virulence in wheat spike infection, whereas the

knockout of *FgNRPS9* didn't reduce *F. graminearum* virulence in wheat spike infection. The role of putative core nonribosomal peptide synthase enzyme FgNRPS5 wasn't identified. FgNRPS5 was even thought to be a pseudogene based on the presence of stop codons and frameshifts in the predicted sequence (our RNA seq results shown in Figure 3d eliminated these concerns and demonstrated that *FgNRPS5* can produce a 35.4k nts huge transcript encoding a 11299 a.a. ORF). So, until this work, it was still not clear whether the whole cluster functions in wheat spike infection (i.e. wheat head blight infection), or if just some individual genes contribute to virulence independent of the cluster product. Here we provide solid evidence that the cluster product is a virulence factor. It is probably not that significant to identify a couple of more genes that are required for full virulence in *F. graminearum*, given that more than 200 genes have been reported as doing so (see PHI base: pathogen host interactions database, <http://www.phi-base.org/>). But it is significant to identify a molecule that directly functions in native pathogen-host interactions.

3) Coleoptiles and spikes are different organs of wheat. It is thought fungal pathogens may have core effectors that function when they infect any organs, but that they might also have organ-specific effectors that only function when they infect a particular organ. So it is understandable that a knockout of *FgNRPS9* didn't reduce *F. graminearum* virulence in wheat spike infection, but reduced virulence in wheat coleoptile infection.

The novelty of this study lies in the proposed identification of the chemical compound produced by cluster 54. The authors ectopically constitutively express a putative transcription factor, *fgm4*, and isolate a compound that they identify as a tripeptide D-seryl-D-leucyl-L-leucinol. Addition of extracts from OE::*fgm4* and synthetically obtained peptide restore virulence of the cluster knock-out mutant and even enable a non-wheat pathogenic *Fusarium oxysporum* strain to infect wheat (Is this true for only the synthetically obtained peptide or also extracts from OE::*fgm4*?).

Answer: Thanks for the comment. We performed experiment to test the effect of *F. graminearum* medium extracts on FocTR4, and now made it clear in this revised manuscript, that medium extracts of FGM4-expressing *F. graminearum* and synthetic sLeuol both restored virulence of the cluster knockout mutant (Fig. 4a) and enabled the FocTR4 strain to invade wheat (Fig. 6).

However, the study fails to explain how a tripeptide could be made by a gene cluster that harbors two NRPS, one with one A domain, the other containing eight A domains. It is unprecedented that large NRPS like the ones studied here would only make a small peptide, without explaining any cleaving of a larger metabolite into the identified tripeptide. Furthermore, the presence of epimerase domains in NRPS5 would suggest that the incorporated amino acids would change their L/D confirmation, unless some would be non-functional. In order to provide more inside into the biosynthesis of the compound produced from cluster 54 more analytical efforts are needed. Single deletion strains of every cluster gene putatively involved in modification of the initial chemical scaffold are desirable to explain how the non-proteinogenic leucinol is being made, which gene could be involved in the reduction of the carboxy group.

Answer: This point is similar to reviewer#1. Thanks for the suggestions. Following these suggestions along with reviewer 1's, we included results from new experiments in the revision, and showed that the tripeptide sLeuol, which has two D-amino acid residues followed by an L-amino-alcohol, might be a processed product corresponding to the last three modules of FgNRPS5. Please see answers in page 1-3 for details.

The authors ectopically express *fgm4* and perform RNA-seq experiments to show expression of cluster 54 is specific. The Southern analysis of the OE::*fgm4* strains seem to be missing and should be added. Why was the transcription factor not over-expressed by a promoter replacement at the original gene locus? It would be more convincing if the scale in Fig. 4d would be uniform. Also more information on Fig S6e is needed to indicate where the data was obtained from. Since the authors performed RNA-seq analysis, what is the change in global gene expression of OE::*fgm4* to WT? The data in Fig.S7 does not contain gDNA controls. Also, do the primer combinations span introns?

Answer: Thanks for pointing this out. Southern blot verification of OE::FGM4 strains, using an intron-less FGM4 genomic DNA fragment as probe, is provided in Supplementary Fig. 8b in this revision. We agree that FGM4 overexpression can also be achieved by promoter replacement at the original gene locus. But in the split marker (Hyg resistance) recombination procedure (Catlett et al., 2003) we used, it will also introduce the hygromycin resistance gene (2 kb) into the cluster, which brings concerns in causing chromosomal architecture changes in the cluster region. Because *FGM4* and *FGM5* are located together but with opposite transcriptional directions (Fig. 1a), the introduced fragment will be just upstream of the adjacent *FGM5* gene promoter, and so it was possible that the transcription of *FGM5* might be affected. Therefore we constructed transgenic strains carrying the *FGM4* gene driven by a constitutive promoter, rather than promoter replacement at the original gene locus.

Fig. 4d confocal microscopic images are at the same scale. We suspect the reviewer is mentioning Figure 3d, i.e. the plot of RNA seq reads. In this revision, we set the scale of the RNA seq plot to the same.

For Supplementary Fig. 6 (renumbered as Supplementary Fig. 7e), the list of secondary metabolite biosynthesis core enzymes were obtained from the following reference:

Connolly, L. R., K. M. Smith, and M. Freitag. 2013. 'The *Fusarium graminearum* histone H3 K27 methyltransferase KMT6 regulates development and expression of secondary metabolite gene clusters', *PLoS Genet*, 9: e1003916.

The expression data for these genes were obtained from the following papers and subjected to global normalization:

In vitro: Seong, K. Y., Zhao, X., Xu, J. R., Guldener, U. & Kistler, H. C. Conidial germination in the filamentous fungus *Fusarium graminearum*. *Fungal Genet. Biol.* 45, 389–399 (2008).

In wheat coleoptile: Zhang, X. W. et al. In planta stage-specific fungal gene profiling elucidates the molecular strategies of *Fusarium graminearum* growing inside wheat coleoptiles. *Plant Cell* 24, 5159–5176 (2012).

In maize stalk: Zhang, Y. et al. Cellular tracking and gene profiling of *Fusarium graminearum* during maize stalk rot disease development elucidates its strategies in confronting phosphorus limitation in the host apoplast. *PLoS Pathog.* 12, e1005485 (2016).

We added these references in the legend of Supplementary Fig. 7 in this revised manuscript.

Based on the RNA-seq comparison of *F. graminearum* strains WT-OE:FGM4 and WT, 465 genes significantly increased expression levels and 530 genes reduced expression levels in WT-OE:FGM4, relative to expression levels in WT. The total gene number of *F. graminearum* identified in RNA seq is around 14,000, so the total differentially expressed genes between WT-OE:FGM4 and WT is relatively small, only 7% of total genes. We now provide the gene lists

in Supplementary Table 5.

The primers for FGM4 span an intron, but the others do not. We provided gDNA controls for FGM5 knockout and FgNRPS5 C-ter deletion mutants. All RT-PCR primer information is provided in Supplemental Table 2d.

Overall, the elucidation of the chemical compound that the authors attribute to the gene cluster is not strong enough to provide evidence that the identified tripeptide really is the final product made by cluster 54 and therefore I do not recommend publication in its current form.

Answer: In the revision, we provided more evidence to support that sILeuol is the final product made by the Fg3_54 cluster, including:

- 1) Construction of N-ter deletion of the first A module of FgNRPS5. Supplementary Fig. 13 (newly added) product analysis showed reduced accumulation of the N1 band (sILeuol), consistent with the idea that the first A module often facilitates nonribosomal protein biosynthesis initiation.
- 2) Identification of the intermediate product of FG3_54 in the FGM9 deletion mutant with OE:FGM4 (Supplementary Fig. 14).
- 3) Adding the speculated complete product of FG3_54 sIsIsILeuol to *F. graminearum*. Supplementary Fig. 20 showed that sILeuol can be detected, presumably as a processed product.
- 4) Identified m/z 318.24 metabolite in *F. graminearum* infected wheat spikes (Figure 6 middle panel), showed that sILeuol is a bona fide native player in Fusarium-wheat interaction.

Because the biosynthesis pathway for sILeuol is complicated, fully elucidation of the biosynthesis pathway will require a lot more work, which is beyond the scope of this work. We will do further analysis to dissect biosynthesis of sILeuol in the future. But with these newly added pieces of evidence, I hope reviewer will agree that sILeuol is linked with FgNRPS5, and can be a processed product of FgNRPS5.

Minor comment:

The Southern strategies in Fig S2 are unconventional. Why did the authors not use a flank as probe that bind in both, the WT and the mutants. Without a schematic overview where the restriction enzymes are located the strategies are hard to follow. It also appears that all three whole cluster gene deletion mutants have different patterns. More details are needed to assure what parts of the cluster are deleted. And which of these mutants was used to create OE::fgm4?

Answer: In the revision, we revised the presentation of Southern blotting in Supplementary Figure 2 and 8. Schematic illustrations of mutant generation and verification are provided in Supplementary Figure 2, 3, 12 and 13. Usually, -1 mutants were used for generation FGM4 OE lines.

Other modifications we made in this revision but not mentioned above include:

Adding an image in Figure 2d (middle panel) representing FG3_54 deletion mutant at 3dpi, showing that mutant hyphae reached the first layer of wheat cells, but not further;

Modification of Figure 8b to include new results of virulence assays of *Fusarium avenaceum* and *Fusarium poae*, adding strength to the correlation between ability to invade coleoptiles and the presence of FG3_54 cluster;

Replacement with an image in Supplementary figure 6b showing FG3_54 deletion mutant caused strong callose deposition that sealed mutant hyphae in the first layer of wheat cells;

New images added in Supplementary figures 15a, 16, 17;

New Supplementary figure 18 showing that the addition of siLeuol can also enable *Fusarium poae* to invade wheat coleoptiles, in addition to Foc TR4. This might further broaden interest.

Reviewers' comments:

Reviewer #2 (Remarks to the Author):

The revised version of the manuscript NCOMMS-16-28154B by Jia et al., unfortunately does not answer the concerns raised regarding how a 8 A-domain and 1-A domain NRPS-containing cluster produces a tripeptide. There is no putative peptidase in the cluster that could perform the shortening of a longer peptide and among the other genes identified to be regulated by Fgm4, no peptidase was identified either. It is unprecedented that an NRPS with this composition produces an unexpectedly small compound. Unless this issue has been resolved with more chemical evidence, I do not recommend publication in Nature Communications.

Reviewers' comments on the last version manuscript NCOMMS-16-28154B:

Reviewer #2 (Remarks to the Author):

The revised version of the manuscript NCOMMS-16-28154B by Jia et al., unfortunately does not answer the concerns raised regarding how a 8 A-domain and 1-A domain NRPS-containing cluster produces a tripeptide. There is no putative peptidase in the cluster that could perform the shortening of a longer peptide and among the other genes identified to be regulated by Fgm4, no peptidase was identified either. It is unprecedented that an NRPS with this composition produces an unexpectedly small compound. Unless this issue has been resolved with more chemical evidence, I do not recommend publication in Nature Communications.

Response to the reviewer:

We appreciate the reviews and actually agree with the reviewer's comments. In the past year, we re-analyzed metabolite production of cluster induction strains after optimizing the culture and extraction protocols and finally identified the complete product of FgNRPS5 and FgNRPS9 encoded by *fg3_54* cluster. We discovered that the real product of NRPS5 and NRPS9 encoded in the *fg3_54* locus is an unprecedented octapeptide, [γ -amino butyl acid (GABA)]₁-[L-Ala]₂-[D-*allo*-Ile]₃-[D-Ser]₄-[D-Val]₅-[D-Ser]₆-[D-Leu]₇-[L-Leuol]₈ (abbr. GABA-AisvslLeuol). The composition and residue organization of this octapeptide (named fusaoctaxin A) are well consistent with the catalytic logic of NRPS5 and NRPS9, which are composed of a loading module, i.e., **M1**-(A₁-T₁), and seven extension modules, i.e., **M2**-(A_{2a}-C₂-A_{2b}-T₂)-**M3**-(C₃-A₃-T₃-E₃)-**M4**-(C₄-A₄-T₄-E₄)-**M5**-(C₅-A₅-T₅-E₅)-**M6**-(C₆-A₆-T₆-E₆)-**M7**-(C₇-A₇-T₇-E₇)-**M8**-(C₈-A₈-T₈-R). This assembly line likely utilizes GABA as a starter unit and sequentially incorporates seven extender units composed of the residues L-Ala, L-*allo*-Ile, L-Ser, L-Val and L-Leu. During the process, each of the residues that are tethered on modules (**M3**-**M7**) containing an E domain can undergo an epimerization reaction to produce a D-configuration before the transpeptidation reaction occurs. The elongation of the peptidyl chain might be terminated by module **M8**-mediated L-Leu incorporation, followed by R domain-catalyzed 4 electron reduction to release the resulting octapeptide from the assembly line as an alcohol. Previously we couldn't identify the octapeptide from fungal extracts because the octapeptide cannot be extracted with ethyl acetate. Now we switched to resin extraction.

Clearly, the previously identified tripeptide (slLeuol) is identical to the three C-terminal residues of fusaoctaxin A, i.e., D-Ser, D-Leu, and L-Leuol, indicating that it is a cleaved product of the newly identified octapeptide. This octapeptide is capable of restoring the cell-to-cell invasion ability of *fg3_54* mutants in wheat (Fig. 5) with a much lower dosage than that of the tripeptide. Also, this octapeptide can convert non-wheat pathogens *Foc tr4* and *F. poae* to wheat pathogens. In addition, fusaoctaxin A was detected more abundant than slLeuol in the infected wheat tissues. Therefore, the newly identified octapeptide, instead of the previously identified tripeptide, is the major virulence factor that confers the hyphal ability of *F. graminearum* for cell-to-cell invasion in wheat.

The current revised manuscript keeps the contents of the previous version concerning the

correlation of the cluster *fg3_54* and its associated biosynthetic genes with the cell-to-cell invasion ability of *F. graminearum* in wheat and the expression of *fg3_54*-related genes *in vitro* by activating *fgm4* for virulence factor production. Significant changes focus on the characterization of the octapeptide fusaoctaxin A as the unprecedented virulence factor of *F. graminearum* that confers the hyphal ability of cell-to-cell invasion in wheat, including structural elucidation, proposed biosynthetic pathway, relevance to the invasion ability of *F. graminearum* and related strains, phylogenetic analysis, and host cell responses and effects on gene expression in *F. graminearum* that allow to propose the mechanisms of action. Because the tripeptide sLeuol appears to be less active and important in invasion, and particularly due to the page limitation, tripeptide-related studies in the previous manuscript are not included in this revision and could be presented in the future after we characterize the process of how fusaoctaxin A undergoes specific cleavage to generate this tripeptide.

Other modifications we made in this revision but not mentioned above include:

Addition of Fig. 5d showing that fusaoctaxin A can also complement mutant deficiency in spike infection of wheat in addition to previously reported coleoptile infection.

Addition of Fig 5h showing that fusaoctaxin A can also enhance virulence of wild-type *F. graminearum* on wheat.

Modification of Figure 6 to include new results of virulence assays of *Fusarium avenaceum* and *Fusarium poae*, adding strength to the correlation between ability to invade coleoptiles and the presence of FG3_54 cluster;

Modification of Figure 7, performed new experiments with the octapeptide to show its effects on wheat cells without complication from fungal growth.

Figure Modification Tracking:

Previous Fig. 1 The revised Fig. 1 No change except omitting the modular prediction of NRPS5 here, this part has been moved to Fig. 4

Previous Fig. 2 The revised Fig. 2 No change in contents, only reorganizing images for clarity

Previous Fig. 3 The revised Fig. 3 No change

Previous Fig. 4 The revised Fig. 5a The previous fractioning and TLC purifying ninhydrin-positive metabolites containing the tripeptide (previous Fig. 4, along with Supple Fig. 9,10) has been changed to HPLC purification and MS analysis of the octapeptide (reduced to Fig. 4a). The purified metabolite activity in restoring cell-to-cell invasion (old Fig. 4c) is replaced by the newly performed activity restoring assay with the octapeptide (Fig. 5a pur.);

Previous Fig. 5 The revised Fig. 4 . Identification and structure elucidation of the previous tripeptide has been replaced by the results of octapeptide identification based on new experiments including NMR (Fig. 4b, Supple Figs. 8, 9). Based on these new results, we added a new presentation of deduced octapeptide biosynthesis pathway (Fig. 4c), which showing the consistency between the composition and residue organization of this octapeptide and the catalytic logic of NRPS5 and NRPS9;

Previous Fig. 6 The revised Fig. 4d The detection of tripeptide in infected plant

tissued has been replaced by new detection results of the octapeptide. The Q-TOF LC-MS results of metabolite extracted from infected plant tissues were reanalyzed, which reveals the presence of the octapeptide (773.51 m/z) (Fig. 4d), therefore the previous tripeptide (318.2 m/z) detection (previous Fig. 6) is omitted;

Previous Fig. 7 The revised Fig. 5d-f We re-performed the experiments of restoring fg3_54 deletion mutants infection ability and enabling non-wheat pathogens Foc TR4 and F. poae to invade wheat coleoptile with the octapeptide (Fig. 5d,e,f) to replace previous tripeptide results (old Fig. 7, old Fig. S18), and results show that threshold amount of peptide to restore wheat coleoptile infection is lower for the octapeptide (3 nmol per seedling) than for the previous tripeptide (~30 nmol per seedling). In addition to wheat seedling coleoptile infection assay above, we performed new experiments of wheat spike infection and added the results showing the octapeptide can also restore spike infection ability of fg3_54 deletion mutants (Fig. 5d), which is more relevant to fusarium head blight disease. In addition to mutant rescue experiments above, we performed new experiments showing the octapeptide can also increase wheat coleoptile infection ability of wild-type F. graminearum in a dosage-dependent manner (Fig. 5h). In searching for host cellular responses to the identified peptide, we not only re-performed the cell plasmolysis assay and chloroplast-focused microscopy in wheat coleoptile cells, cytotoxicity assays in pollen tube cells, Nicotiana benthamiana leaves and mammal cell lines, with the octapeptide (Fig. 7a-d, Supple Fig. 11), instead of previous tripeptide (previous Fig. 7e, Supple Figs. 15, 16), but also performed new cell viability assay (Acridine Orange/Ethidium Bromide staining) on the octapeptide treated coleoptiles and showed that as early as 5 h after treatment the octapeptide slightly alters chloroplast subcellular localization pattern before it causes any changes in host cell viability or plasma membrane integrity (Fig. 7a-d). New experiment of cell plasmodesmal permeability assay was performed and the results showed that, as early as 3 h after treatment, the octapeptide can inhibit the wheat cell plasmodesmal closure, which is an early defense response against non-adapted fungus (Fig. 7e). These new results provide more clues of the octapeptide working mechanism.

Previous Fig. 8 The revised Fig. 6a-c The tree is not changes, b tested more fungal species in assay, supporting the correlation stronger. Finally we add a new diagram to delineate how the octapeptide facilitates wheat infection at cellular and molecular levels (Fig. 9).

Previous Supple Fig. 1 The revised Supple Fig. 1 The heatmap is not changed, the images to show representative fungal stages are added to ease understanding.

Previous Supple Fig. 2, 3 The revised Supple Fig. 2 Contents are combined, restriction map has been added, modified to make it clearer

Previous Supple Fig. 4 The revised Supple Fig. 3 No change excepted omitting the TaPR3 nad TaICS RT-PCR results which now have been included in RNA sequencing results Supplementary Table 5.

Previous Supple Fig. 5 The revised Supple Fig. 4 No change

Previous Supple Fig. 6 The revised Supple Fig. 5 Contents are not changed, just picked higher quality images.

Previous Supple Fig. 7 The revised Supple Fig. 6 No change except for the heatmaps are organized based on secondary metabolite categories and pathways, and 7c is moved to next Fig.

Previous Supple Fig. 8 The revised Supple Fig. 7 No change except for adding the RT-PCR results previously in Fig. 7c.

Previous Supple Fig. 9, 10, 11, 12, 13,14 Most deleted because we changed the purification protocol, the detection of tripeptide has been replaced by detection of octapeptide shown in the revised Fig. 4a

The revised Supple Fig. 8, 9 new data of octapeptide structure

Previous Supple Fig. 15 The revised Supple Fig. 10 The tripeptide assays have been replaced by the octapeptide assays

Previous Supple Fig. 16 The revised Supple Fig. 11 The tripeptide assays have been replaced by the octapeptide assays

Previous Supple Fig. 17 The tripeptide assays have been replaced by the octapeptide assays shown in Fig 6e

Previous Supple Fig. 18 The tripeptide assays have been replaced by the octapeptide assays shown in Fig 6d

The revised Supple Fig. 12 The new RNA seq analysis of octapeptide treated samples.

Previous Supple Fig. 19 The revised Supple Fig. 13 The a.a.selection info is added based on octapeptide.

Reviewers' comments:

Reviewer #3 (Remarks to the Author):

I've read the manuscript as well as the response to reviewers (listed as reviewer #2 in my version) - I think the authors have done an excellent job of addressing these original concerns (which were well founded, no doubt) - this has now strengthened the manuscript given that the octapeptide has now been identified and characterised in a comprehensive manner.

I do not have any real concerns to report expect to say that the references for the NRPS section are very old and newer ones would add more value for the reader, given how much has been added to our knowledge of this field over the last 10-15 years.

Reviewer #4 (Remarks to the Author):

The paper by Lei-Jie et al. identifies a new virulence factor of *F. graminearum* infecting wheat. The paper is very original and provides unprecedented findings about the role of a non ribosomal octapeptide (FA) in supporting fungal spreading within host tissues.

Methodological tools of fungal genetics, transcriptomic analyses, cellular microscopy, peptide characterization and synthesis are very appropriated and advanced.

Major critical points that should be clarified:

The title deals with a general role of FA in "cell-to-cell invasion of *F. graminearum* wheat". *F. graminearum* is mainly considered a spike infecting pathogen but, although showing a contribution of FA in infecting this tissue, the paper gives demonstration of cell-to cell invasion only in wheat coleoptile.

Supplementary Fig. 1: the gene FG_13879 is co-regulated in wheat coleoptiles with the FG3_54 cluster and it is the most expressed gene at 64 hpi, but this result is not discussed. Considering that this transcriptomic analysis was previously published (Zhang et al., 2012), why did authors investigate 240 hpi and not time points closer to 64 hpi?

Line 139: *Nrps9* and *Nrps5* are both necessary for FA synthesis (Fig. 4a) but *nrps9* mutant shows a less dramatic reduction of virulence in comparison to *nrps5* mutant. Is there any explanation for these different phenotypes?

Supplementary Fig. 3a&b and Fig. 5 c&d: what is shown in the representative pictures of spike infection does not seem to correspond with data reported in the histogram (i.e. the counting of symptomatic spikelets). Usually spike infection data are reported as percentage of infected spikelets on the total number spikelets per spike. Symptoms are evaluated only at 14 dpi, what happens later? Pictures show a low level of symptoms also for WT although the high spore dose used for infection (10^6 conidia/ml). Besides, *Nrps5* and *Nrps9* mutants have not been assayed in spike infection although both these genes are important for FA biosynthesis.

Lines 315-319: the extraction protocol for the FA in plant tissue is not reported, besides in Fig. 4a&d it is not specified at what purification stage the chromatographic analysis was done. The concentration of FA detected in the plant tissue would be helpful to understand if the doses used to complement the cluster mutants are within the concentration range detected in the tissue.

Fig. 4c: the stereochemical representation of amino acid residues does not correspond to those claimed on the work. Besides, the previously reported D-Ser-D-Leu-D-Leuol tripeptide cannot be produced from the present "all-L" peptide sequence. The information in Fig. 9 is further supporting the adoption of D configuration for several amino acid residues of the native octapeptide.

Supplementary Fig. 9: it seems difficult to unambiguously assign D-allo Ile at position 3. The authors should provide stronger evidences supporting this assignment. Has it been obtained from the NMR study?

Minor critical points

Line 184: "hole-like structures" are not visible in the picture.

Line 280-281: nonsense that overexpression of fgm4 enhances the transcription of itself.
Line 250: the reason why the gene FG_06448 was selected as a possible regulator of the FA cluster is unclear.
Lines 291 and 681/683: "Fermentation" term is used inappropriately.

Response to Referees' Comments

We really appreciate comments from referees, therefore performed new experiments and revised our manuscript accordingly. A revised manuscript with changes highlighted is submitted and a point-by-point response to your comments is provided below.

Reviewer #3:

I've read the manuscript as well as the response to reviewers (listed as reviewer #2 in my version) - I think the authors have done an excellent job of addressing these original concerns (which were well founded, no doubt) - this has now strengthened the manuscript given that the octapeptide has now been identified and characterised in a comprehensive manner. I do not have any real concerns to report expect to say that the references for the NRPS section are very old and newer ones would add more value for the reader, given how much has been added to our knowledge of this field over the last 10-15 years.

Response: As the reviewer suggested, we have updated the references for the NRPS section which summarizing recent progresses in this field better. Two following review articles are included to replace the original Ref. 45 (Koglin & Walsh, 2009):

Gene H. Hur, Christopher R. Vickery and Michael D. Burkart. Explorations of catalytic domains in non-ribosomal peptide synthetase enzymology. *Nat. Prod. Rep.* **29**, 1074-1098 (2012)

Roderich D. Sgssmuth and Andi Mainz. Nonribosomal Peptide Synthesis-Principles and Prospects. *Angew. Chem. Int. Ed.* **56**, 3770-3821 (2017)

To support the hypothesis that “The elongation of the peptidyl chain might be terminated by module **M8**-mediated L-Leu incorporation, followed by R domain-catalyzed 4 electron reduction to release the resulting octapeptide from the assembly line as an alcohol” (the 1st paragraph of Page 15), we include a newly published literature as follow:

Michael W. Mullowney, Ryan A. McClure, Matthew T. Robey, Neil L. Kelleher, Regan J. Thomson. Natural products from thioester reductase containing biosynthetic pathways. *Nat. Prod. Rep.* **35**, 847-878 (2018)

Reviewer #4-1: *The paper by Lei-Jie et al. identifies a new virulence factor of F. graminearum infecting wheat. The paper is very original and provides unprecedented findings about the role of a non ribosomal octapeptide (FA) in supporting fungal spreading within host tissues. Methodological tools of fungal genetics, transcriptomic analyses, cellular microscopy, peptide characterization and synthesis are very appropriated and advanced.*

Major critical points that should be clarified:

The title deals with a general role of FA in “cell-to-cell invasion of F. graminearum in wheat”. F. graminearum is mainly considered a spike infecting pathogen but, although showing a contribution of FA in infecting this tissue, the paper gives demonstration of cell-to-cell invasion only in wheat coleoptile.

Response: We really appreciate your reviews and comments. We agree that previous version manuscript showed that fusaoctaxin A functions in both coleoptile and spike of

wheat infection by virulence assays and macro-scale symptom measurements, but at cellular level, only evidence in wheat coleoptiles were shown. At cellular level, *F. graminearum* hyphal invasion mainly can be differentiated to intercellular growth and cell-to-cell penetration. Due to the complex structure of wheat spike, including lemma, palea, pistil, anther, rachis, etc., which are tissues composed of multiple different types of cells, the growth patterns of *F. graminearum* hyphae in spike are more diverse (See figures below, scale bar=20 μ m), and various growth types may all contribute to infection outcome.

With focusing on hyphal growth in palea epidermal cells, we now additionally show that *FG- Δ fg3_54* mutant hyphae were less successful than wild-type hyphae in penetrating cell walls and invading neighbor cells, i.e. had defects in cell-to-cell invasion (Supplementary Fig. 5a).

The above representative pictures show mutant and wild-type *F. graminearum* hyphal growth in epidermal cells of wheat palea at 2.5 day post inoculation. White arrowheads indicate where hyphae penetrated cell walls, and red arrowheads indicate where hyphae failed to penetrate host cell wall, i.e. blocked at the edges of one cell. Comparing to *FG-Δfg3_54* mutant, the wild-type hyphae penetrate host cell wall continuously and more often. Nearly every branch of wild type hyphae got through host cell wall, while many branches of mutant hyphae were stopped by host cell wall. For example, the hypha indicated by the yellow line in the right image crossed the cell wall four times. This kind of penetration was rarely observed in the case of mutant hyphae. Although this is still not covering all the different types of hyphal growth in wheat spike, but as at least is one example of cell-to-cell growth difference in spike. With this additional data, we now show that fusaoctaxin A facilitates cell-to-cell invasion in two type of tissues in wheat, therefore supporting our title.

Reviewer #4-2: *Supplementary Fig. 1: the gene FG_13879 is co-regulated in wheat coleoptiles with the FG3_54 cluster and it is the most expressed gene at 64 hpi, but this result is not discussed. Considering that this transcriptomic analysis was previously published (Zhang et al., 2012), why did authors investigate 240 hpi and not time points closer to 64 hpi?*

Response: We apologize for making this mistake in data presentation. There was a misalignment between the gene name and the corresponding expression heat map. As indicated below (by the black lines), the FGSG_13879 was not expressed at all, the top-located up-regulated gene in this cluster was FGSG_13878 *Nrps5* in the cluster.

Previous version with lines mounted to show the alignment:

In this revision, we have corrected it as shown below.

Corrected version of Supplementary Fig. 1a:

Our previous work (Zhang et al., 2012) only studied *F. graminearum* stage-specific transcriptomes at 16, 40, 64 and 240 h post inoculation using laser-microdissected samples, due to the availability of homogenous infection cell populations. It is worthy to know more precise timing of fg3_54 cluster expression to infer timing of fusaoctaxin A function. We here used real-time PCR to investigate the expression levels of *nrps5* and *nrps9* when *F. graminearum* invades wheat coleoptiles at 12, 24, 48, 72, 96, 144 and 192 h post inoculation, and show that *nrps5* and *nrps9* expression reduction appeared since 72 h post inoculation relative to fungal constitutive expression genes (the charts below). Given that fungal biomass increased over infection time, the amount of fusaoctaxin A in wheat coleoptile may accumulate through late stages. We added the quantitative RT_PCR results in Supplementary Fig. 1b and the following sentence “Further quantitative PCR results

showed that *nrps5* and *nrps9* reduced expression after 72 hpi (Supplementary Fig. 1b).” in results part in this revision.

Reviewer #4-3: Line 139: *Nrps9* and *Nrps5* are both necessary for FA synthesis (Fig. 4a) but *nrps9* mutant shows a less dramatic reduction of virulence in comparison to *nrps5* mutant. Is there any explanation for these different phenotypes?

Response: It is intriguing to observe $\Delta nrps9$ mutant caused slightly larger lesion size than $\Delta nrps5$ on wheat coleoptiles, given that both *Nrps9* (with single module) and *Nrps5* (containing seven modules) are required for fusaoctaxin A biosynthesis.

As we showed in **Fig. 4C**, the FA assembly line is composed of NRPS9 and NRPS5, in which NRPS9 serves as a loading module to utilize GABA as a starter unit. The formation of the octapeptidyl skeleton of fusaoctaxin A depends primarily on NRPS5, which incorporates with GABA the other seven extender units composed of the residues L-Ala, L-allo-Ile, L-Ser, L-Val and L-Leu.

Mechanistically, NRPS5 might be flexible and tolerate different thioester-based starter units (In fact, substrate flexibility is characteristic for many NRPSs), thereby allowing for the production of diverse octapeptides that vary in the N-terminal residue. These octapeptides are minor and might contribute the remaining 30% activity in the FG- $\Delta nrps9$ mutant strain. This is a reason that we name the major GABA-started octapeptide found in this study as the “A” component, and the others could be components “B”, “C” and so on after characterization. Related works are on-going, and will be included in the future study focusing on the compatibility of the FA assembly line.

Reviewer #4-4: Supplementary Fig. 3a&b and Fig. 5 c&d: what is shown in the representative pictures of spike infection does not seem to correspond with data reported in the histogram (i.e. the counting of symptomatic spikelets). Usually spike infection data are reported as percentage of infected spikelets on the total number spikelets per spike. Symptoms are evaluated only at 14 dpi, what happens later? Pictures show a low level of symptoms also for WT although the high spore dose used for infection (10^6 conidia/ml). Besides, *Nrps5* and *Nrps9* mutants have not been assayed in spike infection although both these genes are important for FA biosynthesis.

Response: Thanks for your comments, but our measurements were consistent with representative pictures of spikes. We believe that this misunderstanding is due to the fact that only one side of the symptomatic spikelets was displayed when photographing, while we counted the symptomatic spikelets on both sides. As shown in the right, in the symptomatic spike that was photographed from the front or back, the two symptomatic spikelets were overlapping; but from side view, the two symptomatic spikelets can be distinguished. It is pretty common to show front view of symptomatic spike for researchers. So images of infected wheat spikes can only provide a rough idea of infection severity, while it is the charts of symptomatic spikelet numbers which can provide quantification.

We are aware of different ways to show the results of *F. graminearum* infection in wheat spike including the counting of symptomatic spikelets (e.g. Lofgren et al., 2018; Zhao et al., 2018) or the percentage of infected spikelets per spike (e.g. Goswami et al., 2005). We choose the middle spikelets as inoculation site in spike infection assay, and the number of symptomatic spikelet does not reach the total number of spikelets at 14 days post inoculation when we collect results. The hyphal spreading from the inoculated spikelet to the rest spikelets determines the symptom severity, and reflects the virulence of *F. graminearum*. Given that the total number of spikelets per spike varies among wheat cultivars, the absolute number of symptomatic spikelets reflects the fungal infection ability more directly than the percentage of symptomatic spikelets relative to the total number of spikelets per spike. Therefore we use the absolute number of symptomatic spikelets to evaluate virulence of various *F. graminearum* strains.

The infection assay of wheat spike we used in this paper is a well-established measurement method for quantification of disease severity, and 14 dpi a common time point for data collection (Hou et al., 2002). After 14 dpi, the grain of some spike filling completed and the healthy grains also turn dark yellow and dehydrated. It will be difficult to distinguish diseased spikelets and mature spikelets from color. In our assay system, 14 dpi is a good time point to observe the difference between wild type and mutant infection reproducibly.

Regarding the concentration of spore suspension for inoculation, we tried to use the 10^5 conidia/ml spore dose previously, but found the variation in disease severity was larger and caused lower data reproducibility than that using 10^6 conidia/ml in our lab system. So we chose 10^6 conidia/ml spore dose since Zhang et al., 2012, as the same as Wang et al., 2011. Disease can be more severe if conditions such as humidity can be increased. However, we think that the difference of infection ability between the mutant and the wild type can be measured with high reproducibility under our condition, and this condition serves our experimental purpose well.

In wheat coleoptile infection system, we showed the virulence of *nrps5* and *nrps9* mutants were impaired. In wheat spike infection, we didn't assay the virulence of *nrps5* and *nrps9* mutants, but we have showed that the *fg3_54* cluster deletion mutant reduced virulence (Fig. 1b). In addition, we showed that the addition of fusaoctaxin A, the product of *Nrps5* and *Nrps9*, resumed the virulence of *fg3_54* cluster deletion mutant to the wild type level (Fig. 5d,e). Therefore we think it is not necessary to assay the virulence of *nrps5* and *nrps9* mutants in spike infection, considering that we have identified the product of *Nrps5* and *Nrps9* as fusaoctaxin A (Fig. 4a-c), and the conclusion we like to make here is that fusaoctaxin A contributes to *F. graminearum* virulence in wheat spike infection.

References in this response:

- Goswami, R. S. & Kistler, H. C. Pathogenicity and In Planta Mycotoxin Accumulation Among Members of the *Fusarium graminearum* Species Complex on Wheat and Rice. *Phytopathology* 95, 1397–1404 (2005).
- Hou, Z. et al. A Mitogen-Activated Protein Kinase Gene (*MGV1*) in *Fusarium graminearum* Is Required for Female Fertility, Heterokaryon Formation, and Plant Infection. *Molecular Plant-Microbe Interactions* 15, 1119–1127 (2002).
- Lofgren, L. A. et al. *Fusarium graminearum*: pathogen or endophyte of North American grasses? *New Phytologist* 217, 1203–1212 (2018).
- Wang, C. et al. Functional analysis of the kinome of the wheat scab fungus *Fusarium graminearum*. *PLoS Pathogens* 7, e1002460 (2011).
- Zhao, L. et al. Cloning and characterization of a specific UDP-glycosyltransferase gene induced by DON and *Fusarium graminearum*. *Plant Cell Reports* 37, 641–652 (2018).

Reviewer #4-5: Lines 315-319: the extraction protocol for the FA in plant tissue is not reported, besides in Fig. 4a&d it is not specified at what purification stage the chromatographic analysis was done. The concentration of FA detected in the plant tissue would be helpful to understand if the doses used to complement the cluster mutants are within the concentration range detected in the tissue.

Response: Thanks for pointing this out. In our previous version, we roughly described the extraction protocol in lines 677-679 of the method section: “For detection of fusaoctaxin A in *F. graminearum*-wheat pathosystem, about 0.2 g of *F. graminearum* infected coleoptiles and infected spikes at 7 dpi were extracted using methanol under vacuum, and analyzed using Q-TOF LC-MS.” In the revision, we described the extraction protocol in more detail: “For detection of fusaoctaxin A in *F. graminearum*-wheat pathosystem, about 0.2 g of *F. graminearum*-infected coleoptiles and spikes at 7 dpi were immersed in methanol and subjected to vacuum extraction for 30 minutes. Then the extracts were evaporated and re-dissolved in 1 ml methanol. The dissolved samples were analyzed using the Agilent G6520A accurate-mass Q-TOF LC-MS system with an Agilent Zorbax column (SB-C18, 4.6 × 250 mm, 5 μm). The injected volume was 10 μL. The flow rate was 1 mL per min, and the gradient elution used mobile phase A (water supplemented with 0.1% HCOOH) and mobile phase B (acetonitrile supplemented with 0.1% HCOOH). The gradient profile was: 0 – 2 min (5% phase B), 2 – 20 min (5% to 90% phase B), 20 - 25 min (90% phase B),

25 – 25.5 min (90% to 5% phase B), and 25.5 – 30 min (5% phase B). The mass range was 150–1500 m/z ; nebulizer pressure 40 psig; drying gas N₂ 350°C, 9 L/min; ESI Vcap 3500 V, fragmentor 160 V; skimmer 65 V; and Oct RF Vpp 750 V.”

In Fig. 4a&d, we used LC-MS to detect crude extracts of in vitro culture of different strains and crude extracts of plant tissues infected by *Fusarium graminearum* without any purification steps. The chromatogram shown at the left is not total ion current (TIC), but extracted-ion chromatogram at m/z 773.5 ± 0.5 and 773.51 ± 0.5 (EIC).

We appreciate your suggestion on examining FA concentration in plant tissues. Following this advice, we used liquid chromatography tandem mass spectrometry and standard fusaotaxin A to measure the concentration of endogenous fusaotaxin A in the wheat coleoptiles. Approximately 0.3 and 0.7 nmol per coleoptile (i.e. 260 and 480 ng per coleoptile) endogenous fusaotaxin A were detected

in wild-type *F. graminearum* infected wheat coleoptiles collected at 3 dpi and 7 dpi, respectively. [Given that the volume per coleoptile is approximately 12.5 μL , and the fresh weight per coleoptile is approximately 8 mg, the concentration of endogenous fusaotaxin A is approximately 50 μM or 60 $\mu\text{g/g}$ fresh weight in coleoptile at 7 dpi.] Endogenous fusaotaxin A was not detected in mock-inoculated or *Afg3_54* mutant-inoculated coleoptiles of wheat. We showed that addition of 3 nmol per seedling exogenous fusaotaxin A complemented the size of lesion caused by *Afg3_54* mutant to that of wild-type (Fig. 5c). We considered that fusaotaxin A that applied to wheat seedlings may enter central leaf as well as coleoptile, and some may even be kept at surface without entering (See representative picture of a wheat seedling on the right). So only partial of added fusaotaxin A actually enters coleoptile. Therefore we measured the amount of fusaotaxin A in coleoptiles at 3 h after the application of 3 nmol fusaotaxin A per seedling, and result shows (below) approximately 1.2 nmol per coleoptile fusaotaxin A was detected in the coleoptile with 3 nmol fusaotaxin A added. This proves our above idea. The endogenous fusaotaxin A concentration (0.7 nmol per coleoptile) is in the same order of magnitude as the fusaotaxin A concentration that complements *Afg3_54* mutant virulence (1.2 nmol per coleoptile). Given that exogenous application fusaotaxin A might be more or less uniform, while the fusaotaxin A produced by the fungus can be more concentrated in the plant–fungal interaction interface where this molecule actually works, we think the amount of fusaotaxin A we added into the mutant infection is mimicking the endogenous fusaotaxin A level in wild type infection. In brief, the doses used to complement the cluster mutants are within the concentration range detected in the tissue. Therefore we include this piece of data in our revision (Fig. 5a), supporting that the working concentration of fusaotaxin A is in the range around 0.7 nmol per coleoptile (i.e. 50 μM).

With this additional result, we can estimate the endogenous working concentration of fusaotaxin A is 50 μ M, and this is in the same order of magnitude as the working concentration of some plant CLE peptide hormones, including CLV3 in shoot apical meristem development regulation and KIN in vascular tissue development regulation, as reported by Hirakawa et al., (2017).

Hirakawa, Y. et al. Cryptic bioactivity capacitated by synthetic hybrid plant peptides. Nature Communications 8, 14318 (2017).

Reviewer #4-6: Fig. 4c: the stereochemical representation of amino acid residues does not correspond to those claimed on the work. Besides, the previously reported D-Ser-D-Leu-D-Leuol tripeptide cannot be produced from the present “all-L” peptide sequence. The information in Fig. 9 is further supporting the adoption of D configuration for several amino acid residues of the native octapeptide.

Response: Fig. 4c indicates the domain/module organization of the assembly line composed of NRPS9 and NRPS5, the polymerization process and the associated amino acid monomers in the biosynthesis of fusaotaxin A. Although all monomers possess an L-configuration, the modules M3, M4, M5, M6 and M7 within the assembly line share a C-terminal epimerization (E) domain, which catalyzes an epimerization reaction to produce a D-configuration after each elongation, and thereby can convert the substrate residues L-*allo*-Ile₃, L-Ser₄, L-Val₅, L-Ser₆ and L-Leu₇ into D-*allo*-Ile₃, D-Ser₄, D-Val₅, D-Ser₆ and D-Leu₇, respectively, in the resulting octapeptide product. As the reviewer pointed out, the information in **Supplementary Fig. 9** (and **Supplementary Fig. 8**) supports the adoption of D-configuration for these amino acid residues of fusaotaxin A, and thus the composition and residue organization of this octapeptide are well consistent with the catalytic logic of the assembly line composed of NRPS9 and NRPS5. **The previously reported tripeptide is identical to the C-terminal three residues of fusaotaxin A in sequence**, and thus appears to be a degradation product of this octapeptide.

Please see the related description in the main text as below “Overall, the composition and residue organization of fusaotaxin A are well consistent with the catalytic logic of the assembly line composed of NRPS9 and NRPS5, which likely utilizes GABA as a starter unit and sequentially incorporates seven extender units composed of the residues L-Ala,

L-*allo*-Ile, L-Ser, L-Val and L-Leu. During the process, each of the residues that are tethered on modules (**M3-M7**) containing an E domain can undergo an epimerization reaction to produce a D-configuration before the transpeptidation reaction occurs. The elongation of the peptidyl chain might be terminated by module **M8**-mediated L-Leu incorporation, followed by R domain-catalyzed 4 electron reduction to release the resulting octapeptide from the assembly line as an alcohol (Fig. 4c).”

Reviewer #4-7: *Supplementary Fig. 9: it seems difficult to unambiguously assign D-*allo* Ile at position 3. The authors should provide stronger evidences supporting this assignment. Has it been obtained from the NMR study?*

Response: Yes, the assignment of D-*allo*-Ile₃ has been obtained from detailed 1D and 2D-NMR studies as well as comparative analyses using both standard amino acids and the synthesized octapeptide. Please see **Supplementary Fig. 8**, **Supplementary Fig. 9**, and Fusaotaxin A characterization source data file. First, 1D and 2D-NMR spectra-based analyses revealed that the residue 3 of fusaotaxin A is an Ile, which has two chiral centers (at the α and β positions, respectively). Second, using various amino acid standards, the hydrolysis of fusaotaxin A followed by the derivatization with FDAA established the stereochemistry at the α position of this residue (C9), indicating that it is a D-Ile. Next, the stereochemistry of this residue at the β position was determined based on ¹H NMR coupling and ROESY correlation. Specifically, as shown in **Supplementary Fig. 8h**, the large coupling constant $^3J_{\text{H-H}} = 5.3$ Hz indicates the anti-orientation between H-9_{Ile} and H-10_{Ile}, and the key ROESY correlations of H-10/15-NH, H-12/9-NH and H-9/H-13 reveal that the relative configuration between C-9 and C-10 is *erythro*. Given the *R*-configuration at the α position (C9), this residue was established to be a D-*allo*-Ile with the *S*-configuration at the β position. Finally, this conclusion was further confirmed by the comparison between the naturally purified fusaotaxin A and the chemically synthesized octapeptide in the 1D and 2D-NMR spectra, in the latter of which residue 3 arises from the standard amino acid D-*allo*-Ile (**Supplementary Fig. 9h**).

Reviewer #4-8 Minor critical points

Line 184: “hole-like structures” are not visible in the picture.

Response: We increased the presentation area and used an arrow to point to the “hole-like” structure in Fig. 2c in light field (the same as below left). More examples are shown below middle panel. In addition, the “hole-like” structure can also be seen in Supplementary Fig. 5c (the same as below right panel).

Line 280-281: nonsense that overexpression of fgm4 enhances the transcription of itself.

Response: We agree with the review, and revised accordingly to “overexpressing fgm4 selectively enhanced the transcription of the fg3_54 cluster genes (including nrps5 and nrps9)”.

Line 250: the reason why the gene FG_06448 was selected as a possible regulator of the FA cluster is unclear.

Response: We originally considered FGSG_06448 as a candidate regulator of fusaoctaxin A producing cluster for two reasons: 1. FGSG_06448 expression significantly higher in 64 hai sample than all other wheat coleoptile infection and in vitro growth samples, share the co-expression pattern with fg3_54 cluster genes; 2. FGSG_06448 encodes a putative transcription factor. We explored whether FGSG_06448 is responsible for inducing fg3_54 cluster gene expression. However, this possibility was ruled out because deletion of FGSG_06448 didn't cause reduction in virulence (if it was required for induction of fg3_54 cluster expression, deletion of FGSG_06448 would not be able to induce fg3_54 gene expression during wheat infection, therefore should show defects in virulence as fg3_54 deletion mutants. This was only an unsuccessful trial in identifying fg3_54 regulator. In order to be concise, we omitted the part of FGSG_06448 in this revision.

Lines 291 and 681/683: “Fermentation” term is used inappropriately.

Response: We agree with the review, and revised accordingly to “We then scaled-up the WT-OE::fgm4 culture in vitro”.

REVIEWERS' COMMENTS:

Reviewer #4 (Remarks to the Author):

[No comments for author.]